# Generalization and Robustness of the Tilted Empirical Risk

**Gholamali Aminian** [1]  **Amir R. Asadi** [2]  **Tian Li** [3]  **Ahmad Beirami** [4]  **Gesine Reinert** [1 5]  **Samuel N. Cohen** [1 6]

## Abstract

The generalization error (risk) of a supervised statistical learning algorithm quantifies its prediction ability on previously unseen data. Inspired by exponential tilting, Li et al. (2021) proposed the *tilted empirical risk* (TER) as a non-linear risk metric for machine learning applications such as classification and regression problems. In this work, we examine the generalization error of the tilted empirical risk in the robustness regime under *negative tilt*. Our first contribution is to provide uniform and information-theoretic bounds on the *tilted generalization error*, defined as the difference between the population risk and the tilted empirical risk, under negative tilt for unbounded loss function under bounded $(1 + \epsilon)$-th moment of loss function for some $\epsilon \in (0, 1]$ with a convergence rate of $O(n^{-\epsilon/(1+\epsilon)})$ where $n$ is the number of training samples, revealing a novel application for TER under no distribution shift. Secondly, we study the robustness of the tilted empirical risk with respect to noisy outliers at training time and provide theoretical guarantees under distribution shift for the tilted empirical risk. We empirically corroborate our findings in simple experimental setups where we evaluate our bounds to select the value of tilt in a data-driven manner.

## 1. Introduction

Empirical risk minimization (ERM) is a popular framework in machine learning. The performance of the empirical risk (ER) is affected when the data set is strongly imbalanced or contains outliers. For these scenarios, inspired by the log-sum-exponential operator with applications in multinomial linear regression and naive Bayes classifiers (Calafiore et al., 2019; Murphy, 2012; Williams & Barber, 1998), Li et al. (2021) propose the tilted empirical risk (TER) for supervised learning applications, such as classification and regression problems. Li et al. (2021; 2023a) showed that tilted empirical risk minimization (TERM) can handle class imbalance, mitigate the effect of outliers, and enable fairness between subgroups. Different applications of TERM have been explored, e.g., differential privacy (Lowy & Razaviyayn, 2021), semantic segmentation (Szabó et al., 2021), noisy label self-correction (Zhou et al., 2020), and off-policy learning and evaluation (Behnamnia et al., 2025).[1] In this paper, we corroborate the empirical success of the TERM framework through statistical learning theory.

A central concern in statistical learning theory is understanding the efficacy of a learning algorithm when applied to *test* data. This evaluation is typically carried out by investigating the *generalization error*, which quantifies the disparity between the performance of the algorithm on the training dataset and its performance on previously unseen data, drawn from the same underlying distribution, via a risk function. Understanding the generalization behaviour of learning algorithms is one of the most important objectives in statistical learning theory. Various approaches have been developed for empirical risk (Rodrigues & Eldar, 2021), including VC dimension-based bounds (Vapnik, 1999), stability-based bounds (Bousquet & Elisseeff, 2002b), PAC Bayesian bounds (McAllester, 2003), and information-theoretic bounds (Russo & Zou, 2019; Xu & Raginsky, 2017). This paper focuses on the generalization and robustness of the tilted empirical risk (tilted generalization error) of learning algorithms. Our contributions are:

- In Section 3, we provide upper and lower bounds on the tilted generalization error under *unbounded loss* functions for the negative tilt with a bound via uniform and information-theoretical approaches and establish the convergence rate of $O(n^{-\epsilon/(1+\epsilon)})$ for some $\epsilon \in (0, 1]$.

- In Section 4, we study the robustness of the tilted empirical risk under distribution shift induced by noise or outliers for unbounded loss functions with bounded

---

[1]The Alan Turing Institute, London, UK [2]Statistical Laboratory, University of Cambridge, Cambridge, UK [3] Computer Science Department, University of Chicago, USA [4]Google DeepMind, USA [5]Department of Statistics, University of Oxford, Oxford, UK [6]Mathematical Institute, University of Oxford, Oxford, UK. Correspondence to: Gholamali Aminian <gaminian@turing.ac.uk>.

*Proceedings of the 42$^{nd}$ International Conference on Machine Learning*, Vancouver, Canada. PMLR 267, 2025. Copyright 2025 by the author(s).

---

[1]Behnamnia et al. (2025) use the term log-sum-exponential, which is the same as the tilted empirical risk studied herein.

$(1+\epsilon)$-th moment assumption, for some $\epsilon \in (0,1]$, and negative tilt, and derive generalization bounds.

- In Section 5, we provide a data-driven approach to selecting the value of tilt for regression problems, with observed test errors corroborating the theoretical bounds.

- In Section 6, we study the KL-regularized TERM problem under unbounded loss function with bounded second moment assumption and provide an upper bound on the expected tilted generalization error with convergence rate $O(n^{-\epsilon})$ under unbounded loss function with bounded $(1+\epsilon)$-th moment for some $\epsilon \in (0,1]$.

## 2. Preliminaries

**Notations:** Upper-case letters denote random variables (e.g., $Z$), lower-case letters denote the realizations of random variables (e.g., $z$), and calligraphic letters denote sets (e.g., $\mathcal{Z}$). All logarithms are in the natural base. The tilted expectation of a random variable $X$ with tilting $\gamma$ is defined as $\frac{1}{\gamma} \log(\mathbb{E}[\exp(\gamma X)])$. The set of probability distributions (measures) over a space $\mathcal{X}$ with finite variance is denoted by $\mathcal{P}(\mathcal{X})$.

**Information measures:** For two probability measures $P$ and $Q$ defined on the space $\mathcal{X}$, such that $P$ is absolutely continuous with respect to $Q$, the *Kullback-Leibler* (KL) divergence between $P$ and $Q$ is $\mathrm{KL}(P\|Q) := \int_{\mathcal{X}} \log(\mathrm{d}P/\mathrm{d}Q)\,\mathrm{d}P$. If $Q$ is also absolutely continuous with respect to $P$, then the *symmetrized KL divergence* is $D_{\mathrm{SKL}}(P\|Q) := \mathrm{KL}(P\|Q) + \mathrm{KL}(Q\|P)$. The mutual information between two random variables $X$ and $Y$ is defined as the KL divergence between the joint distribution and the product-of-marginal distribution $I(X;Y) := \mathrm{KL}(P_{X,Y}\|P_X \otimes P_Y)$, where $P_X \otimes P_Y$ is the product of the marginal distributions. The *conditional KL divergence* between $P_{Y|X}$ and $P_Y$ over $P_X$ is $\mathrm{KL}(P_{Y|X}\|P_Y|P_X) := \int_{\mathcal{X}} \mathrm{KL}(P_{Y|X=x}\|P_Y)dP_X(x)$. The *symmetrized KL information* between $X$ and $Y$ is given by $I_{\mathrm{SKL}}(X;Y) := D_{\mathrm{SKL}}(P_{X,Y}\|P_X \otimes P_Y)$; see Aminian et al. (2015). The *total variation distance* between two densities $P$ and $Q$ is defined as $\mathbb{TV}(P,Q) := \int_{\mathcal{X}} |\mathrm{d}P - \mathrm{d}Q|$.

### 2.1. Problem Formulation

Let $S = \{Z_i\}_{i=1}^n$ be the training set, where each sample $Z_i = (X_i, Y_i)$ belongs to the instance space $\mathcal{Z} := \mathcal{X} \times \mathcal{Y}$; here $\mathcal{X}$ is the input (feature) space and $\mathcal{Y}$ is the output (label) space. We assume that $Z_i$ are i.i.d. generated from the same data-generating distribution $\mu$. We also assume $\tilde{Z} \sim \mu$ as an i.i.d. sample with respect to the training set.

A set of hypotheses $\mathcal{H}$ has elements $h : \mathcal{X} \mapsto \mathcal{Y} \in \mathcal{H}$. When $\mathcal{H}$ is finite, then its cardinality is denoted by $\mathrm{card}(\mathcal{H})$.

In order to measure the performance of the hypothesis $h$, we use a non-negative loss function $\ell : \mathcal{H} \times \mathcal{Z} \to \mathbb{R}_0^+$.

We apply different methods to study the performance of our algorithms, including uniform and information-theoretic approaches. In uniform approaches, such as the VC-dimension and the Rademacher complexity approach (Vapnik, 1999; Bartlett & Mendelson, 2002), the hypothesis space $\mathcal{H}$ is independent of the learning algorithm. Therefore, these methods are algorithm-independent; our results for these methods do not specify the learning algorithms.

**Learning Algorithms:** For information-theoretic approaches in supervised learning, following Xu & Raginsky (2017), we consider learning algorithms that are characterized by a Markov kernel (a conditional distribution) $P_{H|S}$. Such a learning algorithm maps a data set $S$ to a hypothesis in $\mathcal{H}$, which is chosen according to $P_{H|S}$. This concept thus includes randomized learning algorithms.

### 2.2. Risk Functions

The main quantity we are interested in is the *population risk*, defined by

$$\mathrm{R}(h,\mu) := \mathbb{E}_{\tilde{Z} \sim \mu}[\ell(h, \tilde{Z})], \quad h \in \mathcal{H}.$$

As the distribution $\mu$ is unknown, in classical statistical learning, the (true) population risk for $h \in \mathcal{H}$ is estimated by the (linear) *empirical risk*

$$\widehat{\mathrm{R}}(h,S) = \frac{1}{n} \sum_{i=1}^n \ell(h, Z_i). \tag{1}$$

The *generalization error* for the linear empirical risk is given by

$$\mathrm{gen}(h,S) := \mathrm{R}(h,\mu) - \widehat{\mathrm{R}}(h,S). \tag{2}$$

This is the difference between the true risk and the linear empirical risk. The *TER*, as a non-linear empirical risk with tilt $\gamma$ (Li et al., 2021), a.k.a. log-sum-exponential, estimates the population risk by

$$\widehat{\mathrm{R}}_\gamma(h,S) = \frac{1}{\gamma} \log\left(\frac{1}{n} \sum_{i=1}^n \exp\left(\gamma \ell(h, Z_i)\right)\right).$$

The TER is an increasing function in the tilt parameter $\gamma$ (Li et al., 2023a, Theorem 1), and as $\gamma \to 0$, the TER converges to the linear empirical risk in (1). In this work, we focus on negative tilt $\gamma < 0$. Inspired by Li et al. (2021), the primary objective is to optimize the population risk; the TERM is used in order to help the learning dynamics. For our analysis, we decompose the population risk as follows:

$$\mathrm{R}(h,\mu) = \underbrace{\mathrm{R}(h,\mu) - \widehat{\mathrm{R}}_\gamma(h,S)}_{\text{tilted generalization error}} + \underbrace{\widehat{\mathrm{R}}_\gamma(h,S)}_{\text{tilted empirical risk}}, \tag{3}$$

where we define the *tilted generalization error* as

$$\text{gen}_\gamma(h, S) := \text{R}(h, \mu) - \widehat{\text{R}}_\gamma(h, S). \tag{4}$$

In learning theory, for uniform approaches, most works focus on bounding the linear generalization error $\text{gen}(h, S)$ from (2) such that under the distribution of the dataset $S$, with probability at least $(1 - \delta)$, it holds that for all $h \in \mathcal{H}$,

$$|\text{gen}(h, S)|| \leq g(\delta, n), \tag{5}$$

where $g$ is a real function dependent on $\delta \in (0, 1)$ and $n$ is the number of data samples. Similarly, for the tilted generalization error from (4), we are interested in finding a bound $g_t(\delta, n, \gamma)$ such that with probability at least $1 - \delta$, under the distribution of $S$,

$$\left|\text{gen}_\gamma(h, S)\right| \leq g_t(\delta, n, \gamma), \tag{6}$$

where $g_t$ is a real function. We set $h^*(\mu) := \arg\min_{h \in \mathcal{H}} \text{R}(h, \mu)$ and $h_\gamma^*(S) := \arg\min_{h \in \mathcal{H}} \hat{\text{R}}_\gamma(h, S)$. We use the following notations. The excess risk under the tilted empirical risk is

$$\mathfrak{E}_\gamma(\mu) := \text{R}(h_\gamma^*(S), \mu) - \text{R}(h^*(\mu), \mu). \tag{7}$$

The expected TER with respect to the distribution of $S$ is

$$\overline{\text{R}}_\gamma(h, \mu^{\otimes n}) = \mathbb{E}_{\mu^{\otimes n}}[\widehat{\text{R}}_\gamma(h, S)], \tag{8}$$

and the tilted (true) population risk is

$$\begin{aligned} &\text{R}_\gamma(h, \mu^{\otimes n}) \\ &= \frac{1}{\gamma} \log\left(\mathbb{E}_{\mu^{\otimes n}}\left[\frac{1}{n}\sum_{i=1}^{n}\exp(\gamma\ell(h, Z_i))\right]\right). \end{aligned} \tag{9}$$

Under the i.i.d. assumption, the tilted population risk is equal to an entropic risk function (Howard & Matheson, 1972). We also introduce the non-linear generalization error[2], as

$$\widehat{\text{gen}}_\gamma(h, S) := \text{R}_\gamma(h, \mu^{\otimes n}) - \widehat{\text{R}}_\gamma(h, S). \tag{10}$$

For ease of notation, we consider $\text{R}_\gamma(h, \mu^{\otimes n}) = \text{R}_\gamma(h, \mu)$.

**Information-theoretic Approach:** For the information-theoretic approach, as the hypothesis $H$ is a random variable under a learning algorithm as Markov kernel, i.e., $P_{H|S}$, we take expectations over the hypothesis $H$. We denote the expected true risk, expected empirical risk, expected

---

[2]We refer to this as non-linear generalization error since a non-linear transformation of the population risk, instead of the population risk, is used.

entropic risk, and expected tilted generalization error by

$$\begin{aligned} \text{R}(H, P_H \otimes \mu) &:= \mathbb{E}_{P_H \otimes \mu}[\ell(H, Z)], \\ \overline{\text{R}}_\gamma(H, Q_{H,S}) &:= \mathbb{E}_{Q_{H,S}}[\widehat{\text{R}}_\gamma(H, S)], \\ \text{R}_\gamma(H, Q_{H,S}) &:= \\ &\frac{1}{\gamma}\log\left(\mathbb{E}_{Q_{H,S}}[\frac{1}{n}\sum_{i=1}^{n}\exp(\gamma\ell(H, Z_i))]\right) \\ \overline{\text{gen}}_\gamma(H, S) &:= \mathbb{E}_{P_{H,S}}[\text{gen}_\gamma(H, S)], \end{aligned} \tag{11}$$

where $Q_{H,S} \in \{P_H \otimes \mu^{\otimes n}, P_{H,S}\}$. In addition to bounds of the form (6), we provide upper bounds on the absolute value of expected tilted generalization error with respect to the joint distribution of $S$ and $H$, of the form

$$|\overline{\text{gen}}_\gamma(H, S)| \leq g_e(n, \gamma),$$

where $g_e$ is a real function. We also introduce the non-linear expected generalization error, which plays an important role in deriving our bounds, as

$$\widehat{\text{gen}}_\gamma(H, S) := \text{R}_\gamma(H, P_H \otimes \mu^{\otimes n}) - \text{R}_\gamma(H, P_{H,S}). \tag{12}$$

# 3. Generalization Bounds for Unbounded Loss Functions

In this section, we derive upper bounds on the tilted generalization error via uniform and information-theoretical approaches for negative tilt ($\gamma < 0$) under bounded $(1 + \epsilon)$-th moment of loss function, for some $\epsilon \in (0, 1]$, with convergence rate of $O(n^{-\epsilon/(1+\epsilon)})$. Regarding the bounded loss function, we also provide the tilted generalization error bounds via uniform and information-theoretical approaches in Appendix H. Note that, with a bounded loss function, we can exploit the Lipschitz property of the logarithmic function. However, when dealing with unbounded losses, the loss function lacks the Lipschitz property, making it impossible to apply the same techniques used in the bounded case (see Appendix H).

Several works have already proposed some solutions to overcome the boundedness assumption under *linear empirical risk* (Haddouche & Guedj, 2022; Alquier & Guedj, 2018; Holland, 2019) via a PAC-Bayesian approach under the bounded second-moment assumption. Furthermore, an upper bound on generalization error via VC-dimension and growth function is proposed in Cortes et al. (2019, Corollary 12) with convergence rate of $O(\log(n)n^{-\epsilon/(1+\epsilon)})$. In contrast, we derive bounds with a convergence rate of $O(n^{-\epsilon/(1+\epsilon)})$. A more detailed comparison is provided in Section 7.

## 3.1. Uniform Bounds

The following assumption is made for the uniform analysis.

**Assumption 3.1** (Uniform bounded $(1 + \epsilon)$-th moment[3])**.** There is a constant $\kappa_u \in \mathbb{R}^+$ such that the loss function $(H, Z) \mapsto \ell(H, Z)$ satisfies $\mathbb{E}_\mu[\ell^{1+\epsilon}(h, Z)] \leq \kappa_u^{1+\epsilon}$ uniformly for all $h \in \mathcal{H}$ and some $\epsilon \in (0, 1]$.

The assumption on the $(1 + \epsilon)$-th moment, Assumption 3.1, is satisfied for example if the loss function is sub-Gaussian or sub-Exponential under the distribution $\mu$ for all $h \in \mathcal{H}$, see Boucheron et al. (2013). Inspired by the approach of Behnamnia et al. (2025), which provides a regret bound based on a tilted operator, we aim to analyze the generalization error of tilted empirical risk within a uniform approach. All proof details for the results in this section are deferred to Appendix D.1.

For uniform bounds of the type (6), we decompose the tilted generalization error (4) as follows,

$$\mathrm{gen}_\gamma(h, S) = \underbrace{\mathrm{R}(h, \mu) - \mathrm{R}_\gamma(h, \mu^{\otimes n})}_{I_1} + \widehat{\mathrm{gen}}_\gamma(h, S), \quad (13)$$

We first derive an upper bound on term $I_1$ in the following Proposition.

**Proposition 3.2.** *Under Assumption 3.1, for $\gamma < 0$ and some $\epsilon \in (0, 1]$, the difference between the population risk and the tilted population risk satisfies*

$$0 \leq \mathrm{R}(h, \mu) - \mathrm{R}_\gamma(h, \mu^{\otimes n}) \leq |\gamma|^\epsilon \kappa_u^{1+\epsilon}. \quad (14)$$

Then, using Bernstein's inequality (Boucheron et al., 2013) and properties of the logarithm, we can provide upper and lower bounds on the tilted generalization error.

**Proposition 3.3.** *Given Assumption 3.1 for some $\epsilon \in (0, 1]$, for any fixed $h \in \mathcal{H}$ with probability at least $(1 - \delta)$, then the following upper bound holds on the tilted generalization error for $\gamma < 0$ and some $\epsilon \in (0, 1]$,*

$$\mathrm{gen}_\gamma(h, S) \leq \frac{2\exp(|\gamma|\kappa_u)}{|\gamma|}\sqrt{\frac{2^\epsilon|\gamma|^{1+\epsilon}\kappa_u^{1+\epsilon}\log(2/\delta)}{n}}$$
$$+ \frac{4\exp(|\gamma|\kappa_u)\log(2/\delta)}{3n|\gamma|} + |\gamma|^\epsilon\kappa_u^{1+\epsilon}.$$

**Proposition 3.4.** *Given Assumption 3.1 for some $\epsilon \in (0, 1]$, there exists a $\zeta \in (0, 1)$ such that for $n \geq \frac{(4\gamma^2\kappa_u^2 + 8/3\zeta)\log(2/\delta)}{\zeta^2\exp(2\gamma\kappa_u)}$, for any fixed $h \in \mathcal{H}$ with probability at least $(1 - \delta)$, and $\gamma < 0$, the following lower bound on the tilted generalization error holds,*

$$\mathrm{gen}_\gamma(h, S) \geq -\frac{2\exp(|\gamma|\kappa_u)}{(1 - \zeta)|\gamma|}\sqrt{\frac{2^\epsilon|\gamma|^{1+\epsilon}\kappa_u^{1+\epsilon}\log(2/\delta)}{n}}$$
$$- \frac{4\exp(|\gamma|\kappa_u)(\log(2/\delta))}{3n|\gamma|(1 - \zeta)}.$$

---

[3]Note that we assume that higher order moments, larger than $(1 + \epsilon)$, are unbounded.

Combining Proposition 3.3 and Proposition 3.4, we derive an upper bound on the absolute value of the tilted generalization error.

**Theorem 3.5.** *Under the same assumptions in Proposition 3.4 and a finite hypothesis space, then for $n \geq \frac{(4|\gamma|^{1+\epsilon}\kappa_u^{1+\epsilon} + 8/3\zeta)\log(2/\delta)}{\zeta^2\exp(2\gamma\kappa_u)}$, for $\gamma < 0$ and with probability at least $(1-\delta)$, the absolute value of the titled generalization error satisfies*

$$\sup_{h\in\mathcal{H}}|\mathrm{gen}_\gamma(h, S)| \quad (15)$$

$$\leq \frac{2\exp(|\gamma|\kappa_u)}{(1 - \zeta)|\gamma|}\sqrt{\frac{2^\epsilon|\gamma|^{1+\epsilon}\kappa_u^{1+\epsilon}B(\delta)}{n}} + \frac{4\exp(|\gamma|\kappa_u)B(\delta)}{3n|\gamma|(1 - \zeta)}$$
$$+ |\gamma|^\epsilon\kappa_u^{1+\epsilon},$$

*where $B(\delta) = \log(\mathrm{card}(\mathcal{H})) + \log(2/\delta)$.*

*Remark* 3.6. For $\gamma \asymp n^{-1/(1+\epsilon)}$, $n \geq \frac{(4\kappa_u^{1+\epsilon} + 8/3\zeta)\log(2/\delta)}{\zeta^2\exp(-2\kappa_u)}$ and $-1 < \gamma < 0$, the upper bound in Theorem 3.5 gives a theoretical guarantee on the convergence rate of $O(n^{-\epsilon/(1+\epsilon)})$. Using TER with negative tilt can help to derive an upper bound on the absolute value of the tilted generalization error under the bounded $(1 + \epsilon)$-th moment assumption.

The following Lemma establishes an upper bound on the excess risk of tilted empirical risk, expressed in terms of $\sup_{h\in\mathcal{H}}|\mathrm{gen}_\gamma(h, S)|$.

**Lemma 3.7.** *The excess risk of the tilted empirical risk satisfies,*

$$\mathfrak{E}_\gamma(\mu) \leq 2\sup_{h\in\mathcal{H}}|\mathrm{gen}_\gamma(h, S)|. \quad (16)$$

Combining Lemma 3.7 with Theorem 3.5, an upper bound on the excess risk of the tilted empirical risk can be derived (See Appendix D.1).

The theorems in this section assume that the hypothesis space is finite; this is, for example, the case in classification problems with a finite number of classes. If this assumption is violated, we can apply the growth function technique from Bousquet et al. (2003); Vapnik (1999). In particular, the growth function can be bounded by VC-dimension in binary classification (Vapnik, 1999) or Natarajan dimension (Holden & Niranjan, 1995) for multi-class classification scenarios. Note that the VC-dimension and Rademacher complexity bounds are uniform bounds and are independent of the learning algorithms. Furthermore, one can construct an $\epsilon$-net over the hypothesis space, thereby discretizing the space. In this construction, we select a finite subset $H' \subset \mathbb{R}^m$ such that for every $h \in H$, there exists a $h' \in H'$ with $\|h - h'\| \leq r$. By applying our finite hypothesis result to this discretized set and controlling the approximation error through the Lipschitz property of the loss function, we effectively generalize our bounds to the continuous case. This

method is exemplified in (Xu & Raginsky, 2017), which demonstrates that "for an uncountable hypothesis space, we can always convert it to a finite one by quantizing the output of the smallest set $H'$ such that for all $h \in H$ there is a $h' \in H'$ with $\|h - h'\| \leq r$, where the Lipschitz maximal inequality (Lemma 5.7 in (Vershynin, 2010)) is derived using a similar quantization technique.

## 3.2. Information-theoretic Bounds

In the information-theoretic approach for the unbounded loss function, we relax the uniform assumption, Assumption 3.1, as follows,

**Assumption 3.8** (Bounded $(1 + \epsilon)$-th moment)**.** The learning algorithm $P_{H|S}$, loss function $\ell$, and $\mu$ are such that there is a constant $\kappa_t \in \mathbb{R}^+$ with which the loss function $(H, Z) \mapsto \ell(H, Z)$ satisfies $\max\left(\mathbb{E}_{P_{H,Z}}[\ell^\epsilon(H, Z)], \mathbb{E}_{P_H \otimes \mu}[\ell^{1+\epsilon}(H, Z)]\right) \leq \kappa_t^{1+\epsilon}$ for some $\epsilon \in (0, 1]$.

For our information-theoretical analysis, we use the following decomposition;

$$\overline{\mathrm{gen}}_\gamma(H, S) = \underbrace{\mathrm{R}(H, P_H \otimes \mu) - \mathrm{R}_\gamma(H, P_H \otimes \mu)}_{I_3}$$
$$+ \widehat{\mathrm{gen}}_\gamma(H, S) \qquad (17)$$
$$+ \underbrace{\mathrm{R}_\gamma(H, P_{H,S}) - \mathbb{E}_{P_{H,S}}[\widehat{\mathrm{R}}_\gamma(H, S)]}_{I_4}.$$

We can derive bounds on $I_3$ and $I_4$ in a similar approach to Proposition 3.2.

Then, we provide upper and lower bounds on the non-linear expected generalization error of the tilted empirical risk, $\widehat{\mathrm{gen}}_\gamma(H, S)$, under negative tilt. These results are helpful in deriving an upper bound on the absolute expected tilted generalization error.

**Proposition 3.9.** *Given Assumption 3.8 and assuming that* $\frac{2I(H;S)}{2^\epsilon |\gamma|^{1+\epsilon} \kappa_t^{1+\epsilon}} \leq n$ *holds, then the following inequality holds,*

$$\widehat{\mathrm{gen}}_\gamma(H, S) \leq \frac{\exp(|\gamma|\kappa_t)}{|\gamma|} \sqrt{\frac{2^\epsilon |\gamma|^{1+\epsilon} \kappa_t^{1+\epsilon} I(H; S)}{n}},$$

*and if* $\frac{2I(H;S)}{2^\epsilon |\gamma|^{1+\epsilon} \kappa_t^{1+\epsilon}} > n$ *holds, then following inequality holds,*

$$\widehat{\mathrm{gen}}_\gamma(H, S) \leq \frac{\exp(|\gamma|\kappa_t)}{|\gamma|} \left(\frac{I(H; S)}{n} + \frac{2^\epsilon |\gamma|^{1+\epsilon} \kappa_t^{1+\epsilon}}{2}\right).$$

**Proposition 3.10.** *Under Assumption 3.8 for some* $\epsilon \in (0, 1]$, *assume further that there exists* $\zeta \in (0, 1)$ *such that one of the following cases holds,*

*(a)* $\zeta \leq \frac{2^\epsilon |\gamma|^{1+\epsilon} \kappa_t^{1+\epsilon} \exp(|\gamma|\kappa_t)}{2}$ *and* $n \geq \frac{2^\epsilon |\gamma|^{1+\epsilon} \kappa_t^{1+\epsilon} I(H;S)}{\zeta^2 \exp(2\gamma\kappa_t)}$,

*(b)* $\zeta > \frac{2^\epsilon |\gamma|^{1+\epsilon} \kappa_t^{1+\epsilon} \exp(|\gamma|\kappa_t)}{2}$ *and* $n \geq \frac{2I(H;S)}{2^\epsilon |\gamma|^{1+\epsilon} \kappa_t^{1+\epsilon}}$,

*(c)* $\zeta > \frac{2^\epsilon |\gamma|^{1+\epsilon} \kappa_t^{1+\epsilon} \exp(|\gamma|\kappa_t)}{2}$ *and*

$$\frac{2I(H; S)}{2^\epsilon |\gamma|^{1+\epsilon} \kappa_t^{1+\epsilon}} > n \geq \frac{2I(H; S)}{2\zeta \exp(\gamma\kappa_t) - 2^\epsilon |\gamma|^{1+\epsilon} \kappa_t^{1+\epsilon}}.$$

*Then, the following lower bounds on non-linear expected generalization error hold,*

$$\widehat{\mathrm{gen}}_\gamma(H, S) \geq \begin{cases} \frac{-\exp(|\gamma|\kappa_t)}{|\gamma|(1-\zeta)} \sqrt{\frac{2|\gamma|^{1+\epsilon} \kappa_t^{1+\epsilon} I(H;S)}{n}}, \\ \quad \textit{if (a) or (b) hold,} \\ \frac{-\exp(|\gamma|\kappa_t)}{(1-\zeta)|\gamma|} \left(\frac{I(H;S)}{n} + \frac{2^\epsilon |\gamma|^{1+\epsilon} \kappa_t^{1+\epsilon}}{2}\right), \\ \quad \textit{if (c) holds.} \end{cases}$$

Using Proposition 3.9 and Proposition 3.10, we can obtain upper bound and lower bounds on the expected tilted generalization error, respectively. Then, combining upper and lower bounds on expected tilted generalization error, we can derive the upper bound on the absolute value of the expected tilted generalization error in the following.

**Theorem 3.11.** *Under the same assumptions as in Proposition 3.10, the following upper bounds on absolute expected tilted generalization error hold,*

$$|\overline{\mathrm{gen}}_\gamma(H, S)| \leq \begin{cases} D(\gamma)\sqrt{\frac{2\kappa_t^{\epsilon+1} |\gamma|^{1+\epsilon} I(H;S)}{n}} + C(\gamma), \\ \quad \textit{if (a) or (b) hold,} \\ D(\gamma)\left(\frac{I(H;S)}{n} + \frac{2^\epsilon |\gamma|^{1+\epsilon} \kappa_t^{1+\epsilon}}{2}\right) + C(\gamma), \\ \quad \textit{if (c) holds.} \end{cases}$$

*where* $C(\gamma) = |\gamma|^\epsilon \kappa_t^{1+\epsilon}$ *and* $D(\gamma) = \frac{\exp(|\gamma|\kappa_t)}{|\gamma|(1-\zeta)}$.

**Convergence rate:** Assuming $\gamma = O(n^{-1/(1+\epsilon)})$ and $n \geq \frac{\kappa_t^{1+\epsilon} \exp(2\kappa_t) I(H;S)}{\zeta^2}$, then the upper bound in Theorem 3.11 has the convergence rate $O(n^{-\epsilon/(1+\epsilon)})$. Note that the result in Theorem 3.11 holds for unbounded loss functions, provided that the $(1 + \epsilon)$-th moment of the loss function exists for some $\epsilon \in (0, 1]$.

The results in this section are non-vacuous for bounded $I(H; S)$. If this assumption is violated, we can apply the individual sample method (Bu et al., 2020), chaining methods (Asadi et al., 2018), or conditional mutual information frameworks (Steinke & Zakynthinou, 2020) to derive a tighter upper bound for the expected tilted generalization error.

## 4. Robustness of TER

Previous experimental work by Li et al. (2021; 2023a) has demonstrated that tilted empirical risk with negative tilt

($\gamma < 0$) exhibits robustness against noisy samples and outliers during training. In this section, we investigate the robustness of TER under distributional shift scenarios. Inspired by the concept of influence functions (Marceau & Rioux, 2001; Christmann & Steinwart, 2004; Ronchetti & Huber, 2009), we model distributional shift through a distribution $\tilde{\mu} \in \mathcal{P}(\mathcal{Z})$, which represents the effect of outliers or noise in the noisy training dataset $\hat{S}$.

**Uniform Approach:** Our robustness analysis of TER is based on the following assumption.

**Assumption 4.1** (Uniform bounded second moment under $\tilde{\mu}$). There is a constant $\kappa_s \in \mathbb{R}^+$ such that the loss function $(H, Z) \mapsto \ell(H, Z)$ satisfies $\mathbb{E}_{\tilde{\mu}}[\ell^{1+\epsilon}(h, Z)] \leq \kappa_s^{1+\epsilon}$ uniformly for all $h \in \mathcal{H}$.

For our robustness analysis, we employ the following decomposition of the tilted generalization error,

$$\text{gen}_\gamma(h, \hat{S}) = \underbrace{\text{R}(h, \mu) - \text{R}_\gamma(h, \mu)}_{I_1} + \widehat{\text{gen}}_\gamma(h, \hat{S}) \\ + \underbrace{\text{R}_\gamma(h, \mu) - \text{R}_\gamma(h, \tilde{\mu})}_{I_4}. \quad (18)$$

Using the functional derivative (Cardaliaguet et al., 2019), see also Appendix C, we can bound $I_4$ as follows.

**Proposition 4.2.** *Given Assumption 4.1 and Assumption 3.1, then the difference of tilted population risk under,* (9)*, between $\mu$ and $\tilde{\mu}$ is bounded as follows for all $h \in \mathcal{H}$,*

$$\frac{1}{|\gamma|}\left| \log(\mathbb{E}_{Z \sim \mu}[C(h, Z)]) - \log(\mathbb{E}_{\tilde{Z} \sim \tilde{\mu}}[C(h, \tilde{Z})]) \right| \quad (19)$$

$$\leq \frac{\mathbb{TV}(\mu, \tilde{\mu})}{\gamma^2} \frac{\left( \exp(|\gamma|\kappa_u) - \exp(|\gamma|\kappa_s) \right)}{(\kappa_u - \kappa_s)},$$

*where $C(h, Z) = \exp(\gamma \ell(h, Z))$.*

Using Proposition 4.2, we can provide upper and lower bounds on the tilted generalization error under distributional shift. Then, combining upper and lower bounds, we can derive an upper bound on the absolute value of the tilted generalization error under distribution shift.

**Theorem 4.3.** *Under the same assumptions in Theorem 3.5 for some $\epsilon \in (0, 1]$, then for $n \geq \frac{(4|\gamma|^{1+\epsilon}\kappa_u^{1+\epsilon} + 8/3\zeta) \log(2/\delta)}{\zeta^2 \exp(2\gamma\kappa_u)}$ and $\gamma < 0$, and with probability at least $(1-\delta)$, the absolute value of the tilted generalization error under distributional shift satisfies*

$$\sup_{h \in \mathcal{H}} |\text{gen}_\gamma(h, \hat{S})| \quad (20)$$

$$\leq \frac{2\exp(|\gamma|\kappa_s)}{(1-\zeta)|\gamma|} \sqrt{\frac{2^\epsilon |\gamma|^{1+\epsilon} \kappa_s^{1+\epsilon} B(\delta)}{n}} + \frac{4\exp(|\gamma|\kappa_s)B(\delta)}{3n|\gamma|(1-\zeta)}$$

$$+ |\gamma|^\epsilon \kappa_u^{1+\epsilon} + \frac{\mathbb{TV}(\mu, \tilde{\mu})}{\gamma^2} \frac{\left( \exp(|\gamma|\kappa_u) - \exp(|\gamma|\kappa_s) \right)}{(\kappa_u - \kappa_s)},$$

*where $B(\delta) = \log(\text{card}(\mathcal{H})) + \log(2/\delta)$.*

Using Lemma 3.7, we can derive an upper bound on excess risk under distribution shift.

**Other Divergences:** Our approach can be extended to incorporate other divergence measures that quantify distribution shift based on training and testing samples directly, rather than their underlying distributions. This is in line with methods commonly explored in the domain adaptation literature (e.g., (Ben-David et al., 2006; Zou et al., 2024; Ye et al., 2021)).

**Comparison with ERM:** Theorem 4.3 gives an upper bound on the tilted generalization error of TERM under distribution shifts, which is valid for all distribution shifts. Specifically, this upper bound depends on the total variation distance $\mathbb{TV}(\mu, \tilde{\mu})$, which is bounded for any distributions $\mu$ and $\tilde{\mu}$. This robustness is attributed to the negative tilt and properties of the exponential function. In contrast, for ERM, we need to derive an upper bound on the following term,

$$\text{Dif}(\mu, \tilde{\mu}) := \mathbb{E}_{Z \sim \mu}[\ell(h, Z)] - \mathbb{E}_{\tilde{Z} \sim \tilde{\mu}}[\ell(h, \tilde{Z})]. \quad (21)$$

For unbounded loss functions, deriving an upper bound on $\text{Dif}(\mu, \tilde{\mu})$ in terms of total variation distance is not feasible. While such bounds can be established using KL-divergence $\text{KL}(\mu\|\tilde{\mu})$ under specific conditions (e.g., when the loss function is sub-Gaussian under the data-generating distribution), these bounds may become unbounded for certain $\tilde{\mu}$. In this context, TERM with negative tilt emerges as a robust solution, providing theoretical performance guarantees for distribution-shift under unbounded loss functions.

**Robustness vs Generalization:** The term $\frac{\mathbb{TV}(\mu, \tilde{\mu})}{\gamma^2}$ represents the distributional shift cost (or robustness) associated with the TER. This cost can be reduced by increasing $|\gamma|$. However, increasing $|\gamma|$ also amplifies other terms in the upper bound of the tilted generalization error. Therefore, there is a trade-off between robustness and generalization, particularly for $\gamma < 0$ in the TER. Li et al. (2021) also empirically observed this trade-off for negative tilt.

**Information-theoretic Approach:** Using a similar approach, we can derive an upper bound on expected tilted generalization error under distribution shift. We make the following assumption.

**Assumption 4.4.** The learning algorithm $P_{H|\hat{S}}$, loss function $\ell$, and $\mu$ are such that there is a constant $\kappa_t \in \mathbb{R}^+$ with which the loss function $(H, Z) \mapsto \ell(H, Z)$ satisfies $\max\left(\mathbb{E}_{P_{H, \tilde{Z}}}[\ell^{1+\epsilon}(H, \tilde{Z})], \mathbb{E}_{P_H \otimes \tilde{\mu}}[\ell^{1+\epsilon}(H, \tilde{Z})]\right) \leq \kappa_{st}^{1+\epsilon}$ for some $\epsilon \in (0, 1]$.

Using Theorem 3.11 and similar results to Proposition 4.2, we derive an upper bound on absolute expected tilted generalization error under distribution shift.

**Theorem 4.5.** *Under the same assumptions as in Theorem 3.11 and Assumption 4.4, the following upper bounds on absolute expected tilted generalization error hold under distributional shift, if (a) or (b) hold,*

$$|\overline{\text{gen}}_\gamma(H, \hat{S})| \leq \frac{\exp(|\gamma|\kappa_{st})}{|\gamma|(1-\zeta)} G(I(H; \hat{S})) + |\gamma|^\epsilon \kappa$$

$$+ \frac{\mathbb{TV}(\mu, \tilde{\mu})}{\gamma^2} \frac{\left(\exp(|\gamma|\kappa_t) - \exp(|\gamma|\kappa_{st})\right)}{(\kappa_t - \kappa_{st})},$$

*where $\kappa = \kappa_t^{1+\epsilon} + \kappa_{st}^{1+\epsilon}$, if (a) or (b) hold we have $G(I(H; \hat{S})) = \sqrt{\frac{2\kappa_{st}^{\epsilon+1}|\gamma|^{1+\epsilon}I(H;\hat{S})}{n}}$ and if (c) holds, we have, $G(I(H; \hat{S})) = \frac{I(H;\hat{S})}{n} + \frac{2^\epsilon |\gamma|^{1+\epsilon}\kappa_{st}^{1+\epsilon}}{2}$.*

We can observe the same robustness and generalization in Theorem 4.5.

## 5. Data-driven Choice of Tilt

In this section, we provide a data-driven approach for choosing the tilt ($\gamma$) based on theoretical results. It is noteworthy that the upper bound in Theorem 4.3 can be infinite for $\gamma \to -\infty$ and $\gamma = 0$ and a fixed $n$. Consequently, there must exist a $\gamma \in (-\infty, 0)$ that minimizes this upper bound. To illustrate this point, consider the case where $n \to \infty$; here, the first and second terms in the upper bound, (20), would vanish. Thus, we are led to the following minimization problem:

$$\gamma_{\text{data}} := \underset{\gamma \in (-\infty, 0)}{\arg\min} \left[ |\gamma|^\epsilon \kappa_u^{1+\epsilon} \right.$$
$$\left. + \frac{\mathbb{TV}(\mu, \tilde{\mu})}{\gamma^2} \frac{\left(\exp(|\gamma|\kappa_u) - \exp(|\gamma|\kappa_s)\right)}{(\kappa_u - \kappa_s)} \right], \quad (22)$$

for which a solution $\gamma^*$ exists. As $\gamma^*$ decreases when $\mathbb{TV}(\mu, \tilde{\mu})$ increases, practically, this implies that if the training distribution becomes more adversarial (i.e., further away from the benign test distribution), we would use smaller negative $\gamma$'s to bypass outliers. Therefore, we can consider $\gamma_{\text{data}}$ as a data-driven choice for the TERM problem. We consider a simple experiment to show the effectiveness of data-driven tilt, $\gamma_{\text{data}}$.

In this experiment, we consider the logistic regression setup described in Li et al. (2023a), adding a Gaussian or Pareto outlier dataset to the training dataset at different ratios ($\rho$) relative to the training dataset. We evaluate the following values:

- $\gamma^*$: The optimal tilt based on grid search
- $R(h_{\gamma^*}(\hat{S}), \mu)$: The population risk under the optimal TERM solution where $h_{\gamma^*}(\hat{S}) := \arg\min_{h \in \mathcal{H}} \hat{R}_{\gamma^*}(h, \hat{S})$.
- $R(h_{\text{ERM}}(\hat{S}), \mu)$: The population risk under the ERM solution where $h_{\text{ERM}}(\hat{S}) := \arg\min_{h \in \mathcal{H}} \hat{R}(h, \hat{S})$.

- $\gamma_{\text{data}}$: The data-driven tilt based on optimization of (22).
- $R(h_{\gamma_{\text{data}}}(\hat{S}), \mu)$: The population risk under the data-driven tilt solution where $h_{\gamma_{\text{data}}}(\hat{S}) := \arg\min_{h \in \mathcal{H}} \hat{R}_{\gamma_{\text{data}}}(h, \hat{S})$.

The training dataset consists of 1,000 samples. The results for Gaussian and Pareto outliers are reported in Table 1 and Table 2, respectively. For the Pareto-outlier as a distribution with unbounded second moment, we observe better performance for the data-driven approach in comparison with ERM. Note that the variance of TERM and data-driven TERM in comparison with ERM has less variance as expected. The details of the experiments are provided in Appendix G. More experiments for Linear regression as proposed in (Li et al., 2021) is provided in Appendix G.

Table 1: Logistic Regression with Gaussian Outliers: Results averaged over three runs ($n = 1,000$ samples), showing mean $\pm$ standard deviation.

| $\rho$ | $\gamma^\star$ | $R(h_{\gamma^\star}(\hat{S}), \mu)$ | $R(h_{\text{ERM}}(\hat{S}), \mu)$ | $\gamma_{\text{data}}$ | $R(h_{\gamma_{\text{data}}}(\hat{S}), \mu)$ |
|---|---|---|---|---|---|
| 0.1% | $-0.53_{\pm 0.000}$ | $0.00_{\pm 0.000}$ | $0.05_{\pm 0.001}$ | $-1.40_{\pm 0.000}$ | $0.00_{\pm 0.000}$ |
| 17.6% | $-2.98_{\pm 0.000}$ | $0.15_{\pm 0.004}$ | $0.22_{\pm 0.001}$ | $-4.91_{\pm 0.002}$ | $0.16_{\pm 0.003}$ |
| 35.0% | $-3.86_{\pm 0.000}$ | $0.16_{\pm 0.004}$ | $0.30_{\pm 0.002}$ | $-3.33_{\pm 0.000}$ | $0.20_{\pm 0.002}$ |
| 52.5% | $-2.10_{\pm 0.000}$ | $0.11_{\pm 0.001}$ | $0.28_{\pm 0.002}$ | $-1.93_{\pm 0.000}$ | $0.14_{\pm 0.001}$ |
| 70.0% | $-1.23_{\pm 0.000}$ | $0.14_{\pm 0.002}$ | $0.18_{\pm 0.000}$ | $-2.28_{\pm 0.000}$ | $0.15_{\pm 0.002}$ |

Table 2: Logistic Regression with Pareto Outliers: Results averaged over three runs ($n = 1,000$ samples), showing mean $\pm$ standard deviation.

| $\rho$ | $\gamma^\star$ | $R(h_{\gamma^\star}(\hat{S}), \mu)$ | $R(h_{\text{ERM}}(\hat{S}), \mu)$ | $\gamma_{\text{data}}$ | $R(h_{\gamma_{\text{data}}}(\hat{S}), \mu)$ |
|---|---|---|---|---|---|
| 0.1% | $-1.40_{\pm 0.000}$ | $0.00_{\pm 0.000}$ | $0.03_{\pm 0.001}$ | $-0.70_{\pm 0.000}$ | $0.01_{\pm 0.000}$ |
| 17.58% | $-3.33_{\pm 0.000}$ | $0.00_{\pm 0.003}$ | $0.02_{\pm 0.000}$ | $-0.88_{\pm 0.000}$ | $0.01_{\pm 0.000}$ |
| 35.05% | $-1.05_{\pm 0.000}$ | $0.00_{\pm 0.000}$ | $0.01_{\pm 0.002}$ | $-0.70_{\pm 0.000}$ | $0.01_{\pm 0.000}$ |
| 52.53% | $-1.05_{\pm 0.000}$ | $0.01_{\pm 0.000}$ | $0.01_{\pm 0.002}$ | $-1.06_{\pm 0.000}$ | $0.01_{\pm 0.001}$ |
| 70.00% | $-0.88_{\pm 0.000}$ | $0.00_{\pm 0.002}$ | $0.02_{\pm 0.001}$ | $-0.70_{\pm 0.000}$ | $0.01_{\pm 0.000}$ |

## 6. The KL-Regularized TERM Problem for Unbounded Loss Functions

Our upper bound in Theorem 3.11 on the absolute value of the expected tilted generalization error depends on the mutual information between $H$ and $S$. Therefore, it is of interest to investigate an algorithm that minimizes the regularized expected TERM via mutual information.

$$P_{H|S}^\star = \underset{P_{H|S}}{\arg\inf} \overline{R}_\gamma(H, P_{H,S}) + \frac{1}{\alpha} I(H; S), \quad (23)$$

where $\alpha$ is the inverse temperature. As discussed by Xu & Raginsky (2017); Aminian et al. (2023), the regularization problem in (23) is dependent on the data distribution, $P_S$. Therefore, we relax the problem in (23) by considering the following regularized version via KL divergence,

$$P_{H|S}^\star = \underset{P_{H|S}}{\arg\inf} \overline{R}_\gamma(H, P_{H,S}) \quad (24)$$
$$+ \frac{1}{\alpha} \text{KL}(P_{H|S} \| \pi_H | P_S),$$

where $I(H; S) \leq \text{KL}(P_{H|S} \| \pi_H | P_S)$ and $\pi_H$ is a prior distribution over hypothesis space $\mathcal{H}$. All proof details are deferred to Appendix F.

**Proposition 6.1.** *The solution to the expected TERM regularized via KL divergence, (24), is the tilted Gibbs Posterior (a.k.a. Gibbs Algorithm),*

$$P_{H|S}^{\gamma} := \frac{\pi_H}{F_\alpha(S)} \Big( \frac{1}{n} \sum_{i=1}^{n} \exp(\gamma \ell(H, Z_i)) \Big)^{-\alpha/\gamma}, \quad (25)$$

*where $F_\alpha(S)$ is a normalization factor.*

Note that the Gibbs posterior,

$$P_{H|S}^{\alpha} := \frac{\pi_H}{\tilde{F}_\alpha(S)} \exp\Big( -\alpha \Big( \frac{1}{n} \sum_{i=1}^{n} \ell(H, Z_i) \Big) \Big), \quad (26)$$

is the solution to the KL-regularized ERM minimization problem, where $\tilde{F}_\alpha(S)$ is the normalization factor. Therefore, the tilted Gibbs posterior is different from the Gibbs posterior, (26). It can be shown that for $\gamma \to 0$, the tilted Gibbs posterior converges to the Gibbs posterior. Therefore, it is interesting to study the expected tilted generalization error of the tilted Gibbs posterior. For this purpose, we give an exact characterization of the difference between the expected TER under the joint and the product of marginal distributions of $H$ and $S$.

**Proposition 6.2.** *The difference between the expected TER under the joint and product of marginal distributions of $H$ and $S$ can be expressed as,*

$$\overline{\text{R}}_\gamma(H, P_H \otimes \mu) - \overline{\text{R}}_\gamma(H, P_{H,S}) = \frac{I_{\text{SKL}}(H; S)}{\alpha}. \quad (27)$$

We next provide a parametric upper bound on the tilted generalization error of the tilted Gibbs posterior.

**Theorem 6.3.** *Under the same Assumptions, cases (a) and (b) in Theorem 3.11, the expected tilted generalization error of the tilted Gibbs posterior satisfies*

$$0 \leq \overline{\text{gen}}_\gamma(H, S) \leq \frac{2\alpha \exp(2|\gamma|\kappa_t)\kappa_t^{1+\epsilon}}{(1-\zeta)^2 n |\gamma|^{1-\epsilon}} + \frac{\exp(|\gamma|\kappa_t)\kappa_t^{1+\epsilon}|\gamma|^{1/2+\epsilon}}{(1-\zeta)|\gamma|} \sqrt{\frac{2\alpha}{n}} + 2|\gamma|^\epsilon \kappa_t^{1+\epsilon}. \quad (28)$$

**Convergence rate:** If $\gamma = O(1/n)$, then we obtain a convergence rate of $O(n^{-\epsilon})$ for the upper bound on the tilted generalization error of the tilted Gibbs posterior.

**Comparison with the Gibbs posterior:** Our results offer several advantages over the prior work in Aminian et al. (2021a). While Theorem 3 in their work establishes an $O(1/n)$ upper bound on the expected tilted generalization

error of the Gibbs posterior, it requires the strong assumption of sub-Gaussian loss functions. In contrast, we achieve the same $O(1/n)$ convergence rate under the weaker condition of bounded second moments (when $\epsilon = 1$) for the tilted Gibbs posterior. Additionally, our analysis extends to loss functions with bounded $(1 + \epsilon)$-th moments for some $\epsilon \in (0, 1]$, a scenario not addressed in the Gibbs posterior framework.

## 7. Related Works

In this section, we discuss related works on tilted empirical risk minimization and generalization error under an unbounded loss function. More related works for generalization error analysis are discussed in Appendix B.

**Tilted Empirical Risk Minimization:** The TERM algorithm for machine learning is proposed by Li et al. (2021), where good performance of the TERM under outlier and noisy label scenarios for negative tilting ($\gamma < 0$), and, under imbalance and fairness constraints, for positive tilting ($\gamma > 0$), is demonstrated. Zhang et al. (2023) study the TERM as a target function to improve the robustness of estimators. The application of TERM in federated learning is also studied, in Li et al. (2023b); Zhang et al. (2022).

Inspired by TERM, Wang et al. (2023) propose a class of new tilted sparse additive models based on the Tikhonov regularization scheme. Their results have some limitations. First, in (Wang et al., 2023, Theorem 3.3) the authors derive an upper bound for $\lambda = n^{-\zeta}$ where $\zeta < -1/2$ and $\lambda$ are the regularization parameters in (Wang et al., 2023, Eq.4). This implies $\lambda \to \infty$ as $n \to +\infty$, which is impractical. As the analysis in (Wang et al., 2023) assumes that both the loss function and its derivative are bounded, it can not be applied to the unbounded loss function scenario. Furthermore, we consider KL regularization, which is different from the Tikhonov regularization scheme with the sparsity-induced $\ell_{1,2}$-norm regularizer as introduced in Wang et al. (2023). Therefore, our results do not cover the learning algorithm in Wang et al. (2023). Lee et al. (2020) propose an upper bound on the entropic risk function generalization error via a representation of a coherent risk function and using the Rademacher complexity approach. However, their approach is limited to bounded loss functions.

Tilted risk for off-policy learning and evaluation is proposed in a concurrent work by Behnamnia et al. (2025) where a regret bound analysis of this estimator under heavy-tailed weighted reward assumption is proposed. More details about similarities and differences with this work are provided in Appendix B.

Although rich experiments are given by Li et al. (2021) for the TERM algorithm in different applications, the generalization error of the TERM has not yet been addressed for

unbounded loss functions with bounded $(1 + \epsilon)$-th moment for some $\epsilon \in (0, 1]$.

**Generalization error under unbounded loss functions:** Several studies have investigated the generalization error of linear empirical risk under unbounded loss functions. Some works studied the generalization error under unbounded loss functions via the PAC-Bayesian approach. Losses with heavier tails are studied by Alquier & Guedj (2018) where probability bounds are developed. Using a different estimator than empirical risk, PAC-Bayes bounds for losses with bounded second and third moments are developed by Holland (2019). Notably, their bounds include a term that can increase with the number of samples $n$. Kuzborskij & Szepesvári (2019) and Haddouche & Guedj (2022) also provide bounds for losses with a bounded second moment. The bounds in Haddouche & Guedj (2022) rely on a parameter that must be selected before the training data is drawn. Information-theoretic bounds based on the second moment of $\sup_{h \in \mathcal{H}} |\ell(h, Z) - \mathbb{E}[\ell(h, \tilde{Z})]|$ are derived in Lugosi & Neu (2022; 2023). In contrast, our second moment assumption is more relaxed, being based on the expected version with respect to the distribution over the hypothesis set and the data-generating distribution. Furthermore, an upper bound on generalization error via VC-dimension and growth function under bounded $(1 + \epsilon)$-th moment for $\epsilon \in (0, 1]$ is proposed in (Cortes et al., 2019, Corollary 12) which is motivated by relative deviation generalization bounds in binary classification. In addition, the final convergence rate for unbounded loss is $O(\log(n)n^{\frac{-\epsilon}{1+\epsilon}})$ based on (Cortes et al., 2019, Corollary 12). In contrast, we derive the results for a multi-classification scenario with a convergence rate of $O(n^{\frac{-\epsilon}{1+\epsilon}})$. Our work focuses on tilted empirical risk, which is overlooked in the literature on unbounded loss functions.

## 8. Conclusion

In this paper, we study the tilted empirical risk minimization, as proposed by Li et al. (2021). In particular, we established an upper and lower bound on the tilted generalization error of the tilted empirical risk through uniform and information-theoretic approaches, obtaining theoretical guarantees that the convergence rate is $O(n^{-\epsilon/(1+\epsilon)})$ under unbounded loss functions for negative tilt provided that $(1+\epsilon)$-th moment of loss function is bounded for some $\epsilon \in (0, 1]$. We also study the tilted generalization error under distribution shift in the training dataset due to noise or outliers, where we discussed the generalization and robustness trade-off. We also explore the KL-regularized tilted empirical risk minimization, where the solution involves the tilted Gibbs posterior, and we derive a parametric upper bound on this minimization with a convergence rate of $O(n^{-\epsilon})$ under some conditions and unbounded loss function provided that $(1 + \epsilon)$-th moment of loss functions is bounded for some $\epsilon \in (0, 1]$.

## 9. Future Works

Building on our current results, we highlight several promising directions for future work:

**Overparameterization Regime:** An interesting avenue is to explore generalization bounds for tilted empirical risk in overparameterized models, particularly in the mean-field regime. Techniques from recent works, such as Aminian et al. (2023), may offer valuable insights for extending our results in this setting.

**Removing Sample Size Assumptions for Unbounded Losses:** For unbounded loss functions, we currently require a lower bound on the number of training samples to establish generalization guarantees. While we relaxed this assumption for the bounded loss case (see Appendix H), extending the analysis to the unbounded case without any sample size constraints would be a valuable extension.

**Positive Tilt** ($\gamma > 0$)**:** Our analysis focuses on the case of negative tilting ($\gamma < 0$), assuming that the unbounded loss function has a bounded $(1 + \epsilon)$-th moment for some $\epsilon \in (0, 1]$. Extending the generalization analysis to positive tilting scenarios with unbounded losses remains an open and important direction.

**Alternative Generalization Frameworks:** Our current bounds rely on uniform and information-theoretic frameworks. However, other frameworks—such as PAC-Bayesian analysis (Catoni, 2003), algorithmic stability (Bousquet & Elisseeff, 2002b), and Rademacher complexity (Bartlett & Mendelson, 2002)—offer complementary perspectives on generalization. We derived results using these frameworks under the bounded loss function assumption in Appendix H. Investigating the applicability of these methods to tilted empirical risk under an unbounded loss function could yield deeper theoretical insights.

## Acknowledgment

Gholamali Aminian, Gesine Reinert and Samuel N. Cohen acknowledge the support of the UKRI Prosperity Partnership Scheme (FAIR) under EPSRC Grant EP/V056883/1 and the Alan Turing Institute. Gesine Reinert is also supported in part by EPSRC grants EP/W037211/1 and EP/R018472/1. Amir R. Asadi is supported by Leverhulme Trust grant ECF-2023-189 and Isaac Newton Trust grant 23.08(b). The authors thank the anonymous referees and the area chair for their insightful and constructive comments, which have led to an improved version of the paper. For the purpose of Open Access, the authors note that a CC BY public copyright license applies to any Author Accepted Manuscript version arising from this submission.

## Impact Statement

The goal of this work is to improve our understanding of tilted empirical risk (TER) and to enhance its practical applicability by introducing a data-driven approach for hyperparameter tuning. However, we acknowledge that our current theoretical analysis, which focuses on the negative tilt regime for understanding the robustness of TER, does not capture all aspects of TER—particularly its behavior under positive tilt for biases and imbalance scenarios.

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

# Appendix

## Table of Contents

## A. Summary of Notations

All notations are summarized in Table 3.

Table 3: Summary of notations in the paper

| Notation | Definition | Notation | Definition |
|---|---|---|---|
| $S$ | Training set | $Z_i$ | $i$-th sample |
| $\mathcal{X}$ | Input (feature) space | $\mathcal{Y}$ | Output (label) space |
| $\mu$ | Data-generating distribution | $\tilde{\mu}$ | Data-generating distribution under distributional shift |
| $\gamma$ | Tilt parameter | $\mathrm{KL}(P\|Q)$ | KL-divergence between $P$ and $Q$ |
| $I(X;Y)$ | Mutual information | $D_{\mathrm{SKL}}(P\|Q)$ | Symmetrized KL divergence |
| $\mathbb{TV}(P,Q)$ | Total variation distance | $\kappa_u$ | Bound on second moment |
| $h$ | Hypothesis | $\mathcal{H}$ | Hypothesis space |
| $\ell(h,z)$ | Loss function | $R(h,\mu)$ | Population (true) risk |
| $\widehat{\mathrm{R}}_\gamma(h,S)$ | Tilted empirical risk | $\mathrm{gen}_\gamma(h,S)$ | Tilted generalization error |
| $\overline{\mathrm{R}}_\gamma(h,\mu^{\otimes n})$ | Tilted population risk | $\mathfrak{E}_\gamma(\mu)$ | Excess risk |

## B. Other Related Works

This section details related works about tilted empirical risk minimization, the Rademacher complexity and stability and PAC-Bayesian bounds. We also compare our work with Behnamnia et al. (2025) in more details.

**Generalization Error Analysis:** Different approaches have been applied to study the generalization error of general learning problems under empirical risk minimization, including VC dimension-based, Rademacher complexity, PAC-Bayesian, stability and information-theoretic bounds.

*Uniform Bounds:* Uniform bounds (or VC bounds) are proposed by Vapnik & Chervonenkis (1971); Bartlett et al. (1998; 2019). For any class of functions $\mathcal{F}$ of VC dimension $d$, with probability at least $1 - \delta$ the generalization error is $O\big((d + \log(1/\delta))^{1/2}n^{-1/2}\big)$. This bound depends solely on the VC dimension of the function class and on the sample size; in particular, it is independent of the learning algorithm.

*Information-theoretic bounds:* Russo & Zou (2019); Xu & Raginsky (2017) propose using the mutual information between the input training set and the output hypothesis to upper bound the expected generalization error. Multiple approaches have been proposed to tighten the mutual information-based bound: Bu et al. (2020) provide tighter bounds by considering the individual sample mutual information; Asadi et al. (2018); Asadi & Abbe (2020) propose using chaining mutual information; and Steinke & Zakynthinou (2020); Hafez-Kolahi et al. (2020); Aminian et al. (2020; 2021b) provide different upper bounds on the expected generalization error based on the linear empirical risk framework.

*Rademacher Complexity Bounds:* This approach is a data-dependent method to provide an upper bound on the generalization error based on the Rademacher complexity of the function class $\mathcal{H}$, see (Bartlett & Mendelson, 2002; Golowich et al., 2018). Bounding the Rademacher complexity involves the model parameters. Typically, in Rademacher complexity analysis, a symmetrization technique is used which can be applied to the empirical risk, but not directly to the TER.

*Stability Bounds:* Stability-based bounds for generalization error are given in Bousquet & Elisseeff (2002a); Bousquet et al. (2020); Mou et al. (2017); Chen et al. (2018); Aminian et al. (2023). For stability analysis, the key tool is Lemma 7 in Bousquet & Elisseeff (2002b), which is based on ERM linearity. Therefore, we can not apply stability analysis to TER directly.

*PAC-Bayesian bounds:* First proposed by Shawe-Taylor & Williamson (1997); McAllester (1999) and McAllester (2003), PAC-Bayesian analysis provides high probability bounds on the generalization error in terms of the KL divergence between the data-dependent posterior induced by the learning algorithm and a data-free prior that can be chosen arbitrarily (Alquier, 2021). There are multiple ways to generalize the standard PAC-Bayesian bounds, including using information measures other than KL divergence (Alquier & Guedj, 2018; Bégin et al., 2016; Hellström & Durisi, 2020; Aminian et al., 2021b) and considering data-dependent priors (Rivasplata et al., 2020; Catoni, 2007; Dziugaite & Roy, 2018; Ambroladze et al., 2007). However, this method has not been applied to TER to provide generalization error bounds.

The aforementioned approaches are applied to study the generalization error in the linear empirical risk framework. To our knowledge, the generalization error of the tilted empirical risk minimization from information-theoretical and uniform approach perspectives has not been explored.

**Comparison with Behnamnia et al. (2025):** In a concurrent work in reinforcement learning, Behnamnia et al. (2025) introduce an estimator for off-policy evaluation and learning based on the log-sum-exponential (LSE) operator applied to weighted rewards. This estimator can be viewed as a negatively tilted risk function. Their focus lies in bounding the regret, defined as the gap between the value of the optimal policy and that induced by the estimator, and in deriving concentration inequalities under heavy-tailed reward distributions. We compare our work with theirs from several perspectives:

*Objective Functions:* Our study targets supervised learning, where the goal is to minimize a risk function such as empirical or tilted risk. In contrast, Behnamnia et al. (2025) addresses off-policy learning, aiming to maximize expected reward for a given logged data with respect to a policy, conditional distribution over actions for a given context. Furthermore, in a supervised learning scenario, the loss function is given. In contrast, in off-policy learning, the reward function is not known, and some samples of reward are given. Therefore, the learning paradigms and optimization targets differ. For more details, comparison of these scenarios see (Swaminathan & Joachims, 2015, Table 1).

*Assumptions:* We assume a bounded hypothesis class and heavy-tailed loss functions. In contrast, (Behnamnia et al., 2025, Assumption 5.1) considers a bounded policy class and assume heavy-tailed weighted rewards. These differences reflect the distinct nature of these tasks: off-policy and supervised learning scenarios. Note that the definition of heavy-tailed random variable is general in statistical learning theory (Resnick, 2007) and can be applied in different contexts, e.g., bandit, off-policy learning, and supervised learning.

*Methodology:* Despite differing goals, both works employ some common statistical tools, such as concentration inequalities. We build on analytical techniques similar to theirs, including lemmas concerning variance bounds under heavy-tailed distributions. However, our uniform bounds operate under different assumptions and are adapted to the supervised learning setting. In particular, we used Lemma C.2 and Lemma C.3– where a different proof of Lemma C.3 is proposed in (Behnamnia et al., 2025, Theorem 5.3)– to derive our upper bounds on absolute value of generalization error (Theorem 3.5) using uniform approach under heavy-tailed assumption. However, as discussed in Section 3.1, the uniform approach has inherent limitations, e.g., assuming bounded hypothesis sets which is a complexity measure in the supervised learning scenario. The limitations also apply to results in (Behnamnia et al., 2025, Theorem 5.3) with finite policy set as a complexity measure in off-policy learning. To overcome the limitations of finiteness of sets assumptions, we further develop new bounds using information-theoretic techniques.

*Bounded Scenario:* Both works note limitations on sample complexity under heavy-tailed assumptions. We show in Appendix H that such constraints can be lifted by assuming bounded loss functions—leading to generalization guarantees for all number of training samples.

*Robustness:* A key contribution of our work is the analysis of tilted empirical risk under distribution shift, a direction not explored in earlier literature to our knowledge. Our results highlight the distinct value of TER in handling training-test distribution discrepancies.

*Regularization:* We also explored the KL-regularized problem for tilted empirical risk, where a tighter convergence rate of $O(n^{-\epsilon})$ for tilted empirical risk was derived under a heavy-tailed assumption.

## C. Technical Tools

We first define the functional linear derivative as in (Cardaliaguet et al., 2019).

**Definition C.1.** (Cardaliaguet et al., 2019) A functional $U : \mathcal{P}(\mathbb{R}^n) \to \mathbb{R}$ admits a *functional linear derivative* if there is a map $\frac{\delta U}{\delta m} : \mathcal{P}(\mathbb{R}^n) \times \mathbb{R}^n \to \mathbb{R}$ which is continuous on $\mathcal{P}(\mathbb{R}^n)$, such that for all $m, m' \in \mathcal{P}(\mathbb{R}^n)$, it holds that

$$U(m') - U(m) = \int_0^1 \int_{\mathbb{R}^n} \frac{\delta U}{\delta m}(m_\lambda, a)\,(m' - m)(da)\,\mathrm{d}\lambda,$$

where $m_\lambda = m + \lambda(m' - m)$.

The following lemmas are used in our proofs.

**Lemma C.2** (Lemma 28 in (Lugosi & Neu, 2023)). *For $x > 0$, the following inequality holds for $\epsilon \in (0, 1]$,*

$$\exp(-x) \leq 1 - x + x^{(1+\epsilon)}. \tag{29}$$

The following lemma also appears in the concurrent work by Behnamnia et al. (2025). For the sake of completeness, we provide an alternative proof below.

**Lemma C.3.** *Suppose that $X > 0$ and $\gamma < 0$, then we have for $\epsilon \in (0, 1]$,*

$$\mathrm{Var}(\exp(\gamma X)) \leq 2^{\epsilon} |\gamma|^{1+\epsilon} \mathbb{E}[|X|^{1+\epsilon}]. \tag{30}$$

*Proof.* From the variance definition, we have,

$$\mathrm{Var}(\exp(\gamma X)) = \mathbb{E}\left[\left(\exp(\gamma X) - \mathbb{E}[\exp(\gamma X)]\right)^2\right] \tag{31}$$

$$\overset{(a)}{=} \frac{1}{2} \mathbb{E}_{X,X'}\left[\left(\exp(\gamma X) - \exp(\gamma X')\right)^2\right] \tag{32}$$

$$\overset{(b)}{\leq} \frac{1}{2} \mathbb{E}_{X,X'}\left[\left(\exp(\gamma X) - \exp(\gamma X')\right)^{1+\epsilon}\right] \tag{33}$$

$$\overset{(c)}{\leq} \frac{1}{2} |\gamma|^{1+\epsilon} \mathbb{E}_{X,X'}[|X - X'|^{1+\epsilon}] \tag{34}$$

$$\overset{(d)}{\leq} \frac{1}{2} |\gamma|^{1+\epsilon} \mathbb{E}_{X,X'}[|X + X'|^{1+\epsilon}] \tag{35}$$

$$\overset{(e)}{\leq} 2^{\epsilon} |\gamma|^{1+\epsilon} \mathbb{E}_X[|X|^{1+\epsilon}], \tag{36}$$

where (a) follows from variance representation for i.i.d copy $X$ and $X'$, (b) follows from the fact that $|\exp(\gamma X) - \exp(\gamma X')| \leq 1$, (c) follows from the fact that $\exp(\gamma x)$ is Lipschitz for $\gamma < 0$, with parameter $|\gamma|$, therefore, we have $|\exp(\gamma x) - \exp(\gamma x')| \leq |\gamma||x - x'|$, (d) follows the positiveness of $X$ and $X'$ and (e) follows from Jensen-inequality. $\square$

**Lemma C.4.** *Suppose that $0 < a < X < b < \infty$. Then the following inequality holds,*

$$\frac{\mathrm{Var}_{P_X}(X)}{2b^2} \leq \log(\mathbb{E}[X]) - \mathbb{E}[\log(X)] \leq \frac{\mathrm{Var}_{P_X}(X)}{2a^2},$$

*where $\mathrm{Var}_{P_X}(X)$ is the variance of $X$ under the distribution $P_X$.*

*Proof.* As $\frac{\mathrm{d}^2}{\mathrm{d}x^2}\left(\log(x) + \beta x^2\right) = \frac{-1}{x^2} + 2\beta$, the function $\log(x) + \beta x^2$ is concave for $\beta = \frac{1}{2b^2}$ and convex for $\beta = \frac{1}{2a^2}$. Hence, by Jensen's inequality,

$$\mathbb{E}[\log(X)] = \mathbb{E}\left[\log(X) + \frac{X^2}{2b^2} - \frac{X^2}{2b^2}\right]$$

$$\leq \log(\mathbb{E}[X]) + \frac{1}{2b^2}\mathbb{E}[X]^2 - \frac{1}{2b^2}\mathbb{E}[X^2]$$

$$= \log(\mathbb{E}[X]) - \frac{1}{2b^2}\mathrm{Var}_{P_X}(X),$$

which completes the proof of the lower bound. A similar approach can be applied to derive the upper bound. $\square$

**Lemma C.5.** *Suppose that $0 < a < X < b < \infty$. Then the following inequality holds,*

$$\frac{-\mathrm{Var}_{P_X}(X)\exp(b)}{2} \leq \exp(E[X]) - E[\exp(X)] \leq \frac{-\mathrm{Var}_{P_X}(X)\exp(a)}{2},$$

*where $\mathrm{Var}_{P_X}(X)$ is the variance of $X$ under the distribution $P_X$.*

In the next results, $P_S$ is the distribution of $S$.

**Lemma C.6** (Hoeffding Inequality, Boucheron et al., 2013)**.** *Suppose that $S = \{Z_i\}_{i=1}^n$ are bounded independent random variables such that $a \leq Z_i \leq b, i = 1, \ldots, n$. Then the following inequality holds with probability at least $(1 - \delta)$ under $P_S$,*

$$\left| \mathbb{E}[Z] - \frac{1}{n} \sum_{i=1}^n Z_i \right| \leq (b - a) \sqrt{\frac{\log(2/\delta)}{2n}}. \tag{37}$$

**Lemma C.7** (Bernstein's Inequality, Boucheron et al., 2013)**.** *Suppose that $S = \{Z_i\}_{i=1}^n$ are i.i.d. random variable such that $|Z_i - \mathbb{E}[Z]| \leq R$ almost surely for all $i$, and $\mathrm{Var}(Z) = \sigma^2$. Then the following inequality holds with probability at least $(1 - \delta)$ under $P_S$,*

$$\left| \mathbb{E}[Z] - \frac{1}{n} \sum_{i=1}^n Z_i \right| \leq \sqrt{\frac{4\sigma^2 \log(2/\delta)}{n}} + \frac{4R \log(2/\delta)}{3n}. \tag{38}$$

**Lemma C.8.** *For a positive random variable, $Z > 0$ and $\epsilon \in (0, 1]$, suppose $\mathbb{E}[Z^{1+\epsilon}] \leq \eta$. Then, the following inequality holds,*

$$\mathbb{E}[Z] \leq \eta^{1/1+\epsilon}.$$

**Lemma C.9.** *Suppose $\mathbb{E}[|X|^{1+\epsilon}] < \infty$. Then, for $X > 0$ and $\gamma < 0$ the following inequality holds,*

$$0 \leq \mathbb{E}[X] - \frac{1}{\gamma} \log \mathbb{E}[e^{\gamma X}] \leq |\gamma|^\epsilon \mathbb{E}[X^{1+\epsilon}].$$

*Proof.* The left inequality follows from Jensen's inequality applied to $f(x) = \log(x)$. For the right inequality, from Lemma C.2 for $\epsilon \in (0, 1]$

$$e^{\gamma X} \leq 1 + \gamma X + |\gamma|^{1+\epsilon} X^{1+\epsilon}. \tag{39}$$

Therefore, we have,

$$\begin{aligned} \frac{1}{\gamma} \log \mathbb{E}[e^{\gamma X}] &\geq \frac{1}{\gamma} \log \mathbb{E}\left[1 + \gamma X + |\gamma|^{1+\epsilon} |X|^{1+\epsilon}\right] \\ &= \frac{1}{\gamma} \log \left(1 + \gamma \mathbb{E}[X] + |\gamma|^{1+\epsilon} \mathbb{E}[|X|^{1+\epsilon}]\right) \\ &\geq \frac{1}{\gamma} \left(\gamma \mathbb{E}[X] + |\gamma|^{1+\epsilon} \mathbb{E}[|X|^{1+\epsilon}]\right) \\ &= \mathbb{E}[X] - |\gamma|^\epsilon \mathbb{E}[X^{1+\epsilon}]. \end{aligned}$$

$\square$

**Lemma C.10.** *Suppose that $0 < a < x < b$ and $f(x)$ is an increasing and concave function. Then the following holds,*

$$f'(b)(b - a) \leq f(b) - f(a) \leq f'(a)(b - a). \tag{40}$$

**Lemma C.11** (Uniform bound (Mohri et al., 2018))**.** *Let $\mathcal{F}$ be the set of functions $f : \mathcal{Z} \to [0, M]$ and $\mu$ be a distribution over $\mathcal{Z}$. Let $S = \{z_i\}_{i=1}^n$ be a set of size $n$ i.i.d. drawn from $\mathcal{Z}$. Then, for any $\delta \in (0, 1)$, with probability at least $1 - \delta$ over the choice of $S$, we have*

$$\sup_{f \in \mathcal{F}} \left\{ \mathbb{E}_{Z \sim \mu}[f(Z)] - \frac{1}{n} \sum_{i=1}^n f(z_i) \right\} \leq 2\hat{\mathfrak{R}}_S(\mathcal{F}) + 3M \sqrt{\frac{1}{2n} \log \frac{2}{\delta}}.$$

We use the next two results, namely Talagrand's contraction lemma and Massart's Lemma, to estimate the Rademacher complexity.

**Lemma C.12** (Talagrand's contraction lemma (Shalev-Shwartz & Ben-David, 2014))**.** *Let $\phi_i : \mathbb{R} \to \mathbb{R}$ ($i \in \{1, \ldots, n\}$) be $L$-Lipschitz functions and $\mathcal{F}_r$ be a set of functions from $\mathcal{Z}$ to $\mathbb{R}$. Then it follows that for any $\{z_i\}_{i=1}^n \subset \mathcal{Z}$,*

$$\mathbb{E}_\sigma \left[ \sup_{f \in \mathcal{F}_r} \frac{1}{n} \sum_{i=1}^n \sigma_i \phi_i(f(z_i)) \right] \leq L \mathbb{E}_\sigma \left[ \sup_{f \in \mathcal{F}_r} \frac{1}{n} \sum_{i=1}^n \sigma_i f(z_i) \right].$$

**Lemma C.13** (Massart's lemma (Massart, 2000))**.** *Assume that the hypothesis space $\mathcal{H}$ is finite. Let $B^2 :=$ $\max_{h \in \mathcal{H}} \left( \sum_{i=1}^{n} h^2(z_i) \right)$. Then*

$$\hat{\mathfrak{R}}_S(\mathcal{H}) \leq \frac{B\sqrt{2\log(\text{card}(\mathcal{H}))}}{n}.$$

**Lemma C.14** (Lemma 1 in (Xu & Raginsky, 2017))**.** *For any measureable function $f : \mathcal{Z} \to [0, M]$,*

$$\left| \mathbb{E}_{P_{X,Y}}[f(X,Y)] - \mathbb{E}_{P_X \otimes P_Y}[f(X,Y)] \right| \leq M\sqrt{\frac{I(X;Y)}{2}}.$$

**Lemma C.15** (Coupling Lemma)**.** *Assume the function $f : \mathcal{Z} \to [0, M]$ and the function $g : \mathbb{R}^+ \mapsto \mathbb{R}^+$ are $L$-Lipschitz. Then the following upper bound holds,*

$$\left| \mathbb{E}_{P_{X,Y}}[g \circ f(X,Y)] - \mathbb{E}_{P_X \otimes P_Y}[g \circ f(X,Y)] \right| \leq LM\sqrt{\frac{I(X;Y)}{2}}.$$

We recall that the notation $\mathbb{TV}$ denotes the total variation distance between probability distributions.

**Lemma C.16** (Kantorovich-Rubenstein duality of total variation distance, see (Polyanskiy & Wu, 2022))**.** *The Kantorovich-Rubenstein duality (variational representation) of the total variation distance is as follows:*

$$\mathbb{TV}(m_1, m_2) = \frac{1}{2L} \sup_{g \in \mathcal{G}_L} \left\{ \mathbb{E}_{Z \sim m_1}[g(Z)] - \mathbb{E}_{Z \sim m_2}[g(Z)] \right\}, \tag{41}$$

*where $\mathcal{G}_L = \{g : \mathcal{Z} \to \mathbb{R}, ||g||_\infty \leq L\}$.*

## D. Proofs and Details of Section 3

### D.1. Proofs and Details of Uniform bounds under Unbounded Loss

**Proposition 3.2** (**Restated**)**.** *Under Assumption 3.1, for $\gamma < 0$, the difference between the population risk and the tilted population risk satisfies*

$$0 \leq \text{R}(h, \mu) - \text{R}_\gamma(h, \mu^{\otimes n}) \leq |\gamma|^\epsilon \kappa_u^{1+\epsilon}. \tag{42}$$

*Proof.* The proof follows from Lemma C.9. $\qquad\square$

> **Proposition 3.3** (**Restated**)**.** *Given Assumption 3.1, for any fixed $h \in \mathcal{H}$ with probability at least $(1 - \delta)$, then the following upper bound holds on the tilted generalization error for $\gamma < 0$ and some $\epsilon \in (0, 1]$,*
>
> $$\text{gen}_\gamma(h, S) \leq \frac{2\exp(-\gamma\kappa_u)}{|\gamma|}\sqrt{\frac{2^\epsilon|\gamma|^{1+\epsilon}\kappa_u^{1+\epsilon}\log(2/\delta)}{n}} + \frac{4\exp(-\gamma\kappa_u)\log(2/\delta)}{3n|\gamma|} + |\gamma|^\epsilon\kappa_u^{1+\epsilon}. \tag{43}$$

*Proof.* Using Bernstein's inequality, Lemma C.7, for $X_i = \exp(\gamma\ell(h, Z_i))$ and considering $0 < X_i < 1$, we have

$$\frac{1}{n}\sum_{i=1}^{n} \exp\gamma\ell(h, Z_i) \leq \mathbb{E}[\exp(\gamma\ell(h, \tilde{Z}))] + \sqrt{\frac{4\text{Var}(\exp(\gamma\ell(h, \tilde{Z})))\log(2/\delta)}{n}} + \frac{4\log(2/\delta)}{3n},$$

where we also used that

$$\log(x + y) = \log(y) + \log(1 + \frac{x}{y}) \leq \log(y) + \frac{x}{y} \text{ for } y > x > 0. \tag{44}$$

Thus,

$$\log \left( \frac{1}{n} \sum_{i=1}^{n} \exp \gamma \ell(h, Z_i) \right)$$

$$\leq \log \left( \mathbb{E}[\exp(\gamma \ell(h, \tilde{Z}))] \right)$$

$$+ \frac{1}{\mathbb{E}[\exp(\gamma \ell(h, \tilde{Z}))]} \sqrt{\frac{4 \mathrm{Var}(\exp(\gamma \ell(h, \tilde{Z}))) \log(2/\delta)}{n}} + \frac{1}{\mathbb{E}[\exp(\gamma \ell(h, \tilde{Z}))]} \frac{4 \log(2/\delta)}{3n}$$

$$\leq \log \left( \mathbb{E}[\exp(\gamma \ell(h, \tilde{Z}))] \right)$$

$$+ \exp(-\gamma \kappa_u) \sqrt{\frac{4 \mathrm{Var}(\exp(\gamma \ell(h, \tilde{Z}))) \log(2/\delta)}{n}} + \exp(-\gamma \kappa_u) \frac{4 \log(2/\delta)}{3n}.$$

Therefore, for $\gamma < 0$ we have

$$\frac{1}{\gamma} \log \left( \mathbb{E}[\exp(\gamma \ell(h, \tilde{Z}))] \right) - \frac{1}{\gamma} \log \left( \frac{1}{n} \sum_{i=1}^{n} \exp \gamma \ell(h, Z_i) \right)$$

$$\leq \frac{\exp(-\gamma \kappa_u)}{|\gamma|} \sqrt{\frac{4 \mathrm{Var}(\exp(\gamma \ell(h, \tilde{Z}))) \log(2/\delta)}{n}} + \exp(-\gamma \kappa_u) \frac{4 \log(2/\delta)}{3n|\gamma|}.$$

Using Lemma C.3, completes the proof. $\square$

---

**Proposition 3.4 (Restated).** *Given Assumption 3.1, there exists a $\zeta \in (0, 1)$ such that for $n \geq \frac{(4\gamma^2 \kappa_u^2 + 8/3\zeta) \log(2/\delta)}{\zeta^2 \exp(2\gamma \kappa_u)}$, for any fixed $h \in \mathcal{H}$ with probability at least $(1-\delta)$, and $\gamma < 0$, the following lower bound on the tilted generalization error holds,*

$$\mathrm{gen}_\gamma(h, S) \geq -\frac{2 \exp(|\gamma| \kappa_u)}{(1 - \zeta)|\gamma|} \sqrt{\frac{2^\epsilon |\gamma|^{1+\epsilon} \kappa_u^{1+\epsilon} \log(2/\delta)}{n}} - \frac{4 \exp(|\gamma| \kappa_u)(\log(2/\delta))}{3n|\gamma|(1 - \zeta)}. \tag{45}$$

---

*Proof.* Recall that $\tilde{Z} \sim \mu$ and

$$\mathrm{gen}_\gamma(h, S) = \mathrm{R}(h, \mu) - \frac{1}{\gamma} \log(\mathbb{E}[\exp(\gamma \ell(h, \tilde{Z}))])$$

$$+ \frac{1}{\gamma} \log(\mathbb{E}[\exp(\gamma \ell(h, \tilde{Z}))]) - \frac{1}{\gamma} \log \left( \frac{1}{n} \sum_{i=1}^{n} \exp(\gamma \ell(h, Z_i)) \right).$$

First, we apply Lemma C.9 to yield $\mathrm{R}(h, \mu) - \frac{1}{\gamma} \log(\mathbb{E}[\exp(\gamma \ell(h, \tilde{Z}))]) \geq 0$. Next we focus on the second line of this display. Bernstein's inequality, Lemma C.7, for $X_i = \exp(\gamma \ell(h, Z_i))$, so that $0 < X_i < 1$, gives that with probability at least $(1 - \delta)$,

$$\frac{1}{n} \sum_{i=1}^{n} \exp \gamma \ell(h, Z_i) \geq \mathbb{E}[\exp(\gamma \ell(h, \tilde{Z}))] - \sqrt{\frac{4 \mathrm{Var}(\exp(\gamma \ell(h, \tilde{Z}))) \log(2/\delta)}{n}} - \frac{4 \log(2/\delta)}{3n}. \tag{46}$$

Assume for now that there is a $\zeta \in (0, 1)$ such that

$$\sqrt{\frac{4 \mathrm{Var}(\exp(\gamma \ell(h, \tilde{Z}))) \log(2/\delta)}{n}} + \frac{4 \log(2/\delta)}{3n} \leq \zeta \mathbb{E}[\exp(\gamma \ell(h, \tilde{Z}))]. \tag{47}$$

As $\log(y - x) = \log(y) + \log(1 - \frac{x}{y}) \geq \log(y) - \frac{x}{y-x}$ for $y > x > 0$, then by taking $y = \mathbb{E}[\exp(\gamma\ell(h, \tilde{Z}))]$ and $x = \sqrt{\frac{4\operatorname{Var}(\exp(\gamma\ell(h,\tilde{Z})))\log(2/\delta)}{n}} + \frac{4\log(2/\delta)}{3n}$, so that with (47) we have $y - x \geq (1 - \zeta)y > 0$, taking logarithms on both sides of (46) gives that with probability at least $(1 - \delta)$,

$$\log\left(\frac{1}{n}\sum_{i=1}^{n}\exp\gamma\ell(h, Z_i)\right)$$

$$\geq \log\left(\mathbb{E}[\exp(\gamma\ell(h, \tilde{Z}))]\right) - \frac{1}{\mathbb{E}[\exp(\gamma\ell(h, \tilde{Z}))]}\left(\sqrt{\frac{4\operatorname{Var}(\exp(\gamma\ell(h, \tilde{Z})))\log(2/\delta)}{n}} + \frac{4\log(2/\delta)}{3n}\right)$$

$$\geq \log\left(\mathbb{E}[\exp(\gamma\ell(h, \tilde{Z}))]\right) - \frac{1}{(1-\zeta)\mathbb{E}[\exp(\gamma\ell(h, \tilde{Z}))]}\sqrt{\frac{4\operatorname{Var}(\exp(\gamma\ell(h, \tilde{Z})))\log(2/\delta)}{n}} \tag{48}$$
$$- \frac{1}{(1-\zeta)\mathbb{E}[\exp(\gamma\ell(h, \tilde{Z}))]}\frac{4\log(2/\delta)}{3n}$$

$$\geq \log\left(\mathbb{E}[\exp(\gamma\ell(h, \tilde{Z}))]\right) - \frac{\exp(-\gamma\kappa_u)}{(1-\zeta)}\sqrt{\frac{4\operatorname{Var}(\exp(\gamma\ell(h, \tilde{Z})))\log(2/\delta)}{n}} - \frac{\exp(-\gamma\kappa_u)}{(1-\zeta)}\frac{4\log(2/\delta)}{3n}$$

$$\geq \log\left(\mathbb{E}[\exp(\gamma\ell(h, \tilde{Z}))]\right) - \frac{|\gamma|2\kappa_u\exp(-\gamma\kappa_u)}{(1-\zeta)}\sqrt{\frac{\log(2/\delta)}{n}} - \frac{\exp(-\gamma\kappa_u)}{(1-\zeta)}\frac{4\log(2/\delta)}{3n}.$$

Here we used that by Assumption 3.1 and Lemma C.8,

$$\mathbb{E}[\exp(\gamma\ell(h, \tilde{Z}))] \leq \exp[\mathbb{E}(\gamma\ell(h, \tilde{Z}))] \leq \exp(\gamma\kappa_u)$$

and by Assumption 3.1 and Lemma C.3,

$$\operatorname{Var}(\exp(\gamma\ell(h, \tilde{Z}))) \leq 2^\epsilon|\gamma|^{1+\epsilon}\mathbb{E}[\ell(h, \tilde{Z})^{1+\epsilon}] \leq 2^\epsilon|\gamma|^{1+\epsilon}\kappa_u^{1+\epsilon}.$$

This gives the stated bound assuming that (47) holds. In order to satisfy (47), viewing (47) as a quadratic inequality in $\sqrt{n}$ and using that $(a + b)^2 \leq 2a^2 + 2b^2$ yields

$$n \geq \frac{(4\operatorname{Var}(\exp(\gamma\ell(h, \tilde{Z}))) + 8/3\zeta\mathbb{E}[\exp(\gamma\ell(h, \tilde{Z}))])\log(2/\delta)}{\zeta^2(\mathbb{E}[\exp(\gamma\ell(h, \tilde{Z}))])^2},$$

Now applying $\exp(\gamma\kappa_u) \leq \mathbb{E}[\exp(\gamma\ell(h, \tilde{Z}))] \leq 1$ and $\operatorname{Var}(\exp(\gamma\ell(h, \tilde{Z}))) \leq 2^\epsilon|\gamma|^{1+\epsilon}\kappa_u^{1+\epsilon}$ completes the proof. $\qquad\square$

---

**Theorem 3.5 (Restated).** *Under the same assumptions in Proposition 3.4 and a finite hypothesis space, then for $n \geq \frac{(4|\gamma|^{1+\epsilon}\kappa_u^{1+\epsilon}+8/3\zeta)\log(2/\delta)}{\zeta^2\exp(2\gamma\kappa_u)}$, for $\gamma < 0$ and with probability at least $(1 - \delta)$, the absolute value of the titled generalization error satisfies*

$$\sup_{h\in\mathcal{H}}|\mathrm{gen}_\gamma(h, S)| \leq \frac{2\exp(|\gamma|\kappa_u)}{(1-\zeta)|\gamma|}\sqrt{\frac{2^\epsilon|\gamma|^{1+\epsilon}\kappa_u^{1+\epsilon}B(\delta)}{n}} + \frac{4\exp(|\gamma|\kappa_u)B(\delta)}{3n|\gamma|(1-\zeta)} + |\gamma|^\epsilon\kappa_u^{1+\epsilon}, \tag{49}$$

*where $B(\delta) = \log(\mathrm{card}(\mathcal{H})) + \log(2/\delta)$.*

*Proof.* Combining the upper and lower bounds, Proposition 3.3 and Proposition 3.4, we can derive the following bound for a fixed $h \in \mathcal{H}$,

$$|\mathrm{gen}_\gamma(h, S)| \leq \frac{2\exp(-\gamma\kappa_u)}{(1-\zeta)}\sqrt{\frac{2^\epsilon|\gamma|^{1+\epsilon}\kappa_u^{1+\epsilon}(\log(2/\delta))}{n}} + \frac{4\exp(-\gamma\kappa_u)(\log(2/\delta))}{3n|\gamma|(1-\zeta)} + |\gamma|^\epsilon\kappa_u^{1+\epsilon}. \tag{50}$$

Then, using the union bound completes the proof. $\qquad\square$

**Lemma 3.7.** *The excess risk of the tilted empirical risk satisfies,*

$$\mathfrak{E}_\gamma(\mu) \le 2 \sup_{h \in \mathcal{H}} |\mathrm{gen}_\gamma(h, S)|$$

*Proof.* It can be proved that,

$$\mathfrak{E}_\gamma(\mu) \le 2 \sup_{h \in \mathcal{H}} |\mathrm{gen}_\gamma(h, S)|$$

and

$$\mathrm{R}(h_\gamma^*(S), \mu) \le \hat{\mathrm{R}}_\gamma(h_\gamma^*(S), \mu) + U \quad \le \hat{\mathrm{R}}_\gamma(h^*(\mu), \mu) + U \quad \le \mathrm{R}(h^*(\mu), \mu) + 2U,$$

where $U = \sup_{h \in \mathcal{H}} |\mathrm{R}(h, \mu) - \hat{\mathrm{R}}_\gamma(h, S)| = \sup_{h \in \mathcal{H}} |\mathrm{gen}_\gamma(h, S)|$. $\qquad\square$

---

**Corollary D.1.** *Under the same assumption in Proposition 3.4 and a finite hypothesis space, then for $n \ge \frac{(4\gamma^2 \kappa_u^2 + 8/3\zeta) \log(2/\delta)}{\zeta^2 \exp(2\gamma\kappa_u)}$ with probability at least $(1 - \delta)$ and $\gamma < 0$, the excess risk of tilted empirical risk satisfies,*

$$\mathfrak{E}_\gamma(\mu) \le \frac{4 \exp(|\gamma|\kappa_u)}{(1 - \zeta)|\gamma|} \sqrt{\frac{2^\epsilon |\gamma|^{1+\epsilon} \kappa_u^{1+\epsilon} B(\delta)}{n}} + 2|\gamma|^\epsilon \kappa_u^{1+\epsilon} + \frac{8 \exp(|\gamma|\kappa_u) B(\delta)}{3n|\gamma|(1 - \zeta)}, \tag{51}$$

*where $B(\delta) = \log(\mathrm{card}(\mathcal{H})) + \log(2/\delta)$ and $\mathfrak{E}_\gamma(\mu)$ is defined in (7).*

---

*Proof.* Combining Theorem 3.5 with Lemma 3.7 completes the proof. $\qquad\square$

## D.2. Proofs and Details of Information-theoretic Bounds under Unbounded Loss

**Proposition D.2.** *Under Assumption 3.8, for $\gamma < 0$, it holds for some $\epsilon \in (0, 1]$ that,*

$$0 \le \mathrm{R}(H, P_H \otimes \mu) - \mathrm{R}_\gamma(H, P_H \otimes \mu) \le |\gamma|^\epsilon \kappa_t^{1+\epsilon},$$
$$- |\gamma|^\epsilon \kappa_t^{1+\epsilon} \le \mathrm{R}_\gamma(H, P_{H,S}) - \mathbb{E}_{P_{H,S}}[\hat{\mathrm{R}}_\gamma(H, S)] \le 0. \tag{52}$$

*Proof.* The proof follows from Lemma C.9. $\qquad\square$

---

**Proposition 3.9.** *Given Assumption 3.8, the following inequality holds for $\gamma < 0$,*

$$\mathrm{R}_\gamma(H, P_H \otimes \mu) - \mathrm{R}_\gamma(H, P_{H,S}) \le \begin{cases} \frac{\exp(|\gamma|\kappa_t)}{|\gamma|} \sqrt{\frac{2|\gamma|^{1+\epsilon} \kappa_t^{1+\epsilon} I(H;S)}{n}}, & \text{if } \frac{I(H;S)}{n} \le \frac{2^\epsilon |\gamma|^{1+\epsilon} \kappa_t^{1+\epsilon}}{2} \\ \frac{\exp(|\gamma|\kappa_t)}{|\gamma|} \left( \frac{I(H;S)}{n} + \frac{2^\epsilon |\gamma|^{1+\epsilon} \kappa_t^{1+\epsilon}}{2} \right), & \text{if } \frac{I(H;S)}{n} > \frac{2^\epsilon |\gamma|^{1+\epsilon} \kappa_t^{1+\epsilon}}{2} \end{cases}$$

---

*Proof.* For $\gamma < 0$, we use that $0 \le \exp(\gamma\ell(H, \tilde{Z})) \le 1$ and $\mathrm{Var}(\exp(\gamma\ell(H, \tilde{Z}))) \le |\gamma|^{1+\epsilon} \mathbb{E}[\ell(H, \tilde{Z})^{1+\epsilon}] \le |\gamma|^{1+\epsilon} \kappa_t^{1+\epsilon}$. Note that the variable $\exp(\gamma\ell(H, \tilde{Z}))$ is sub-exponential with parameters $(|\gamma|^{1+\epsilon} \kappa_t^{1+\epsilon}, 1)$ under the distribution $P_H \otimes \mu$. Using the approach in (Bu et al., 2020; Aminian et al., 2021a) for the sub-exponential case, we have

$$\left| \mathbb{E}_{P_H \otimes \mu}[\exp(\gamma\ell(H, \tilde{Z}))] - \mathbb{E}_{P_{H,S}}\left[ \frac{1}{n} \sum_{i=1}^n \exp(\gamma\ell(H, Z_i)) \right] \right|$$
$$\le \begin{cases} \sqrt{2|\gamma|^{1+\epsilon} \kappa_t^{1+\epsilon} \frac{I(H;S)}{n}} & \text{if } \frac{I(H;S)}{n} \le \frac{2^\epsilon |\gamma|^{1+\epsilon} \kappa_t^{1+\epsilon}}{2} \\ \frac{I(H;S)}{n} + \frac{2^\epsilon |\gamma|^{1+\epsilon} \kappa_t^{1+\epsilon}}{2} & \text{if } \frac{I(H;S)}{n} > \frac{2^\epsilon |\gamma|^{1+\epsilon} \kappa_t^{1+\epsilon}}{2}. \end{cases} \tag{53}$$

Therefore, we have for $\frac{I(H;S)}{n} \le \frac{2^\epsilon |\gamma|^{1+\epsilon} \kappa_t^{1+\epsilon}}{2}$,

$$\mathbb{E}_{P_{H,S}}\left[ \frac{1}{n} \sum_{i=1}^n \exp(\gamma\ell(H, Z_i)) \right] \le \left( \mathbb{E}_{P_H \otimes \mu}\left[ \exp(\gamma\ell(H, \tilde{Z})) \right] + \sqrt{2|\gamma|^{1+\epsilon} \kappa_t^{1+\epsilon} \frac{I(H;S)}{n}} \right). \tag{54}$$

Using (44) gives

$$\frac{1}{\gamma} \log \left( \mathbb{E}_{P_{H,S}}[\frac{1}{n} \sum_{i=1}^{n} \exp(\gamma \ell(H, Z_i))] \right) - \frac{1}{\gamma} \log(\mathbb{E}_{P_H \otimes \mu}[\exp(\gamma \ell(H, \tilde{Z}))])$$

$$\geq \frac{1}{\gamma \mathbb{E}_{P_H \otimes \mu}[\exp(\gamma \ell(H, \tilde{Z}))]} \sqrt{2|\gamma|^{1+\epsilon} \kappa_t^{1+\epsilon} \frac{I(H; S)}{n}}. \tag{55}$$

For $\frac{I(H;S)}{n} > \frac{2^\epsilon |\gamma|^{1+\epsilon} \kappa_t^{1+\epsilon}}{2}$, we have

$$\mathbb{E}_{P_{H,S}}\left[\frac{1}{n} \sum_{i=1}^{n} \exp(\gamma \ell(H, Z_i))\right] \leq \left( \mathbb{E}_{P_H \otimes \mu}\left[ \exp(\gamma \ell(H, \tilde{Z})) \right] + \frac{I(H; S)}{n} + \frac{2^\epsilon |\gamma|^{1+\epsilon} \kappa_t^{1+\epsilon}}{2} \right). \tag{56}$$

Using (44) again, we obtain,

$$\frac{1}{\gamma} \log \left( \mathbb{E}_{P_{H,S}}[\frac{1}{n} \sum_{i=1}^{n} \exp(\gamma \ell(H, Z_i))] \right) - \frac{1}{\gamma} \log(\mathbb{E}_{P_H \otimes \mu}[\exp(\gamma \ell(H, \tilde{Z}))])$$

$$\geq \frac{1}{\gamma \mathbb{E}_{P_H \otimes \mu}[\exp(\gamma \ell(H, \tilde{Z}))]} \left( \frac{I(H; S)}{n} + \frac{2^\epsilon |\gamma|^{1+\epsilon} \kappa_t^{1+\epsilon}}{2} \right). \tag{57}$$

As under Assumption 3.8 and Lemma C.8, we have $\exp(\gamma \kappa_t) \leq \mathbb{E}_{P_H \otimes \mu}[\exp(\gamma \ell(H, \tilde{Z}))]$, the final result follows. $\square$

---

**Proposition 3.10.** *Under Assumption 3.8, there exists $\zeta \in (0,1)$ such that one of the following cases holds,*
*(a)* $\zeta \leq \frac{2^\epsilon |\gamma|^{1+\epsilon} \kappa_t^{1+\epsilon} \exp(|\gamma| \kappa_t)}{2}$ *and* $n \geq \frac{2^\epsilon |\gamma|^{1+\epsilon} \kappa_t^{1+\epsilon} I(H;S)}{\zeta^2 \exp(2\gamma \kappa_t)}$,
*(b)* $\zeta > \frac{2^\epsilon |\gamma|^{1+\epsilon} \kappa_t^{1+\epsilon} \exp(|\gamma| \kappa_t)}{2}$ *and* $n \geq \frac{2 I(H;S)}{2^\epsilon |\gamma|^{1+\epsilon} \kappa_t^{1+\epsilon}}$,
*(c)* $\zeta > \frac{2^\epsilon |\gamma|^{1+\epsilon} \kappa_t^{1+\epsilon} \exp(|\gamma| \kappa_t)}{2}$ *and* $\frac{2 I(H;S)}{2^\epsilon |\gamma|^{1+\epsilon} \kappa_t^{1+\epsilon}} > n \geq \frac{2 I(H;S)}{2\zeta \exp(\gamma \kappa_t) - |\gamma|^{1+\epsilon} \kappa_t^{1+\epsilon}}$,
*Then, the following lower bounds on non-linear expected tilted generalization error hold,*

$$\widehat{\text{gen}}_\gamma(H, S) \geq \begin{cases} \frac{-\exp(|\gamma| \kappa_t)}{|\gamma|(1-\zeta)} \sqrt{\frac{2|\gamma|^{1+\epsilon} \kappa_t^{1+\epsilon} I(H;S)}{n}}, & \text{if (a) or (b) hold,} \\ \frac{-\exp(|\gamma| \kappa_t)}{(1-\zeta)|\gamma|} \left( \frac{I(H;S)}{n} + \frac{2^\epsilon |\gamma|^{1+\epsilon} \kappa_t^{1+\epsilon}}{2} \right), & \text{if (c) holds.} \end{cases}$$

---

*Proof.* Note that, we have,

$$\left| \mathbb{E}_{P_H \otimes \mu}[\exp(\gamma \ell(H, \tilde{Z}))] - \mathbb{E}_{P_{H,S}}[\frac{1}{n} \sum_{i=1}^{n} \exp(\gamma \ell(H, Z_i))] \right|$$

$$\leq \begin{cases} \sqrt{2|\gamma|^{1+\epsilon} \kappa_t^{1+\epsilon} \frac{I(H;S)}{n}} & \text{if } \frac{I(H;S)}{n} \leq \frac{2^\epsilon |\gamma|^{1+\epsilon} \kappa_t^{1+\epsilon}}{2} \\ \frac{I(H;S)}{n} + \frac{2^\epsilon |\gamma|^{1+\epsilon} \kappa_t^{1+\epsilon}}{2} & \text{if } \frac{I(H;S)}{n} > \frac{2^\epsilon |\gamma|^{1+\epsilon} \kappa_t^{1+\epsilon}}{2}. \end{cases} \tag{58}$$

For $\frac{I(H;S)}{n} \leq \frac{2^\epsilon |\gamma|^{1+\epsilon} \kappa_t^{1+\epsilon}}{2}$, we have,

$$\mathbb{E}_{P_{H,S}}\left[\frac{1}{n} \sum_{i=1}^{n} \exp(\gamma \ell(H, Z_i))\right] \geq \left( \mathbb{E}_{P_H \otimes \mu}\left[ \exp(\gamma \ell(H, \tilde{Z})) \right] - \sqrt{2|\gamma|^{1+\epsilon} \kappa_t^{1+\epsilon} \frac{I(H; S)}{n}} \right). \tag{59}$$

Assume for now that there is a $\zeta \in (0,1)$ such that

$$\sqrt{2|\gamma|^{1+\epsilon} \kappa_t^{1+\epsilon} \frac{I(H; S)}{n}} \leq \zeta \mathbb{E}_{P_H \otimes \mu}\left[ \exp(\gamma \ell(H, \tilde{Z})) \right] \tag{60}$$

As $\log(y - x) = \log(y) + \log(1 - \frac{x}{y}) \geq \log(y) - \frac{x}{y-x}$ for $y > x > 0$, then by taking $y = \mathbb{E}_{P_H \otimes \mu}\left[\exp(\gamma\ell(H, \tilde{Z}))\right]$ and $x = \sqrt{2|\gamma|^{1+\epsilon}\kappa_t^{1+\epsilon}\frac{I(H;S)}{n}}$, so that with (47) we have $y - x \geq (1 - \zeta)y > 0$, taking logarithms on both sides of (59) gives that

$$\frac{1}{\gamma}\log(\mathbb{E}_{P_H \otimes \mu}[\exp(\gamma\ell(H, \tilde{Z}))]) - \frac{1}{\gamma}\log\left(\mathbb{E}_{P_{H,S}}[\frac{1}{n}\sum_{i=1}^{n}\exp(\gamma\ell(H, Z_i))]\right)$$
$$\geq \frac{-1}{\gamma(1-\zeta)\mathbb{E}_{P_H \otimes \mu}[\exp(\gamma\ell(H, \tilde{Z}))]}\sqrt{\frac{2|\gamma|^{1+\epsilon}\kappa_t^{1+\epsilon}I(H;S)}{n}},$$

(61)

where it holds for $n \geq \frac{2|\gamma|^{1+\epsilon}\kappa_t^{1+\epsilon}I(H;S)}{\zeta^2\exp(2\gamma\kappa_t)}$. As we also have $\frac{I(H;S)}{n} \leq \frac{2^\epsilon|\gamma|^{1+\epsilon}\kappa_t^{1+\epsilon}}{2}$, we consider the condition $n \geq \max\left(\frac{2|\gamma|^{1+\epsilon}\kappa_t^{1+\epsilon}I(H;S)}{\zeta^2\exp(2\gamma\kappa_t)}, \frac{2I(H;S)}{2^\epsilon|\gamma|^{1+\epsilon}\kappa_t^{1+\epsilon}}\right)$.

For $\frac{2I(H;S)}{2^\epsilon|\gamma|^{1+\epsilon}\kappa_t^{1+\epsilon}} > n$, we have

$$\mathbb{E}_{P_{H,S}}\left[\frac{1}{n}\sum_{i=1}^{n}\exp(\gamma\ell(H, Z_i))\right] \geq \left(\mathbb{E}_{P_H \otimes \mu}\left[\exp(\gamma\ell(H, \tilde{Z}))\right] - \frac{I(H;S)}{n} - \frac{2^\epsilon|\gamma|^{1+\epsilon}\kappa_t^{1+\epsilon}}{2}\right).$$

(62)

As $\log(y - x) = \log(y) + \log(1 - \frac{x}{y}) \geq \log(y) - \frac{x}{y-x}$, we obtain,

$$\frac{1}{\gamma}\log\left(\mathbb{E}_{P_{H,S}}[\frac{1}{n}\sum_{i=1}^{n}\exp(\gamma\ell(H, Z_i))]\right) - \frac{1}{\gamma}\log(\mathbb{E}_{P_H \otimes \mu}[\exp(\gamma\ell(H, \tilde{Z}))])$$
$$\leq \frac{-1}{\gamma(1-\zeta')\mathbb{E}_{P_H \otimes \mu}[\exp(\gamma\ell(H, \tilde{Z}))]}\left(\frac{I(H;S)}{n} + \frac{2^\epsilon|\gamma|^{1+\epsilon}\kappa_t^{1+\epsilon}}{2}\right),$$

(63)

Assume for now that there is a $\zeta' \in (0, 1)$ such that

$$\zeta'\mathbb{E}_{P_H \otimes \mu}\left[\exp(\gamma\ell(H, \tilde{Z}))\right] \geq \frac{I(H;S)}{n} + \frac{2^\epsilon|\gamma|^{1+\epsilon}\kappa_t^{1+\epsilon}}{2}.$$

(64)

where it holds for $\frac{2I(H;S)}{2\zeta'\exp(\gamma\kappa_t) - |\gamma|^{1+\epsilon}\kappa_t^{1+\epsilon}} \leq n$. Therefore, the final condition is

$$\frac{2I(H;S)}{2\zeta'\exp(\gamma\kappa_t) - |\gamma|^{1+\epsilon}\kappa_t^{1+\epsilon}} \leq n \leq \frac{2I(H;S)}{2^\epsilon|\gamma|^{1+\epsilon}\kappa_t^{1+\epsilon}}$$

Note that, we also have due to $2ab \leq a^2 + b^2$,

$$\sqrt{\frac{2|\gamma|^{1+\epsilon}\kappa_t^{1+\epsilon}I(H;S)}{n}} \leq \frac{I(H;S)}{n} + \frac{2^\epsilon|\gamma|^{1+\epsilon}\kappa_t^{1+\epsilon}}{2}$$

(65)

Therefore, we can choose $\zeta = \zeta'$. Furthermore, we can also discuss the following cases.

- If $\zeta\exp(\gamma\kappa_t) \leq \frac{2^\epsilon|\gamma|^{1+\epsilon}\kappa_t^{1+\epsilon}}{2}$ holds, then for $n \geq \frac{2^\epsilon|\gamma|^{1+\epsilon}\kappa_t^{1+\epsilon}I(H;S)}{\zeta^2\exp(2\gamma\kappa_t)}$ we have,

$$\mathrm{R}_\gamma(H, P_H \otimes \mu) - \mathrm{R}_\gamma(H, P_{H,S}) \geq \frac{-\exp(|\gamma|\kappa_t)}{|\gamma|(1-\zeta)}\sqrt{\frac{2|\gamma|^{1+\epsilon}\kappa_t^{1+\epsilon}I(H;S)}{n}}.$$

- If $\zeta\exp(\gamma\kappa_t) \geq \frac{2^\epsilon|\gamma|^{1+\epsilon}\kappa_t^{1+\epsilon}}{2}$ holds, then we have,

$$\mathrm{R}_\gamma(H, P_H \otimes \mu) - \mathrm{R}_\gamma(H, P_{H,S})$$
$$\geq \begin{cases} \frac{-\exp(|\gamma|\kappa_t)}{|\gamma|(1-\zeta)}\sqrt{\frac{2|\gamma|^{1+\epsilon}\kappa_t^{1+\epsilon}I(H;S)}{n}}, & \text{if } \frac{2I(H;S)}{2^\epsilon|\gamma|^{1+\epsilon}\kappa_t^{1+\epsilon}} \leq n \\ \frac{-\exp(|\gamma|\kappa_t)}{(1-\zeta)|\gamma|}\left(\frac{I(H;S)}{n} + \frac{2^\epsilon|\gamma|^{1+\epsilon}\kappa_t^{1+\epsilon}}{2}\right), & \text{if } \frac{2I(H;S)}{2^\epsilon|\gamma|^{1+\epsilon}\kappa_t^{1+\epsilon}}) > n \geq \frac{2I(H;S)}{2\zeta'\exp(\gamma\kappa_t) - |\gamma|^{1+\epsilon}\kappa_t^{1+\epsilon}}. \end{cases}$$

□

Using Proposition 3.9, we can obtain the following upper bound on the expected tilted generalization error.

**Proposition D.3.** *Given Assumption 3.8, the following upper bound holds on the expected tilted generalization error for* $\gamma < 0$,

$$\overline{\mathrm{gen}}_\gamma(H, S) \leq \begin{cases} \frac{\exp(|\gamma|\kappa_t)}{|\gamma|} \sqrt{\frac{2|\gamma|^{1+\epsilon}\kappa_t^{1+\epsilon}I(H;S)}{n}} + |\gamma|^\epsilon \kappa_t^{1+\epsilon}, & \text{if } \frac{I(H;S)}{n} \leq \frac{2^\epsilon|\gamma|^{1+\epsilon}\kappa_t^{1+\epsilon}}{2} \\ \frac{\exp(|\gamma|\kappa_t)}{|\gamma|} \left( \frac{I(H;S)}{n} + \frac{2^\epsilon|\gamma|^{1+\epsilon}\kappa_t^{1+\epsilon}}{2} \right) + |\gamma|^\epsilon \kappa_t^{1+\epsilon}, & \text{if } \frac{I(H;S)}{n} > \frac{2^\epsilon|\gamma|^{1+\epsilon}\kappa_t^{1+\epsilon}}{2} \end{cases}$$

*Proof.* We use the following decomposition,

$$\mathbb{E}_{P_{H,S}}[\mathrm{gen}_\gamma(H, S)] = \mathbb{E}_{P_{H,S}}[\mathrm{R}(H, \mu)] - \mathrm{R}_\gamma(H, P_H \otimes \mu) + \mathrm{R}_\gamma(H, P_H \otimes \mu) - \mathrm{R}_\gamma(H, P_{H,S}) \tag{66}$$
$$+ \mathrm{R}_\gamma(H, P_{H,S}) - \mathbb{E}_{P_{H,S}}[\widehat{\mathrm{R}}_\gamma(H, S)].$$

Then, using Lemma C.9, we have

$$\mathbb{E}_{P_{H,S}}[\mathrm{R}(H, \mu)] - \mathrm{R}_\gamma(H, P_H \otimes \mu) \leq |\gamma|^\epsilon \mathbb{E}_{P_H \otimes \mu}[\ell^{1+\epsilon}(H, \tilde{Z})] \leq |\gamma|^\epsilon \kappa_t^{1+\epsilon}, \tag{67}$$

and now Jensen's inequality for $\gamma < 0$ yields

$$\mathrm{R}_\gamma(H, P_{H,S}) - \mathbb{E}_{P_{H,S}}[\widehat{\mathrm{R}}_\gamma(H, S)] \leq 0. \tag{68}$$

Applying Proposition 3.9, we obtain

$$\mathrm{R}_\gamma(H, P_H \otimes \mu) - \mathrm{R}_\gamma(H, P_{H,S}) \leq \begin{cases} \frac{\exp(|\gamma|\kappa_t)}{|\gamma|} \sqrt{\frac{2|\gamma|^{1+\epsilon}\kappa_t^{1+\epsilon}I(H;S)}{n}}, & \text{if } \frac{I(H;S)}{n} \leq \frac{2^\epsilon|\gamma|^{1+\epsilon}\kappa_t^{1+\epsilon}}{2} \\ \frac{\exp(|\gamma|\kappa_t)}{|\gamma|} \left( \frac{I(H;S)}{n} + \frac{2^\epsilon|\gamma|^{1+\epsilon}\kappa_t^{1+\epsilon}}{2} \right), & \text{if } \frac{I(H;S)}{n} > \frac{2^\epsilon|\gamma|^{1+\epsilon}\kappa_t^{1+\epsilon}}{2} \end{cases}$$

Combining (68), (69) and (67) with (66) completes the proof. □

**Proposition D.4.** *Under Assumption 3.8, there exists* $\zeta \in (0, 1)$ *such that one of the following cases holds,*

*(a)* $\zeta \leq \frac{2^\epsilon|\gamma|^{1+\epsilon}\kappa_t^{1+\epsilon}\exp(|\gamma|\kappa_t)}{2}$ *and* $n \geq \frac{2^\epsilon|\gamma|^{1+\epsilon}\kappa_t^{1+\epsilon}I(H;S)}{\zeta^2 \exp(2\gamma\kappa_t)}$,

*(b)* $\zeta > \frac{2^\epsilon|\gamma|^{1+\epsilon}\kappa_t^{1+\epsilon}\exp(|\gamma|\kappa_t)}{2}$ *and* $n \geq \frac{2I(H;S)}{2^\epsilon|\gamma|^{1+\epsilon}\kappa_t^{1+\epsilon}}$,

*(c)* $\zeta > \frac{2^\epsilon|\gamma|^{1+\epsilon}\kappa_t^{1+\epsilon}\exp(|\gamma|\kappa_t)}{2}$ *and* $\frac{2I(H;S)}{2^\epsilon|\gamma|^{1+\epsilon}\kappa_t^{1+\epsilon}} > n \geq \frac{2I(H;S)}{2\zeta \exp(\gamma\kappa_t) - |\gamma|^{1+\epsilon}\kappa_t^{1+\epsilon}}$,

*Then, the following lower bounds on expected tilted generalization error hold,*

$$\overline{\mathrm{gen}}_\gamma(H, S) \geq \begin{cases} \frac{-\exp(|\gamma|\kappa_t)}{|\gamma|(1-\zeta)} \sqrt{\frac{2|\gamma|^{1+\epsilon}\kappa_t^{1+\epsilon}I(H;S)}{n}} - |\gamma|^\epsilon \kappa_t^{1+\epsilon}, & \text{if (a) or (b) hold,} \\ \frac{-\exp(|\gamma|\kappa_t)}{(1-\zeta)|\gamma|} \left( \frac{I(H;S)}{n} + \frac{2^\epsilon|\gamma|^{1+\epsilon}\kappa_t^{1+\epsilon}}{2} \right) - |\gamma|^\epsilon \kappa_t^{1+\epsilon}, & \text{if (c) holds.} \end{cases}$$

*Proof.* The proof is similar to Proposition D.3 and using Proposition 3.10. □

---

**Theorem 3.11.** *Under Assumption 3.8, there exists* $\zeta \in (0, 1)$ *such that one of the same cases in*

*(a)* $\zeta \leq \frac{2^\epsilon|\gamma|^{1+\epsilon}\kappa_t^{1+\epsilon}\exp(|\gamma|\kappa_t)}{2}$ *and* $n \geq \frac{2^\epsilon|\gamma|^{1+\epsilon}\kappa_t^{1+\epsilon}I(H;S)}{\zeta^2 \exp(2\gamma\kappa_t)}$,

*(b)* $\zeta > \frac{2^\epsilon|\gamma|^{1+\epsilon}\kappa_t^{1+\epsilon}\exp(|\gamma|\kappa_t)}{2}$ *and* $n \geq \frac{2I(H;S)}{2^\epsilon|\gamma|^{1+\epsilon}\kappa_t^{1+\epsilon}}$,

*(c)* $\zeta > \frac{2^\epsilon|\gamma|^{1+\epsilon}\kappa_t^{1+\epsilon}\exp(|\gamma|\kappa_t)}{2}$ *and* $\frac{2I(H;S)}{2^\epsilon|\gamma|^{1+\epsilon}\kappa_t^{1+\epsilon}} > n \geq \frac{2I(H;S)}{2\zeta \exp(\gamma\kappa_t) - |\gamma|^{1+\epsilon}\kappa_t^{1+\epsilon}}$,

*Then, the following lower bounds on non-linear expected generalization error hold,*

$$|\overline{\mathrm{gen}}_\gamma(H, S)| \leq \begin{cases} \frac{\exp(|\gamma|\kappa_t)}{|\gamma|(1-\zeta)} \sqrt{\frac{2\kappa_t^2 I(H;S)}{n}} + |\gamma|^\epsilon \kappa_t^{1+\epsilon}, & \text{if (a) or (b) hold,} \\ \frac{\exp(|\gamma|\kappa_t)}{(1-\zeta)|\gamma|} \left( \frac{I(H;S)}{n} + \frac{2^\epsilon|\gamma|^{1+\epsilon}\kappa_t^{1+\epsilon}}{2} \right) + |\gamma|^\epsilon \kappa_t^{1+\epsilon}, & \text{if (c) holds.} \end{cases}$$

*Proof.* This result follows by combining the upper bound from Proposition D.3 with the lower bound from Proposition D.4.
$\qquad\square$

*Remark* D.5 (Individual Sample Bound Discussion). We can derive the results based on individual sample approach (Bu et al., 2020), replacing the following inequality with (53), which is based on individual sample,

$$\left| \mathbb{E}_{P_H \otimes \mu}[\exp(\gamma \ell(H, \tilde{Z}))] - \mathbb{E}_{P_{H,S}}[\frac{1}{n} \sum_{i=1}^{n} \exp(\gamma \ell(H, Z_i))] \right|$$

$$\leq \begin{cases} \sqrt{2|\gamma|^{1+\epsilon} \kappa_t^{1+\epsilon} \frac{1}{n} \sum_{i=1}^{n} I(H; Z_i)} & \text{if } \max_{i \in [n]} I(H; Z_i) \leq \frac{2^{\epsilon}|\gamma|^{1+\epsilon}\kappa_t^{1+\epsilon}}{2} \\ \frac{1}{n} \sum_{i=1}^{n} I(H; Z_i) + \frac{2^{\epsilon}|\gamma|^{1+\epsilon}\kappa_t^{1+\epsilon}}{2} & \text{if } \min_{i \in [n]} I(H; Z_i) > \frac{2^{\epsilon}|\gamma|^{1+\epsilon}\kappa_t^{1+\epsilon}}{2}. \end{cases} \tag{69}$$

# E. Proof and details of Section 4

**Proposition 4.2** (**Restated**). *Given Assumption 4.1 and Assumption 3.1, then the difference of tilted population risk under,* (9)*, between $\mu$ and $\tilde{\mu}$ is bounded as follows for all $h \in \mathcal{H}$,*

$$\left| \frac{1}{\gamma} \log(\mathbb{E}_{\tilde{Z} \sim \mu}[\exp(\gamma \ell(h, \tilde{Z}))]) - \frac{1}{\gamma} \log(\mathbb{E}_{\tilde{Z} \sim \tilde{\mu}}[\exp(\gamma \ell(h, \tilde{Z}))]) \right| \leq \frac{\mathbb{TV}(\mu, \tilde{\mu})}{\gamma^2} \frac{\left(\exp(|\gamma|\kappa_u) - \exp(|\gamma|\kappa_s)\right)}{(\kappa_u - \kappa_s)}. \tag{70}$$

*Proof.* We have for a fixed $h \in \mathcal{H}$ that

$$\left| \frac{1}{\gamma} \log(\mathbb{E}_{\tilde{Z} \sim \mu}[\exp(\gamma \ell(h, \tilde{Z}))]) - \frac{1}{\gamma} \log(\mathbb{E}_{\tilde{Z} \sim \tilde{\mu}}[\exp(\gamma \ell(h, \tilde{Z}))]) \right|$$

$$\overset{(a)}{=} \left| \int_0^1 \int_{\mathcal{Z}} \frac{\exp(\gamma \ell(h, z))}{|\gamma| \mathbb{E}_{\tilde{Z} \sim \mu_\lambda}[\exp(\gamma \ell(h, \tilde{Z}))]} (\tilde{\mu} - \mu)(\mathrm{d}z) \mathrm{d}\lambda \right|$$

$$\overset{(b)}{\leq} \frac{\mathbb{TV}(\mu, \tilde{\mu}) \exp(|\gamma|\kappa_s)}{|\gamma|} \int_0^1 \exp(|\gamma|\lambda(\kappa_u - \kappa_s)) \mathrm{d}\lambda \tag{71}$$

$$= \frac{\mathbb{TV}(\mu, \tilde{\mu})}{|\gamma|} \frac{\exp(|\gamma|\kappa_u) - \exp(|\gamma|\kappa_s)}{|\gamma|(\kappa_u - \kappa_s)},$$

where (a) and (b) follow from the functional derivative with $\mu_\lambda = \tilde{\mu} + \lambda(\mu - \tilde{\mu})$ and Lemma C.16. The same approach can be applied for the lower bound.
$\qquad\square$

*Remark* E.1. For positive $\gamma$, the result in Proposition 4.2 does not hold and can be unbounded.

---

**Proposition E.2** (Upper Bound). *Given Assumptions 3.1 and 4.1, for any fixed $h \in \mathcal{H}$ and with probability least $(1 - \delta)$ for $\gamma < 0$, then the following upper bound holds on the tilted generalization error*

$$\mathrm{gen}_\gamma(h, \hat{S}) \leq \frac{2\exp(|\gamma|\kappa_s)}{|\gamma|} \sqrt{\frac{2^{\epsilon}|\gamma|^{1+\epsilon}\kappa_s^{1+\epsilon} \log(2/\delta)}{n}}$$

$$+ \frac{4\exp(|\gamma|\kappa_s)(\log(2/\delta))}{3n|\gamma|} + |\gamma|^{\epsilon}\kappa_u^{1+\epsilon} + \frac{\mathbb{TV}(\mu, \tilde{\mu})}{\gamma^2} \frac{\left(\exp(|\gamma|\kappa_u) - \exp(|\gamma|\kappa_s)\right)}{(\kappa_u - \kappa_s)},$$

*where $\hat{S}$ is the training dataset under the distributional shift.*

---

*Proof.* The proof follows directly from the following decomposition of the tilted generalization error under distribution shift,

$$\mathrm{gen}_\gamma(h, \hat{S}) = \underbrace{\mathrm{R}(h, \mu) - \mathrm{R}_\gamma(h, \mu^{\otimes n})}_{I_5} + \underbrace{\mathrm{R}_\gamma(h, \mu) - \mathrm{R}_\gamma(h, \tilde{\mu})}_{I_6} + \underbrace{\mathrm{R}_\gamma(h, \tilde{\mu}) - \widehat{\mathrm{R}}_\gamma(h, \hat{S})}_{I_7},$$

where $I_5$, $I_6$ and $I_7$ can be bounded using Lemma C.9, Proposition 4.2 and Proposition 3.3, respectively.
$\qquad\square$

**Proposition E.3** (Lower Bound). *Given Assumptions 3.1 and 4.1, for any fixed $h \in \mathcal{H}$ and with probability least $(1-\delta)$, there exists a $\zeta \in (0,1)$ such that for $n \geq \frac{(4\gamma^2 \kappa_u^2 + 8/3\zeta) \log(2/\delta)}{\zeta^2 \exp(2\gamma\kappa_u)}$ and $\gamma < 0$, such that the following upper bound holds on the tilted generalization error*

$$\mathrm{gen}_\gamma(h, \hat{S}) \geq -\frac{2\exp(|\gamma|\kappa_s)}{(1-\zeta)|\gamma|}\sqrt{\frac{2^\epsilon |\gamma|^{1+\epsilon}\kappa_s^{1+\epsilon}\log(2/\delta)}{n}} - \frac{4\exp(|\gamma|\kappa_s)(\log(2/\delta))}{3n|\gamma|(1-\zeta)}$$
$$-\frac{\mathbb{TV}(\mu, \tilde{\mu})}{\gamma^2}\frac{\big(\exp(|\gamma|\kappa_u) - \exp(|\gamma|\kappa_s)\big)}{(\kappa_u - \kappa_s)},$$

*where $\hat{S}$ is the training dataset under the distributional shift.*

*Proof.* The proof follows directly from the following decomposition of the tilted generalization error under distribution shift,

$$\mathrm{gen}_\gamma(h, \hat{S}) = \underbrace{\mathrm{R}(h, \mu) - \mathrm{R}_\gamma(h, \mu^{\otimes n})}_{I_5} + \underbrace{\mathrm{R}_\gamma(h, \mu) - \mathrm{R}_\gamma(h, \tilde{\mu})}_{I_6} + \underbrace{\mathrm{R}_\gamma(h, \tilde{\mu}) - \widehat{\mathrm{R}}_\gamma(h, \hat{S})}_{I_7},$$

where $I_5$, $I_6$ and $I_7$ can be bounded using Lemma C.9, Proposition 4.2 and Proposition 3.4, respectively. $\square$

**Theorem 4.3** (**Restated**). *Under the same assumptions in Theorem 3.5, then for $n \geq \frac{(4\gamma^2 \kappa_u^2 + 8/3\zeta)\log(2/\delta)}{\zeta^2 \exp(2\gamma\kappa_u)}$ and $\gamma < 0$, and with probability at least $(1-\delta)$, the absolute value of the tilted generalization error under distributional shift satisfies*

$$\sup_{h \in \mathcal{H}} |\mathrm{gen}_\gamma(h, \hat{S})| \tag{72}$$

$$\leq \frac{2\exp(|\gamma|\kappa_u)}{(1-\zeta)|\gamma|}\sqrt{\frac{2^\epsilon |\gamma|^{1+\epsilon}\kappa_u^{1+\epsilon}B(\delta)}{n}} + \frac{4\exp(|\gamma|\kappa_u)B(\delta)}{3n|\gamma|(1-\zeta)}$$
$$+ |\gamma|^\epsilon \kappa_u^{1+\epsilon} + \frac{\mathbb{TV}(\mu, \tilde{\mu})}{\gamma^2}\frac{\big(\exp(|\gamma|\kappa_u) - \exp(|\gamma|\kappa_s)\big)}{(\kappa_u - \kappa_s)},$$

*where $B(\delta) = \log(\mathrm{card}(\mathcal{H})) + \log(2/\delta)$.*

*Proof.* The result follows from combining the results of Proposition E.2, Proposition E.3, and applying the union bound. $\square$

**Theorem 4.5.** *Under the same assumptions as in Theorem 3.11 and Assumption 4.4, the following upper bounds on absolute expected tilted generalization error hold under distributional shift, if (a) or (b) hold,*

$$|\overline{\mathrm{gen}}_\gamma(H, \hat{S})| \leq \frac{\exp(|\gamma|\kappa_{st})}{|\gamma|(1-\zeta)}G\big(I(H; \hat{S})\big) + |\gamma|^\epsilon(\kappa_t^{1+\epsilon} + \kappa_{st}^{1+\epsilon})$$
$$+ \frac{\mathbb{TV}(\mu, \tilde{\mu})}{\gamma^2}\frac{\big(\exp(|\gamma|\kappa_t) - \exp(|\gamma|\kappa_{st})\big)}{(\kappa_t - \kappa_{st})},$$

*where if (a) and (b) hold we have $G(I(H; \hat{S})) = \sqrt{\frac{2\kappa_t^{\epsilon+1}|\gamma|^{1+\epsilon}I(H;\hat{S})}{n}}$ and if (c) holds, we have, $G(I(H; \hat{S})) = \frac{I(H;\hat{S})}{n} + \frac{2^\epsilon |\gamma|^{1+\epsilon}\kappa_t^{1+\epsilon}}{2}$.*

*Proof.* The proof follows from Theorem 3.11 and Assumption 4.4. Note that, we have,

$$0 \leq \mathrm{R}(H, P_H \otimes \mu) - \mathrm{R}_\gamma(H, P_H \otimes \mu) \leq |\gamma|^\epsilon \kappa_t^{1+\epsilon},$$
$$-|\gamma|^\epsilon \kappa_{st}^{1+\epsilon} \leq \mathrm{R}_\gamma(H, P_{H,\hat{S}}) - \mathbb{E}_{P_{H,\hat{S}}}[\widehat{\mathrm{R}}_\gamma(H, \hat{S})] \leq 0. \tag{73}$$

We also have in a similar approach to Proposition 4.2,

$$\left| \frac{1}{\gamma} \log(\mathbb{E}_{P_H \otimes \mu}[\exp(\gamma \ell(H, \tilde{Z}))]) - \frac{1}{\gamma} \log(\mathbb{E}_{P_H \otimes \tilde{\mu}}[\exp(\gamma \ell(H, \tilde{Z}))]) \right|$$

$$\leq \frac{\mathbb{TV}(P_H \otimes \mu, P_H \otimes \tilde{\mu})}{\gamma^2} \frac{\left( \exp(|\gamma| \kappa_t) - \exp(|\gamma| \kappa_{st}) \right)}{(\kappa_t - \kappa_{st})}. \tag{74}$$

$\square$

Note that $\mathbb{TV}(P_H \otimes \mu, P_H \otimes \tilde{\mu}) = \mathbb{TV}(\mu, \tilde{\mu})$. Combining Theorem 3.11 with (74) and (73) completes the proof.

### E.1. Convergence rate under distribution shift

In this section, we study the convergence rate under distribution shift. Suppose that we have some outlier samples in training dataset with distribution $\nu$. Then, we can assume that $\tilde{\mu} = (1 - \tau)\mu + \tau\nu$ for some $\tau \in [0, 1]$. We analysis the following term in Theorem 4.3,

$$\frac{\mathbb{TV}(\mu, \tilde{\mu})}{\gamma^2} \frac{\left( \exp(|\gamma| \kappa_u) - \exp(|\gamma| \kappa_s) \right)}{(\kappa_u - \kappa_s)}. \tag{75}$$

Using Taylor's expansion, we can show that $\frac{\left( \exp(|\gamma| \kappa_u) - \exp(|\gamma| \kappa_s) \right)}{(\kappa_u - \kappa_s)} = O(\gamma)$. Furthermore, we have,

$$\begin{aligned} \mathbb{TV}(\mu, \tilde{\mu}) &= \int_{\mathcal{Z}} |\mu - \tilde{\mu}| \mathrm{d}z \\ &= \tau \int_{\mathcal{Z}} |\mu - \nu| \mathrm{d}z \\ &\leq \tau \mathbb{TV}(\mu, \nu) \\ &\leq 2\tau. \end{aligned} \tag{76}$$

Therefore, we have,

$$\frac{\mathbb{TV}(\mu, \tilde{\mu})}{\gamma^2} \frac{\left( \exp(|\gamma| \kappa_u) - \exp(|\gamma| \kappa_s) \right)}{(\kappa_u - \kappa_s)} = O\left( \frac{\tau}{\gamma} \right). \tag{77}$$

In general, choosing $\gamma = O(n^{-1/(1+\epsilon)})$ for $n \geq \frac{(4\kappa_u^{1+\epsilon} + 8/3\zeta) \log(2/\delta)}{\zeta^2 \exp(-2\kappa_u)}$, we have the overall convergence rate on absolute titled generalization error, $\max\left( O\left( \tau n^{1/(1+\epsilon)} \right), O\left( n^{-\epsilon/(1+\epsilon)} \right) \right)$. Note that choosing $\tau = O(1/n)$, we can have the convergence rate of $O\left( n^{-\epsilon/(1+\epsilon)} \right)$. For example, we have one outlier sample, we can choose $\tau = \frac{1}{n+1}$.

## F. Proof and details of Section 6

**Proposition 6.1.** *The solution to the expected TERM regularized via KL divergence,* (24)*, is the tilted Gibbs Posterior (a.k.a. Gibbs Algorithm),*

$$P_{H|S}^\gamma := \frac{\pi_H}{F_\alpha(S)} \left( \frac{1}{n} \sum_{i=1}^n \exp(\gamma \ell(H, Z_i)) \right)^{-\alpha/\gamma}, \tag{78}$$

*where $F_\alpha(S)$ is a normalization factor.*

*Proof.* From (Zhang, 2006), we know that,

$$P_X^\star = \min_{P_X} \mathbb{E}_{P_X}[f(x)] + \frac{1}{\alpha} \mathrm{KL}(P_X \| Q_X), \tag{79}$$

where $P_X^\star = \frac{Q_X \exp(-\alpha f(X))}{\mathbb{E}_{Q_X}[\exp(-\alpha f(X))]}$. Using (79), it can be shown that the tilted Gibbs posterior is the solution to (25).

$\square$

**Proposition 6.2** (Restated). *The difference between the expected TER under the joint and product of marginal distributions of $H$ and $S$ can be characterized as,*

$$\overline{\mathrm{R}}_\gamma(H, P_H \otimes \mu) - \overline{\mathrm{R}}_\gamma(H, P_{H,S}) = \frac{I_{\mathrm{SKL}}(H; S)}{\alpha}. \tag{80}$$

*Proof.* As in Aminian et al. (2015), the symmetrized KL information between two random variables $(S, H)$ can be written as

$$I_{\mathrm{SKL}}(H; S) = \mathbb{E}_{P_H \otimes \mu^{\otimes n}}[\log(P_{H|S})] - \mathbb{E}_{P_{H,S}}[\log(P_{H|S})]. \tag{81}$$

The results follows by substituting the tilted Gibbs posterior in (81). □

**Theorem 6.3** (Restated). *Under the same Assumptions in Theorem 3.11 for some $\epsilon \in (0, 1]$, the expected tilted generalization error of the tilted Gibbs posterior satisfies*

$$0 \leq \overline{\mathrm{gen}}_\gamma(H, S) \leq \frac{2\alpha \exp(2|\gamma|\kappa_t)\kappa_t^{1+\epsilon}}{(1-\zeta)^2 n|\gamma|^{1-\epsilon}} + \frac{\exp(|\gamma|\kappa_t)\kappa_t^{1+\epsilon}|\gamma|^{1/2+\epsilon}}{(1-\zeta)|\gamma|}\sqrt{\frac{\alpha}{n}} + 2|\gamma|^\epsilon \kappa_t^{1+\epsilon}. \tag{82}$$

*Proof.* We expand

$$\begin{aligned}
\overline{\mathrm{gen}}_\gamma(H, S) = {} & \mathrm{R}(H, P_H \otimes \mu) - \overline{\mathrm{R}}_\gamma(H, P_H \otimes \mu^{\otimes n}) \\
& + \overline{\mathrm{R}}_\gamma(H, P_H \otimes \mu^{\otimes n}) - \mathrm{R}_\gamma(H, P_H \otimes \mu) \\
& + \mathrm{R}_\gamma(H, P_H \otimes \mu) - \mathrm{R}_\gamma(H, P_{H,S}) \\
& + \mathrm{R}_\gamma(H, P_{H,S}) - \overline{\mathrm{R}}_\gamma(H, P_{H,S}).
\end{aligned} \tag{83}$$

From Proposition 6.2, we have,

$$\overline{\mathrm{R}}_\gamma(H, P_H \otimes \mu) - \overline{\mathrm{R}}_\gamma(H, P_{H,S}) = \frac{I_{\mathrm{SKL}}(H; S)}{\alpha}. \tag{84}$$

We also have,

$$\begin{aligned}
0 \leq \mathrm{R}(H, P_H \otimes \mu) - \overline{\mathrm{R}}_\gamma(H, P_H \otimes \mu^{\otimes n}) &\leq \frac{|\gamma|^\epsilon \kappa_u^{(1+\epsilon)}}{2}, \\
0 \leq \overline{\mathrm{R}}_\gamma(H, P_H \otimes \mu^{\otimes n}) - \mathrm{R}_\gamma(H, P_H \otimes \mu) &\leq |\gamma|^\epsilon \kappa_u^{(1+\epsilon)}, \\
-|\gamma|^\epsilon \kappa_u^{(1+\epsilon)} \leq \mathrm{R}_\gamma(H, P_{H,S}) - \overline{\mathrm{R}}_\gamma(H, P_{H,S}) &\leq 0.
\end{aligned} \tag{85}$$

From Theorem 3.11, there exist $\zeta \in (0,1)$ such the following upper bound on absolute value of non-linear expected generalization error holds provided that $1 < \frac{2^\epsilon |\gamma|^{1+\epsilon} \kappa_t^{1+\epsilon} \exp(|\gamma|\kappa_t)}{2}$ and $n \geq \frac{2^\epsilon |\gamma|^{1+\epsilon} \kappa_t^{1+\epsilon} I(H;S)}{\zeta^2 \exp(2\gamma\kappa_t)}$,

$$\left| \mathrm{R}_\gamma(H, P_H \otimes \mu) - \mathrm{R}_\gamma(H, P_{H,S}) \right| \leq \frac{\exp(|\gamma|\kappa_t)}{(1-\zeta)|\gamma|}\sqrt{\frac{2\kappa_t^{1+\epsilon}|\gamma|^{1+\epsilon}I(H;S)}{n}} + |\gamma|^\epsilon \kappa_t^{1+\epsilon}. \tag{86}$$

Using the fact that $I(H; S) \leq I_{\mathrm{SKL}}(H; S)$, we have the following inequality by considering (84) and (86).

$$\frac{I(H; S)}{\alpha} \leq \frac{\exp(|\gamma|\kappa_t)}{(1-\zeta)|\gamma|}\sqrt{\frac{2\kappa_t^{1+\epsilon}|\gamma|^{1+\epsilon}I(H;S)}{n}} + |\gamma|^\epsilon \kappa_t^{1+\epsilon}, \tag{87}$$

where results in $\sqrt{I(H;S)} \leq \frac{A + \sqrt{A^2 + 4B}}{2} \leq A + \sqrt{B}$, where $A = \frac{\alpha \exp(|\gamma|\kappa_t)}{(1-\zeta)|\gamma|}\sqrt{\frac{2\kappa_t^{1+\epsilon}|\gamma|^{1+\epsilon}}{n}}$ and $B = \alpha|\gamma|^\epsilon \kappa_t^{1+\epsilon}$. We have,

$$\sqrt{I(H;S)} \leq \frac{\alpha \exp(|\gamma|\kappa_t)}{(1-\zeta)|\gamma|}\sqrt{\frac{2\kappa_t^{1+\epsilon}|\gamma|^{1+\epsilon}}{n}} + \sqrt{\frac{\alpha|\gamma|^\epsilon \kappa_t^{1+\epsilon}}{\epsilon}},$$

where $C$ is constant independent from $n$. Therefore,

$$\left|\mathrm{R}_\gamma(H, P_H \otimes \mu) - \mathrm{R}_\gamma(H, P_{H,S})\right| \leq \frac{2\alpha \exp(2|\gamma|\kappa_t)\kappa_t^{1+\epsilon}}{(1-\zeta)^2 n|\gamma|^{1-\epsilon}} + \frac{\exp(|\gamma|\kappa_t)\kappa_t^{1+\epsilon}|\gamma|^{1/2+\epsilon}}{(1-\zeta)|\gamma|}\sqrt{\frac{2\alpha}{n}} + 2|\gamma|^\epsilon \kappa_t^{1+\epsilon}. \tag{88}$$

Combining (88) with (85) completes the proof. $\qquad\square$

## G. Experiment Details

**Data-Driven choice of Tilt:** A similar discussion in Section 4 applies to the upper bound on the expected tilted generalization error under distribution shift in Theorem 4.5. However, in the large $n$ regime, the results remain the same.

**Logistic Regression:** For logistic regression, we consider 500 samples from 2d Gaussian distributions, $\mathcal{N}(3,1) \times \mathcal{N}(1,1)$ and $\mathcal{N}(-10,1) \times \mathcal{N}(-5,1)$. For logistic regression, we consider 0.01 as learning rate and 10000 iterations. Our loss function is $\ell(h,z) = \log(1 + \exp(-yh^T x))$ for $h \in \mathcal{H} \subset \mathbb{R}^2$.

- **Gaussian Outlier:** We add outlier samples $\rho \times 1000$ from Gaussian distribution $\mathcal{N}(-40,5) \times \mathcal{N}(-40,5)$.

- **Pareto Outlier:** Note that for $Z \sim \mathrm{Pareto}(1, \alpha)$ as a heavy-tailed distribution, we have $f_Z(z) = \frac{\alpha}{z^{\alpha+1}}$. We consider $\alpha = 1.5$ to have unbounded variance (heavy-tailed distribution). We add outlier samples $\rho \times 1000$ from Pareto distribution to Gaussian true sample dataset.

$\gamma_{\mathbf{data}}$ **computation:** suppose that we have $m$ samples as outlier, $\{\tilde{z}_j\}_{j=1}^m$. We model the distribution shift as follows,

$$\tilde{\mu} = \frac{n}{n+m}\mu + \frac{m}{n+m}\left(\sum_{j=1}^m \frac{P(Z = \tilde{z}_j)}{\sum_{k=1}^m P(Z = \tilde{z}_k)}\mathcal{N}(\tilde{z}_j, 0.01)\right), \tag{89}$$

where $P(Z = \tilde{z}_j)$ is the probability of $j$-th outlier data sampled from Gaussian or Pareto distribution. Then, we have,

$$\begin{aligned}
\mathbb{TV}(\mu, \tilde{\mu}) &= \int_{\mathcal{Z}} |\mu - \tilde{\mu}|\mathrm{d}z \\
&\leq \frac{m}{n+m}\sum_{j=1}^m \frac{P(Z = \tilde{z}_j)}{\sum_{k=1}^m P(Z = \tilde{z}_k)}\mathbb{TV}\big(\mu, \mathcal{N}(\tilde{z}_j, 0.01)\big) \\
&\leq \frac{2m}{m+n},
\end{aligned} \tag{90}$$

where $\mu$ is Gaussian distribution as data generating distribution. We calculate the empirical values of $\kappa_u$ using the true dataset and $\kappa_s$ using the training dataset containing outliers. Then, we consider the Hypothesis space as the set of parameters where the $(1 + \epsilon)$-th moment of loss is bounded by empirical $\kappa_s^{1+\epsilon}$. For Gaussian, we consider $\epsilon = 1$ and for Pareto distribution with $\alpha = 1.5$ we consider $\epsilon = 0.5$. The Logistic regression scenario under Gaussian and Pareto outliers with 10 and 5 number of samples, respectively, are shown in Fig. 1.

**Logistic Regression without outlier:** Similar to the distribution shift scenario, we can propose a data-driven $\gamma$ inspired by Corollary D.1. For this purpose, we assume that $n$ is large enough and $\gamma$ is a small negative value close to zero. Additionally, we assume that the first term in the upper bound of Corollary D.1 is zero due to large $n$. For simplicity in computation, we consider $\exp(-\gamma\kappa_u) \approx 1$ for small $\gamma$. Under these assumptions, we define:

$$\gamma_{\mathrm{data}} := \underset{\gamma \in (-\infty, 0)}{\arg\min} \left[\frac{|\gamma|}{2}\kappa_u^2 + \frac{4B(\delta)}{3n|\gamma|(1-\zeta)}\right], \tag{91}$$

where $B(\delta)$ is defined in Corollary D.1. Note that the term $B(\delta)$ can be large due to the cardinality of the hypothesis space, resulting in a very small $\gamma$. In this scenario, we expect that a $\gamma$ near zero would have a better performance.

Now, we conduct an experiment for a logistic regression problem by sampling from a Pareto distribution as a heavy-tailed distribution without outlier. We observed that TERM achieves better population risk with an optimal negative

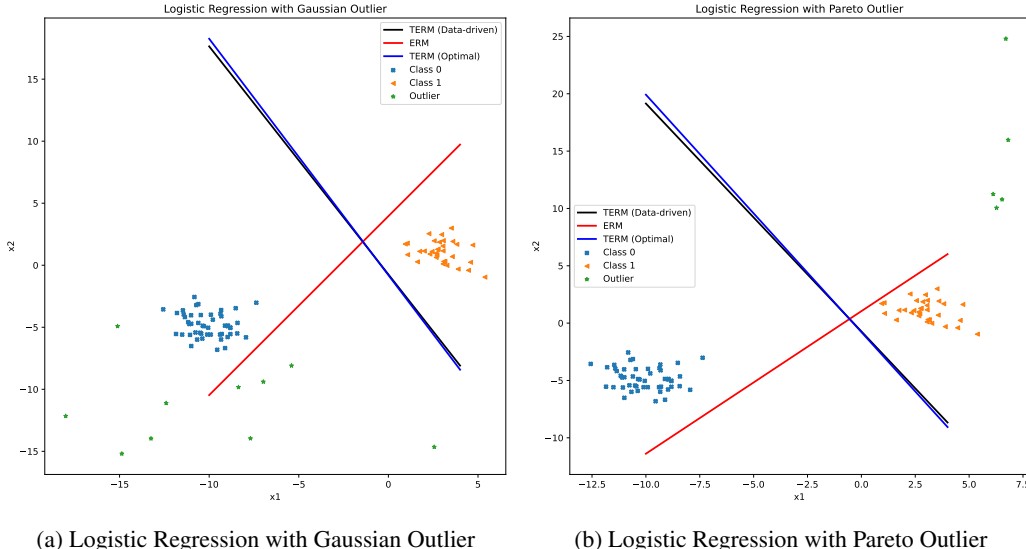

(a) Logistic Regression with Gaussian Outlier      (b) Logistic Regression with Pareto Outlier

Figure 1: Logistic Regression Experiments with Outlier

tilt near zero. In this scenario, the data-driven $\gamma$ performed similarly to the ERM solution. However, the optimal solution based on grid search outperformed the ERM solution. The experiment was conducted with 1,000 training samples and we have $\gamma^\star = -0.5263 \pm 0.0001$, $\mathrm{R}(h_{\gamma^\star}(\hat{S}), \mu) = 0.05220 \pm 0.00001$, $\mathrm{R}(h_{\mathrm{ERM}}(\hat{S}), \mu) = 0.05249 \pm 0.00002$, $\gamma_{\mathrm{data}} = -0.0001 \pm 0.0000$ and $\mathrm{R}(h_{\gamma_{\mathrm{data}}}(\hat{S}), \mu) = 0.05249 \pm 0.00001$.

**Linear Regression:** We also consider the Linear Regression toy example as mentioned in (Li et al., 2021) where the loss function is square loss function $\ell(h, z) = \frac{1}{2}(h^T x - y)$ for $h \in \mathcal{H} \subset \mathbb{R}^2$. Similar to logistic regression, we consider two scenario. One, the outlier is Gaussian, and another case where the outlier is Pareto, where we consider $n = 2000$. For true data, we consider $X \sim \mathcal{N}(1, 0.5)$ and $y = -x + 0.5 + \mathcal{N}(0, 0.5)$. For outlier, we consider

- Gaussian outlier: $X_{\mathrm{outlier}} \sim \mathcal{N}(-10, 0.5)$ and $y_{\mathrm{outlier}} \sim \mathcal{N}(2, 0.5)$,

- Pareto outlier: $X_{\mathrm{outlier}} \sim 2\mathcal{P}(1.5, 1)$ and $y_{\mathrm{outlier}} \sim \mathcal{P}(1.5, 1)$, where $\mathcal{P}(1.5, 1)$ is Pareto distribution with shape and scale equal to $1.5$ and $1$, respectively.

The results are reported in Table 5 and Table 4. The linear regression scenario under Gaussian and Pareto outliers with 10 number of samples are shown in Fig. 2.

Table 4: Linear Regression with Pareto Outliers: Results averaged over three runs ($n = 2,000$ samples), showing mean $\pm$ standard deviation.

| $\rho$ | $\gamma^\star$ | $\mathrm{R}(h_{\gamma^\star}(\hat{S}), \mu)$ | $\mathrm{R}(h_{\mathrm{ERM}}(\hat{S}), \mu)$ | $\gamma_{\mathrm{data}}$ | $\mathrm{R}(h_{\gamma_{\mathrm{data}}}(\hat{S}), \mu)$ |
|---|---|---|---|---|---|
| $0.049\%$ | $-0.26667 \pm 0.0555$ | $0.12530 \pm 0.0001$ | $0.25795 \pm 0.0001$ | $-0.1000 \pm 0.0000$ | $0.12530 \pm 0.0002$ |
| $9.13\%$ | $-0.7000 \pm 0.0000$ | $0.12183 \pm 0.001$ | $0.6731 \pm 0.0045$ | $-0.3333 \pm 0.04222$ | $0.1700 \pm 0.0005$ |
| $16.70\%$ | $-1.43333 \pm 0.1155$ | $0.1238 \pm 0.0002$ | $0.7580 \pm 0.0006$ | $-0.5333 \pm 0.01555$ | $0.2755 \pm 0.0001$ |
| $33.35\%$ | $-1.8000 \pm 0.0200$ | $0.1275 \pm 0.0001$ | $1.5096 \pm 0.0050$ | $-0.2333 \pm 0.0022$ | $0.3972 \pm 0.0002$ |
| $44.45\%$ | $-2.43333 \pm 0.0088$ | $0.1229 \pm 0.0001$ | $2.2190 \pm 0.0011$ | $-0.1000 \pm 0.0000$ | $0.4908 \pm 0.0001$ |

**Linear Regression without outlier:** In this scenario, we consider linear regression based on training samples from a Pareto distribution with shift parameter equal to 1.5 . We observe that for $\gamma^\star = -0.1 \pm 0.00001$, $\mathrm{R}(h_{\gamma^\star}(\hat{S}), \mu) = 10.4924 \pm 6.7273$, $\mathrm{R}(h_{\mathrm{ERM}}(\hat{S}), \mu) = 20.73892 \pm 12.9752$ and $\gamma_{\mathrm{data}} = -0.005 \pm 0.0000$ and $\mathrm{R}(h_{\gamma_{\mathrm{data}}}(\hat{S}), \mu) = 19.05249 \pm 10.4675$.

Table 5: Linear Regression with Gaussian Outliers: Results averaged over three runs ($n = 2,000$ samples), showing mean $\pm$ standard deviation.

| $\rho$ | $\gamma^\star$ | $\mathrm{R}(h_{\gamma^\star}(\hat{S}), \mu)$ | $\mathrm{R}(h_{\mathrm{ERM}}(\hat{S}), \mu)$ | $\gamma_{\mathrm{data}}$ | $\mathrm{R}(h_{\gamma_{\mathrm{data}}}(\hat{S}), \mu)$ |
|---|---|---|---|---|---|
| 0.049% | $-10.0000 \pm 0.0000$ | $0.7555 \pm 0.0000$ | $1.19923 \pm 0.0001$ | $-3.6333 \pm 0.0288$ | $0.75617 \pm 0.0001$ |
| 9.13% | $-0.1000 \pm 0.0000$ | $0.16095 \pm 0.0001$ | $0.3215 \pm 0.0005$ | $-3.8000 \pm 0.3466$ | $0.17083 \pm 0.0009$ |
| 16.70% | $-3.4000 \pm 0.1000$ | $0.15515 \pm 0.0000$ | $0.31122 \pm 0.0002$ | $-3.9000 \pm 0.0001$ | $0.15796 \pm 0.0001$ |
| 33.35% | $-0.1000 \pm 0.0200$ | $0.1564 \pm 0.0000$ | $0.3127 \pm 0.0001$ | $-5.0666 \pm 0.0022$ | $0.15858 \pm 0.0000$ |
| 44.45% | $-3.43333 \pm 0.0001$ | $0.1530 \pm 0.0001$ | $0.3070 \pm 0.0001$ | $-5.8000 \pm 0.0080$ | $0.15399 \pm 0.0001$ |

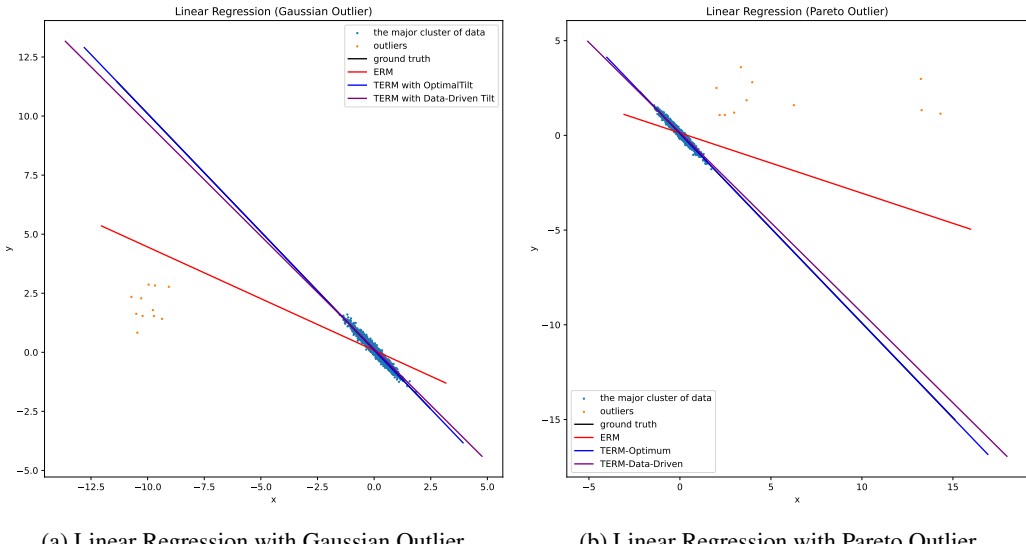

(a) Linear Regression with Gaussian Outlier    (b) Linear Regression with Pareto Outlier

Figure 2: Linear Regression Experiments with Outlier

# H. Generalization Bounds for Bounded Loss Functions

Upper bounds under linear empirical risk for bounded loss functions via information theoretic and uniform bounds are studied by Shalev-Shwartz & Ben-David (2014) and Xu & Mannor (2012), respectively. Inspired by these works, in this section, we provide upper bounds on the tilted generalization error via uniform and information-theoretic approaches for bounded loss functions with the convergence rate of $O(1/\sqrt{n})$ which is similar to generalization error under linear empirical risk. The following results are derived for bounded loss function scenario,

- Uniform bounds

- Information-theoretical bounds

- KL-regularized TERM under bounded loss function

- Rademacher Complexity bounds

- Stability bounds

- PAC-Bayesian bounds

The theoretical analysis in this section hinges on two fundamental properties: the Lipschitz continuity of the logarithmic function on a bounded interval and the boundedness of the loss function. When working with bounded loss functions, we can analyze the positive tilt scenario. Furthermore, the bounded case does not restrict us on sample size $n$ that would otherwise be necessary in the unbounded scenario.

### H.1. Uniform bounds for bounded loss

In this section the following assumption is made.

**Assumption H.1** (Bounded loss function). There is a constant $M$ such that the loss function, $(h, z) \mapsto \ell(h, z)$ satisfies $0 \leq \ell(h, z) \leq M$ uniformly for all $h \in \mathcal{H}, z \in \mathcal{Z}$.

For uniform bounds of the type (6), we decompose the tilted generalization error (4) as follows,

$$\text{gen}_\gamma(h, S) = \underbrace{\text{R}(h, \mu) - \text{R}_\gamma(h, \mu^{\otimes n})}_{I_1} + \underbrace{\text{R}_\gamma(h, \mu^{\otimes n}) - \widehat{\text{R}}_\gamma(h, S)}_{I_2}, \tag{92}$$

where $I_1$ is the difference between the population risk and the tilted population risk, and $I_2$ is the non-linear generalization error.

We first derive an upper bound on term $I_1$ in the following Proposition.

**Proposition H.2.** *Under Assumption H.1, for $\gamma \in \mathbb{R}$, the difference between the population risk and the tilted population risk satisfies*

$$\frac{-1}{2\gamma}\text{Var}\big(\exp(\gamma\ell(h, Z))\big) \leq \text{R}(h, \mu) - \text{R}_\gamma(h, \mu^{\otimes n}) \leq \frac{-\exp(-2\gamma M)}{2\gamma}\text{Var}\big(\exp(\gamma\ell(h, Z))\big). \tag{93}$$

*Proof.* For any $h \in \mathcal{H}$ we have

$$
\begin{aligned}
\text{R}(h, \mu) &= \mathbb{E}[\ell(h, Z)] \\
&= \mathbb{E}\Big[\frac{1}{\gamma}\log\big(\exp(\gamma\ell(h, Z))\big)\Big] \\
&= \frac{1}{|\gamma|}\Big[\mathbb{E}_{Z\sim\mu}\Big[-\log\big(\exp(\gamma\ell(h, Z))\big) - \frac{\exp(-2\gamma M)}{2}\exp(2\gamma\ell(h, Z)) \\
&\quad + \frac{\exp(-2\gamma M)}{2}\exp(2\gamma\ell(h, Z))\Big]\Big] \\
&\leq \frac{1}{\gamma}\log(\mathbb{E}[\exp(\gamma\ell(h, \mu))]) + \frac{\exp(-2\gamma M)}{2|\gamma|}\text{Var}(\exp(\gamma\ell(h, Z))) \\
&= \text{R}_\gamma(h, \mu^{\otimes n}) + \frac{\exp(-2\gamma M)}{2|\gamma|}\text{Var}(\exp(\gamma\ell(h, Z))).
\end{aligned} \tag{94}
$$

A similar approach can be applied for $\gamma > 0$ by using Lemma C.4 and the final result holds. $\qquad\square$

Note that for $\gamma \to 0$, the upper and lower bounds in Proposition H.2 are zero. As the log function is Lipschitz on a bounded interval, applying the Hoeffding inequality to term $I_2$ and Proposition H.2 to term $I_1$ in (13), we obtain the following upper bound on the tilted generalization error.

**Theorem H.3.** *Given Assumption H.1, for any fixed $h \in \mathcal{H}$ with probability at least $(1 - \delta)$ the tilted generalization error satisfies the upper bound,*

$$\text{gen}_\gamma(h, S) \leq \frac{-\exp(-2\gamma M)}{2\gamma}\text{Var}\big(\exp(\gamma\ell(h, Z))\big) + \frac{\big(\exp(|\gamma|M) - 1\big)}{|\gamma|}\sqrt{\frac{\log(2/\delta)}{2n}}. \tag{95}$$

*Proof.* We can apply the Proposition H.2 to provide an upper bound on term $I_1$. Regarding the term $I_5$, we have for $\gamma < 0$

$$
\begin{aligned}
& R_\gamma(h, \mu^{\otimes n}) - \widehat{R}_\gamma(h, S) \\
&= \frac{1}{\gamma} \log(\mathbb{E}_{\mu^{\otimes n}}[\frac{1}{n} \sum_{i=1}^n \exp(\gamma \ell(h, Z_i))]) - \frac{1}{\gamma} \log(\frac{1}{n} \sum_{i=1}^n \exp(\gamma \ell(h, Z_i))) \\
&\leq \frac{\exp(-\gamma M)}{|\gamma|} \big| \mathbb{E}_{\tilde{Z} \sim \mu}[\exp(\gamma \ell(h, \tilde{Z}))] - \frac{1}{n} \sum_{i=1}^n \exp(\gamma \ell(h, Z_i)) \big| \\
&\leq \frac{\exp(-\gamma M)(1 - \exp(\gamma M))}{|\gamma|} \sqrt{\frac{\log(2/\delta)}{2n}}.
\end{aligned}
\tag{96}
$$

Similarly, for $\gamma > 0$, we have

$$
R_\gamma(h, \mu^{\otimes n}) - \widehat{R}_\gamma(h, S) \leq \frac{(\exp(\gamma M) - 1)}{|\gamma|} \sqrt{\frac{\log(2/\delta)}{2n}}.
\tag{97}
$$

Combining this bound with Proposition H.2 completes the proof. □

**Theorem H.4.** *Under the same assumptions of Theorem H.3, for a fixed $h \in \mathcal{H}$, with probability at least $(1 - \delta)$, the tilted generalization error satisfies the lower bound*

$$
\mathrm{gen}_\gamma(h, S) \geq \frac{-1}{2\gamma} \mathrm{Var}\big(\exp(\gamma \ell(h, Z))\big) - \frac{(\exp(|\gamma| M) - 1)}{|\gamma|} \sqrt{\frac{\log(2/\delta)}{2n}}.
\tag{98}
$$

*Proof.* The proof is similar to that of Theorem H.3, by using the lower bound in Proposition H.2. □

Combining Theorem H.3 and Theorem H.4, we derive an upper bound on the absolute value of the titled generalization error.

**Corollary H.5.** *Let $A(\gamma) = (1 - \exp(\gamma M))^2$. Under the same assumptions in Theorem H.3, with probability at least $(1 - \delta)$, and a finite hypothesis space, the absolute value of the titled generalization error satisfies*

$$
\sup_{h \in \mathcal{H}} |\mathrm{gen}_\gamma(h, S)| \leq \frac{(\exp(|\gamma| M) - 1)}{|\gamma|} \sqrt{\frac{\log(\mathrm{card}(\mathcal{H})) + \log(2/\delta)}{2n}} + \frac{\max(1, \exp(-2\gamma M)) A(\gamma)}{8|\gamma|},
$$

*where $A(\gamma) = (1 - \exp(\gamma M))^2$.*

*Proof.* We can derive the following upper bound on the absolute of tilted generalization error by combining Theorem H.3 and Theorem H.4 for any fixed $h \in \mathcal{H}$

$$
|\mathrm{gen}_\gamma(h, S)| \leq \frac{(\exp(|\gamma| M) - 1)}{|\gamma|} \sqrt{\frac{\log(\mathrm{card}(\mathcal{H})) + \log(2/\delta)}{2n}} + \frac{\max(1, \exp(-2\gamma M)) A(\gamma)}{8|\gamma|},
\tag{99}
$$

where $A(\gamma) = (1 - \exp(\gamma M))^2$. Then, the final result follow by applying the uniform bound for all $h \in \mathcal{H}$ using (99). □

**Corollary H.6.** *Under the same Assumptions as in Theorem H.3 and assuming $\gamma$ is of order $O(n^{-\beta})$ for $\beta > 0$, the upper bound on the tilted generalization error in Theorem H.3 has a convergence rate of $\max\big(O(1/\sqrt{n}), O(n^{-\beta})\big)$ as $n \to \infty$.*

*Proof.* Using the inequality $\frac{x}{x+1} \leq \log(1 + x) \leq x$ and Taylor expansion for the exponential function,

$$
\exp(|\gamma| M) = 1 + |\gamma| M + \frac{|\gamma|^2 M^2}{2} + \frac{|\gamma|^3 M^3}{6} + O(\gamma^4),
\tag{100}
$$

it follows that

$$\frac{\big(\exp(|\gamma|M)-1\big)}{|\gamma|} \approx M + \frac{|\gamma|M^2}{2} + \frac{|\gamma|^2 M^3}{6} + O(\gamma^3). \tag{101}$$

This results in a convergence rate of $O(1/\sqrt{n})$ for $\frac{\big(\exp(|\gamma|M)-1\big)}{|\gamma|}\sqrt{\frac{\log(\mathrm{card}(\mathcal{H}))+\log(2/\delta)}{2n}}$ under $\gamma \to 0$.

For the term $\frac{\max(1,\exp(-2\gamma M))(1-\exp(\gamma M))^2}{8|\gamma|}$, using Taylor expansion, we have the convergence rate of $O(|\gamma|)$ for the first term; this completes the proof. $\qquad\square$

*Remark* H.7. Choosing $\beta \geq 1/2$ in Corollary H.6 gives a convergence rate of $O(1/\sqrt{n})$ for the tilted generalization error.

*Remark* H.8 (The influence of $\gamma$). As $\gamma \to 0$, the upper bound in Corollary H.5 on the absolute value of tilted generalization error converges to the upper bound on absolute value of the generalization error under the ERM algorithm obtained by Shalev-Shwartz & Ben-David (2014),

$$\sup_{h\in\mathcal{H}} |\mathrm{gen}(h,S)| \leq M\sqrt{\frac{\log(\mathrm{card}(\mathcal{H}))+\log(2/\delta)}{2n}}. \tag{102}$$

In particular, $\big(\exp(|\gamma|M)-1\big)/|\gamma| \to M$ and the first term in Corollary H.5 vanishes. Therefore, the upper bound converges to a uniform bound on the linear empirical risk.

Using Corollary H.6, we derive an upper bound on the excess risk.

> **Corollary H.9.** *Under the same assumptions in Theorem H.3, and a finite hypothesis space, with probability at least $(1-\delta)$, the excess risk of tilted empirical risk satisfies*
> $$\mathfrak{E}_\gamma(\mu) \leq \frac{2\big(\exp(|\gamma|M)-1\big)}{|\gamma|}\sqrt{\frac{\log(\mathrm{card}(\mathcal{H}))+\log(2/\delta)}{2n}} + \frac{2\max(1,\exp(-2\gamma M))A(\gamma)}{8|\gamma|},$$
> *where $A(\gamma) = (1-\exp(\gamma M))^2$.*

*Proof.* It can be proved that,
$$\mathfrak{E}_\gamma(\mu) \leq 2\sup_{h\in\mathcal{H}} |\mathrm{gen}_\gamma(h,S)|$$

and
$$\mathrm{R}(h_\gamma^*(S),\mu) \leq \hat{\mathrm{R}}_\gamma(h_\gamma^*(S),\mu) + U \quad \leq \hat{\mathrm{R}}_\gamma(h^*(\mu),\mu) + U \;\leq \mathrm{R}(h^*(\mu),\mu) + 2U,$$

where $U = \sup_{h\in\mathcal{H}} |\mathrm{R}(h,\mu) - \hat{\mathrm{R}}_\gamma(h,S)| = \sup_{h\in\mathcal{H}} |\mathrm{gen}_\gamma(h,S)|$.

Note that $\sup_{h\in\mathcal{H}} |\mathrm{gen}_\gamma(h,S)|$ can be bounded using Corollary H.5. $\qquad\square$

## H.2. Information-theoretic bounds for bounded loss functions

Next, we provide an upper bound on the expected tilted generalization error. For information-theoretic bounds, we employ the following decomposition of the expected tilted generalization error,

$$\overline{\mathrm{gen}}_\gamma(H,S) = \{\mathrm{R}(H,P_H\otimes\mu) - \overline{\mathrm{R}}_\gamma(H,P_H\otimes\mu^{\otimes n})\} + \{\overline{\mathrm{R}}_\gamma(H,P_H\otimes\mu^{\otimes n}) - \overline{\mathrm{R}}_\gamma(H,P_{H,S})\}. \tag{103}$$

The following is helpful in deriving the upper bound.

**Proposition H.10.** *Under Assumption H.1, the following inequality holds,*

$$\left|\overline{\mathrm{R}}_\gamma(H,P_H\otimes\mu^{\otimes n}) - \overline{\mathrm{R}}_\gamma(H,P_{H,S})\right| \leq \frac{(\exp(|\gamma|M)-1)}{|\gamma|}\sqrt{\frac{I(H;S)}{2n}}.$$

*Proof.* The proof follows directly from applying Lemma C.15 to the $\log(.)$ function and then applying Lemma C.14. $\qquad\square$

Using Proposition H.10, we derive the following upper and lower bounds on the expected tilted generalization error.

**Theorem H.11.** *Under Assumption H.1, the expected tilted generalization error satisfies*

$$\overline{\mathrm{gen}}_\gamma(H, S) \leq \frac{(\exp(|\gamma|M) - 1)}{|\gamma|} \sqrt{\frac{I(H; S)}{2n}} - \frac{\gamma \exp(-\gamma M)}{2}\Big(1 - \frac{1}{n}\Big) \mathbb{E}_{P_H}\big[\mathrm{Var}_{\tilde{Z} \sim \mu}(\ell(H, \tilde{Z}))\big].$$

*Proof.* We expand

$$\overline{\mathrm{gen}}_\gamma(H, S) = \mathrm{R}(H, P_H \otimes \mu) - \overline{\mathrm{R}}_\gamma(H, P_H \otimes \mu^{\otimes n}) + \overline{\mathrm{R}}_\gamma(H, P_H \otimes \mu^{\otimes n}) - \overline{\mathrm{R}}_\gamma(H, P_{H,S}). \tag{104}$$

Using Proposition H.10, it follows that

$$|\overline{\mathrm{R}}_\gamma(H, P_H \otimes \mu^{\otimes n}) - \overline{\mathrm{R}}_\gamma(H, P_{H,S})| \leq \frac{(\exp(|\gamma|M) - 1)}{|\gamma|} \sqrt{\frac{I(H; S)}{2n}}. \tag{105}$$

Using the Lipschitz property of the $\log(.)$ function under Assumption H.1, we have for $\gamma > 0$,

$$
\begin{aligned}
&\mathrm{R}(H, P_H \otimes \mu) - \overline{\mathrm{R}}_\gamma(H, P_H \otimes \mu^{\otimes n}) \\
&= \mathbb{E}_{P_H \otimes \mu^{\otimes n}}\Big[\frac{1}{\gamma} \log(\exp(\frac{\gamma}{n} \sum_{i=1}^n \ell(H, Z_i)))\Big] - \mathbb{E}_{P_H \otimes \mu^{\otimes n}}\Big[\frac{1}{\gamma} \log(\frac{1}{n} \sum_{i=1}^n \exp(\gamma \ell(H, Z_i)))\Big] \\
&\leq \frac{\exp(-\gamma M)}{\gamma} \mathbb{E}_{P_H \otimes \mu^{\otimes n}}\Big[\exp(\frac{\gamma}{n} \sum_{i=1}^n \ell(H, Z_i)) - \frac{1}{n} \sum_{i=1}^n \exp(\gamma \ell(H, Z_i))\Big] \\
&\leq \frac{-\exp(-\gamma M)}{2\gamma} \mathbb{E}_{P_H \otimes \mu^{\otimes n}}\Big[\Big(\frac{1}{n} \sum_{i=1}^n \gamma^2 \ell(H, Z_i)^2\Big) - \Big(\frac{1}{n^2}\big(\sum_{i=1}^n \gamma \ell(H, Z_i)\big)^2\Big)\Big] \\
&= \frac{-\exp(-\gamma M)}{2\gamma}(1 - 1/n)\mathbb{E}_{P_H}\big[\mathrm{Var}_{\tilde{Z} \sim \mu}(\gamma \ell(H, \tilde{Z}))\big] \\
&= \frac{-\exp(-\gamma M)\gamma}{2}(1 - 1/n)\mathbb{E}_{P_H}\big[\mathrm{Var}_{\tilde{Z} \sim \mu}(\ell(H, \tilde{Z}))\big]
\end{aligned}
\tag{106}
$$

where $\tilde{Z} \sim \mu$. A similar results also holds for $\gamma < 0$. Combining (105), (106) with (104) completes the proof. $\square$

We now give a lower bound via the information-theoretic approach.

**Theorem H.12.** *Under the same assumptions in Theorem H.11, the expected tilted generalization error satisfies*

$$\overline{\mathrm{gen}}_\gamma(H, S) \geq -\frac{(\exp(|\gamma|M) - 1)}{|\gamma|} \sqrt{\frac{I(H; S)}{2n}} - \frac{\gamma \exp(\gamma M)}{2}\Big(1 - \frac{1}{n}\Big) \mathbb{E}_{P_H}\big[\mathrm{Var}_{\tilde{Z} \sim \mu}(\ell(H, \tilde{Z}))\big].$$

*Proof.* Similarly as in the proof of Theorem H.11, we can prove the lower bound. Using the Lipschitz property of the $\log(.)$

function under Assumption H.1, we have for $\gamma > 0$,

$$
\begin{aligned}
&\mathrm{R}(H, P_H \otimes \mu) - \overline{\mathrm{R}}_\gamma(H, P_H \otimes \mu^{\otimes n}) \\
&= \mathbb{E}_{P_H \otimes \mu^{\otimes n}}\Big[\frac{1}{\gamma}\log(\exp(\frac{\gamma}{n}\sum_{i=1}^n \ell(H, Z_i)))\Big] - \mathbb{E}_{P_H \otimes \mu^{\otimes n}}\Big[\frac{1}{\gamma}\log(\frac{1}{n}\sum_{i=1}^n \exp(\gamma\ell(H, Z_i)))\Big] \\
&\geq \frac{1}{\gamma}\mathbb{E}_{P_H \otimes \mu^{\otimes n}}\Big[\exp(\frac{\gamma}{n}\sum_{i=1}^n \ell(H, Z_i)) - \frac{1}{n}\sum_{i=1}^n \exp(\gamma\ell(H, Z_i))\Big] \\
&\geq \frac{-\exp(\gamma M)}{2\gamma}\mathbb{E}_{P_H \otimes \mu^{\otimes n}}\Big[\Big(\frac{1}{n}\sum_{i=1}^n \gamma^2\ell(H, Z_i)^2\Big) - \Big(\frac{1}{n^2}\big(\sum_{i=1}^n \gamma\ell(H, Z_i)\big)^2\Big)\Big] \\
&= \frac{-\exp(\gamma M)}{2\gamma}\big(1 - \frac{1}{n}\big)\mathbb{E}_{P_H}\big[\mathrm{Var}_{\tilde{Z}\sim\mu}(\gamma\ell(H, \tilde{Z}))\big] \\
&= \frac{-\gamma\exp(\gamma M)}{2}\big(1 - \frac{1}{n}\big)\mathbb{E}_{P_H}\big[\mathrm{Var}_{\tilde{Z}\sim\mu}(\ell(H, \tilde{Z}))\big].
\end{aligned}
\tag{107}
$$

Similar results also holds for $\gamma < 0$. Combining (105), (107) with (104) completes the proof. $\square$

Combining Theorem H.11 and Theorem H.12, we derive an upper bound on the absolute value of the expected tilted generalization error.

**Corollary H.13.** *Under the same assumptions in Theorem H.11, the absolute value of the expected titled generalization error satisfies*

$$
|\overline{\mathrm{gen}}_\gamma(H, S)| \leq \frac{(\exp(|\gamma|M) - 1)}{|\gamma|}\sqrt{\frac{I(H; S)}{2n}} + \frac{|\gamma|M^2\exp(|\gamma|M)}{8}\Big(1 - \frac{1}{n}\Big).
\tag{108}
$$

*Proof.* We can derive the upper bound on absolute value of the expected tilted generalization error by combining Theorem H.11 and Theorem H.12. $\square$

*Remark* H.14. In Corollary H.13, we observe that by choosing $\gamma = O(n^{-\beta})$, the overall convergence rate of the generalization error upper bound is $\max(O(1/\sqrt{n}), O(n^{-\beta}))$ for bounded $I(H; S)$. For $\beta \geq 1/2$, the convergence rate of (108) is the same as the convergence rate of the expected upper bound in (Xu & Raginsky, 2017). In addition, for $\gamma \to 0$, the upper bound in Corollary H.13 converges to the expected upper bound in (Xu & Raginsky, 2017).

Similar to unbounded loss function, the results in this section are non-vacuous for bounded $I(H; S)$. If this assumption is violated, we can apply the individual sample method (Bu et al., 2020), chaining methods (Asadi et al., 2018), or conditional mutual information frameworks (Steinke & Zakynthinou, 2020) to derive tighter upper bound for the tilted generalization error.

### H.2.1. INDIVIDUAL SAMPLE BOUND DISCUSSION

We can apply previous information-theoretic bounding techniques (e.g., (Bu et al., 2020), (Asadi et al., 2018), and (Steinke & Zakynthinou, 2020)), exploiting the Lipschitz property of the logarithm function (or Lemma C.10) over a bounded support. For example, to derive an upper bound based on the individual sample (Bu et al., 2020), using the approach for Proposition H.10 and Lemma C.10, we have for $\gamma > 0$, we have

$$
\begin{aligned}
\overline{\mathrm{R}}_\gamma(H, P_H \otimes \mu^{\otimes n}) - \overline{\mathrm{R}}_\gamma(H, P_{H,S}) &\leq \frac{1}{\gamma}\Big(\mathrm{E}_{P_H \otimes \mu^{\otimes n}}\big[\frac{1}{n}\sum_{i=1}^n \exp(\gamma\ell(H, Z_i))\big] - \mathrm{E}_{P_{H,S}}\big[\frac{1}{n}\sum_{i=1}^n \exp(\gamma\ell(H, Z_i))\big]\Big) \\
&\leq \frac{(\exp(\gamma M) - 1)}{\gamma}\sum_{i=1}^n \frac{1}{n}\sqrt{\frac{I(H; Z_i)}{2}},
\end{aligned}
\tag{109}
$$

where for the first equality, we applied the Lemma C.10 and for the second inequality, we use that $1 \leq \exp(\gamma\ell(H, Z_i)) \leq \exp(\gamma M)$ for $\gamma > 0$ and the approach in (Bu et al., 2020) for bounding via individual samples. A similar approach also can

be applied to $\gamma < 0$. Therefore, the following upper bound holds on the absolute value of the expected tilted generalization error,

$$|\overline{\text{gen}}_\gamma(H,S)| \leq \frac{(\exp(|\gamma|M)-1)}{|\gamma|n} \sum_{i=1}^{n} \sqrt{\frac{I(H;Z_i)}{2n}} + \frac{|\gamma|M^2 \exp(|\gamma|M)}{8}\left(1-\frac{1}{n}\right).$$

## H.3. The KL-Regularized TERM Problem

In this section, similar to unbounded case (Section 6), we study the KL-regularized under bounded loss function for both negative and positive tilts. We consider the same definitions in Section 6. We next provide a parametric upper bound on the tilted generalization error of the tilted Gibbs posterior under bounded loss function.

> **Theorem H.15.** *Under Assumption H.1, the expected tilted generalization error of the tilted Gibbs posterior satisfies,*
> $$\overline{\text{gen}}_\gamma(H,S) \leq \frac{\alpha(\exp(|\gamma|M)-1)^2}{2\gamma^2 n} + \frac{\text{Var}(\exp(\gamma\ell(H,\tilde{Z})))}{2\gamma}\left(1/n - \exp(-2\gamma M)\right). \tag{110}$$

*Proof.* Note that, we have

$$\begin{aligned}
\frac{I(H;S)}{\alpha} &\leq \frac{I_{\text{SKL}}(H;S)}{\alpha} \\
&= \overline{R}_\gamma(H, P_H \otimes \mu) - \overline{R}_\gamma(H, P_{H,S}) \\
&\leq \left|\overline{R}_\gamma(H, P_H \otimes \mu^{\otimes n}) - \overline{R}_\gamma(H, P_{H,S})\right| \\
&\leq \frac{(\exp(|\gamma|M)-1)}{|\gamma|}\sqrt{\frac{I(H;S)}{2n}}.
\end{aligned} \tag{111}$$

Therefore, we have

$$\frac{I(H;S)}{\alpha} \leq \frac{(\exp(|\gamma|M)-1)}{|\gamma|}\sqrt{\frac{I(H;S)}{2n}}. \tag{112}$$

Solving (112), results in,

$$\sqrt{I(H;S)} \leq \alpha\frac{(\exp(|\gamma|M)-1)}{|\gamma|}\sqrt{\frac{1}{2n}}. \tag{113}$$

Therefore, we obtain,

$$\left|\overline{R}_\gamma(H, P_H \otimes \mu^{\otimes n}) - \overline{R}_\gamma(H, P_{H,S})\right| \leq \alpha\frac{(\exp(|\gamma|M)-1)^2}{2\gamma^2 n}. \tag{114}$$

Using Theorem H.11, the final result follows. $\qquad \square$

Similar to Corollary H.13, we derive the following upper bound on the absolute value of the expected tilted generalization error of the tilted generalization error.

**Corollary H.16.** *Under the same assumptions in Theorem H.15, the absolute value of the expected tilted generalization error of the tilted Gibbs posterior satisfies*

$$|\overline{\text{gen}}_\gamma(H,S)| \leq \frac{\alpha(\exp(|\gamma|M)-1)^2}{2\gamma^2 n} + \frac{|\gamma|M^2 \exp(|\gamma|M)}{8}\left(1-\frac{1}{n}\right). \tag{115}$$

*Remark* H.17 (Convergence rate). If $\gamma = O(1/n)$, then we obtain a theoretical bound on the convergence rate of $O(1/n)$ for the upper bound on the tilted generalization error of the tilted Gibbs posterior.

*Remark* H.18 (Discussion of $\gamma$). From the upper bound in Theorem H.15, we can observe that under $\gamma \to 0$ and Assumption H.1, the upper bound converges to the upper bound on the Gibbs posterior (Aminian et al., 2021a). For positive tilt ($\gamma > 0$), and sufficient large value of $n$, the upper bound in Theorem H.15, can be tighter than the upper bound on the Gibbs posterior.

In addition to KL-regularized linear risk minimization, the Gibbs posterior is also the solution to another problem. For this formulation we recall that the $\alpha$-*Rényi divergence* between $P$ and $Q$ is given by $R_\alpha(P\|Q) := \frac{1}{\alpha-1}\log\left(\int_{\mathcal{X}}\left(\frac{dP}{dQ}\right)^\alpha dQ\right)$, for $\alpha \in (0,1)\cup(1,\infty)$. We also define the *conditional Rényi divergence* between $P_{X|Y}$ and $Q_{X|Y}$ as $R_\alpha(P_{X|Y}\|Q_{X|Y}|P_Y) := \frac{1}{\alpha-1}\mathbb{E}_{P_Y}\left[\log\left(\int_{\mathcal{X}}\left(\frac{dP_{X|Y}}{dQ_{X|Y}}\right)^\alpha dQ_{X|Y}\right)\right]$, for $\alpha \in (0,1)\cup(1,\infty)$. Here, $P_{X|Y}$ denotes the conditional distribution of $X$ given $Y$.

**Proposition H.19** (Gibbs posterior). *Suppose that $\gamma = \frac{1}{\alpha}-1$ and $\alpha \in (0,1)\cup(1,\infty)$. Then the solution to the minimization problem*

$$P_{H|S}^\alpha = \underset{P_{H|S}}{\arg\inf}\left\{\mathbb{E}_{P_S}\left[\frac{1}{\gamma}\log\left(\mathbb{E}_{P_{H|S}}\left[\exp\left(\gamma\hat{R}(H,S)\right)\right]\right)\right] + R_\alpha(P_{H|S}\|\pi_H|P_S)\right\}, \tag{116}$$

*with $\hat{R}(H,S)$ the linear empirical risk (1), and the Gibbs posterior,*

$$P_{H|S}^\alpha = \frac{\pi_H[\exp(-\gamma\hat{R}(H,S))]}{\mathbb{E}_{\pi_H}[\exp(-\gamma\hat{R}(H,S))]},$$

*where $\pi_H$ is the prior distribution on the space $\mathcal{H}$ of hypotheses.*

*Proof.* Let us consider the following minimization problem,

$$\text{find } \underset{P_Y}{\arg\min}\left\{\frac{1}{\gamma}\log(\mathbb{E}_{P_Y}[\exp(\gamma f(Y))]) + R_\alpha(P_Y\|Q_Y)\right\}, \tag{117}$$

where $\gamma = \frac{1}{\alpha}-1$. As shown by Dvijotham & Todorov (2012), the solution to (117) is the Gibbs posterior,

$$P_Y^\star = \frac{Q_Y\exp(-\alpha f(Y))}{\mathbb{E}_{Q_Y}[\exp(-\alpha f(Y))]}.$$

$\square$

If $\alpha \to 1$, then $\gamma \to 0$ and (116) converges to the KL-regularized ERM problem.

The tilted generalization error under the Gibbs posterior can be bounded as follows.

**Proposition H.20.** *Under Assumption H.1 when training with the Gibbs posterior, (26), the following upper bound holds on the expected tilted generalization error,*

$$\overline{\text{gen}}_\gamma(H,S) \leq \frac{M^2\alpha}{2n} - \frac{\text{Var}(\exp(\gamma\ell(H,Z)))}{2\gamma}\exp(-2\gamma M). \tag{118}$$

*Proof.* Let us consider the following decomposition,

$$\overline{\text{gen}}_\gamma(H,S) = R(H,P_H\otimes\mu) - \mathbb{E}_{P_{H,S}}[\frac{1}{n}\sum_{i=1}^n \ell(H,Z_i)] + \mathbb{E}_{P_{H,S}}[\frac{1}{n}\sum_{i=1}^n \ell(H,Z_i)] - \overline{R}_\gamma(H,P_{H,S}). \tag{119}$$

From Aminian et al. (2021a), for the Gibbs posterior we have

$$R(H,P_H\otimes\mu) - \mathbb{E}_{P_{H,S}}\left[\frac{1}{n}\sum_{i=1}^n \ell(H,Z_i)\right] \leq \frac{\alpha M^2}{2n}.$$

In addition, using Lemma C.4 for uniform distribution, we have

$$\frac{1}{n}\sum_{i=1}^n \frac{1}{\gamma}\log\left(\exp(\gamma\ell(H,Z_i))\right) - \frac{1}{\gamma}\log\left(\frac{1}{n}\sum_{i=1}^n \exp(\gamma\ell(H,Z_i))\right) \leq \frac{-\text{Var}(\exp(\gamma\ell(H,Z)))}{2\gamma}\exp(-2\gamma M).$$

This completes the proof. $\square$

Furthermore, we can provide an upper bound on the absolute value of the expected tilted generalization error under the Gibbs posterior,

$$\left|\overline{\text{gen}}_\gamma(H, S)\right| \leq \frac{M^2\alpha}{2n} + \frac{\max(1, \exp(-2\gamma M))}{8\gamma}(1 - \exp(\gamma M))^2. \tag{120}$$

In (120), choosing $\gamma = O(1/n)$ we obtain a proof of a convergence rate of $O(1/n)$ for the upper bound on the absolute value of the expected tilted generalization error of the Gibbs posterior.

## H.4. Other Bounds

In this section, we provide upper bounds via Rademacher complexity (Bartlett & Mendelson, 2002) and PAC-Bayesian approaches (Alquier, 2021) under bounded loss functions assumption. The results are based on the assumption of bounded loss functions (Assumption H.1).

### H.4.1. RADEMACHER COMPLEXITY

Inspired by the work (Bartlett & Mendelson, 2002), we provide an upper bound on the tilted generalization error via Rademacher complexity analysis. For this purpose, we need to define the *Rademacher complexity*.

As in Bartlett & Mendelson (2002), for a hypothesis set $\mathcal{H}$ of functions $h : \mathcal{X} \mapsto \mathcal{Y}$, the *Rademacher complexity* with respect to the dataset $S$ is

$$\mathfrak{R}_S(\mathcal{H}) := \mathbb{E}_{S,\boldsymbol{\sigma}}\left[\sup_{h \in \mathcal{H}} \frac{1}{n}\sum_{i=1}^{n} \sigma_i h(X_i)\right],$$

where $\boldsymbol{\sigma} = \{\sigma_i\}_{i=1}^{n}$ are i.i.d *Rademacher* random variables; $\sigma_i \in \{-1, 1\}$ and $\sigma_i = 1$ or $\sigma_i = -1$ with probability 1/2, for $i \in [n]$. The *empirical Rademacher complexity* $\hat{\mathfrak{R}}_S(\mathcal{H})$ with respect to $S$ is defined by

$$\hat{\mathfrak{R}}_S(\mathcal{H}) := \mathbb{E}_{\boldsymbol{\sigma}}\left[\sup_{h \in \mathcal{H}} \frac{1}{n}\sum_{i=1}^{n} \sigma_i h(X_i)\right]. \tag{121}$$

To provide an upper bound on the tilted generalization error, first, we apply the uniform bound, Lemma C.11, and Talagrand's contraction lemma (Talagrand, 1996) in order to derive a high-probability upper bound on the tilted generalization error; we employ the notation (121).

> **Proposition H.21.** *Given Assumptions H.1 and assuming the loss function is $M_{\ell'}$-Lipschitz-continuous in a binary classification problem, the tilted generalization error satisfies with probability at least $(1 - \delta)$ that*
>
> $$\widehat{\text{gen}}_\gamma(h, S) \leq 2\exp(|\gamma|M)M_{\ell'}\hat{\mathfrak{R}}_S(\mathcal{H}) + \frac{3(\exp(|\gamma|M) - 1)}{|\gamma|}\sqrt{\frac{\log(1/\delta)}{2n}}.$$

*Proof.* Note that $\exp(\gamma M) \leq x \leq 1$ for $\gamma < 0$ and $1 \leq x \leq \exp(\gamma M)$ for $\gamma > 0$. Therefore, we have the Lipschitz constant $\exp(-\gamma M)$ and 1 for negative and positive $\gamma$, respectively. Similarly, for $\exp(\gamma x)$ and $0 < x < M$, we have the

Lipschitz constants $\gamma$ and $\gamma \exp(\gamma M)$, for $\gamma < 0$ and $\gamma > 0$, respectively. For $\gamma < 0$, we have

$$
\begin{aligned}
\widehat{\mathrm{gen}}_\gamma(h, S) &= \frac{1}{\gamma} \log \left( \mathbb{E}_{Z \sim \mu}[\exp(\gamma \ell(h, Z))] \right) - \frac{1}{\gamma} \log \left( \frac{1}{n} \sum_{i=1}^n \exp \left( \gamma \ell(h, Z_i) \right) \right) \\
&\leq \left| \frac{1}{\gamma} \log \left( \mathbb{E}_{Z \sim \mu}[\exp(\gamma \ell(h, Z))] \right) - \frac{1}{\gamma} \log \left( \frac{1}{n} \sum_{i=1}^n \exp \left( \gamma \ell(h, Z_i) \right) \right) \right| \\
&\leq \frac{1}{|\gamma|} \left| \log \left( \mathbb{E}_{Z \sim \mu}[\exp(\gamma \ell(h, Z))] \right) - \log \left( \frac{1}{n} \sum_{i=1}^n \exp \left( \gamma \ell(h, Z_i) \right) \right) \right| \\
&\overset{(a)}{\leq} \frac{\exp(-\gamma M)}{|\gamma|} \left| \mathbb{E}_{Z \sim \mu}[\exp(\gamma \ell(h, Z))] - \frac{1}{n} \sum_{i=1}^n \exp \left( \gamma \ell(h, Z_i) \right) \right| \\
&\overset{(b)}{\leq} \frac{\exp(-\gamma M)}{|\gamma|} 2\hat{\mathfrak{R}}_S(\mathcal{E} \circ \mathcal{L} \circ \mathcal{H}) + \frac{3\exp(-\gamma M)(1 - \exp(\gamma M))}{|\gamma|} \sqrt{\frac{\log(1/\delta)}{2n}} \\
&\overset{(c)}{\leq} 2\exp(-\gamma M)\hat{\mathfrak{R}}_S(\mathcal{L} \circ \mathcal{H}) + \frac{3\exp(-\gamma M)(1 - \exp(\gamma M))}{|\gamma|} \sqrt{\frac{\log(1/\delta)}{2n}} \\
&\overset{(d)}{\leq} 2\exp(-\gamma M) M_{\ell'} \hat{\mathfrak{R}}_S(\mathcal{H}) + \frac{3(\exp(-\gamma M) - 1)}{|\gamma|} \sqrt{\frac{\log(1/\delta)}{2n}},
\end{aligned}
\tag{122}
$$

where (a) holds due to the Lipschitzness of $\log(x)$ in a bounded interval, (b) holds due to the uniform bound Lemma C.11, (c) and (d) hold due to Talagrand's contraction Lemma C.12).

Similarly, we can prove for $\gamma > 0$, we have

$$
\widehat{\mathrm{gen}}_\gamma(h, S) \leq 2\exp(\gamma M) M_{\ell'} \hat{\mathfrak{R}}_S(\mathcal{H}) + \frac{3(\exp(\gamma M) - 1)}{\gamma} \sqrt{\frac{\log(1/\delta)}{2n}}.
\tag{123}
$$

$\square$

Then, we obtain an upper bound on the generalization error by combining Proposition H.21, Massart's lemma (Massart, 2000) and Lemma C.4.

---

**Theorem H.22.** *Under the same assumptions as in Proposition H.21, assuming a finite hypothesis space, the tilted generalization error satisfies with probability at least $(1 - \delta)$ that*

$$
\begin{aligned}
\mathrm{gen}_\gamma(h, S) \leq &\frac{\max(1, \exp(-2\gamma M))}{8\gamma}(\exp(\gamma M) - 1)^2 + 2AM_{\ell'}B \frac{\sqrt{2\log(\mathrm{card}(\mathcal{H}))}}{n} \\
&+ \frac{3(A(\gamma) - 1)}{|\gamma|} \sqrt{\frac{\log(1/\delta)}{2n}},
\end{aligned}
$$

*where $A(\gamma) = \exp(|\gamma|M)$ and $B^2 = \max_{h \in \mathcal{H}} \left( \sum_{i=1}^n h^2(z_i) \right)$.*

---

*Proof.* We consider the following decomposition for the Rademacher complexity,

$$
\mathrm{gen}_\gamma(h, S) = \mathrm{R}(h, \mu) - \mathrm{R}_\gamma(h, \mu^{\otimes n}) + \mathrm{R}_\gamma(h, \mu^{\otimes n}) - \widehat{\mathrm{R}}_\gamma(h, S),
$$

where $\mathrm{R}(h, \mu) - \mathrm{R}_\gamma(h, \mu^{\otimes n})$ can be bounded using Proposition H.2. The second term can be bounded by using Proposition H.21 and Massart's lemma (Lemma C.13). $\square$

Similar to Remark H.7, assuming $\gamma = O(1/\sqrt{n})$, we have the convergence rate of $O(1/\sqrt{n})$ for the tilted generalization error. For an infinite hypothesis space, covering number bounds can be applied to the empirical Rademacher complexity, see, e.g., (Kakade et al., 2008). We note that the VC-dimension and Rademacher complexity bounds are uniform bounds and are independent of the learning algorithms.

### H.4.2. A STABILITY BOUND

In this section, we also study the upper bound on the tilted generalization error from the stability perspective (Bousquet & Elisseeff, 2002b). In the stability approach, (Bousquet & Elisseeff, 2002b), the learning algorithm is a deterministic function of $S$.

For stability analysis, we define the replace-one sample dataset as

$$S_{(i)} = \{Z_1, \cdots, \tilde{Z}_i, \cdots, Z_n\},$$

where the sample $Z_i$ is replaced by an i.i.d. data sample $\tilde{Z}_i$ sampled from $\mu$. To distinguish the hypothesis in the stability approach from the uniform approaches, we consider $h_s : \mathcal{Z}^n \mapsto \mathcal{H}$ as the learning algorithm. In the stability approach, the hypothesis is a deterministic function $h_s(S)$ of the dataset. We are interested in providing an upper bound on the expected tilted generalization error $\mathbb{E}_{P_S}\left[\text{gen}_\gamma(h_s(S), S)\right]$.

**Theorem H.23.** *Under Assumption H.1, the following upper bound holds with probability at least $(1 - \delta)$ under distribution $P_S$,*

$$\mathbb{E}_{P_S}\left[\text{gen}_\gamma(h_s(S), S)\right]$$
$$\leq \frac{(1 - \exp(\gamma M))^2}{8\gamma}\left(1 + \exp(-2\gamma M)\right) + \exp(|\gamma| M)\mathbb{E}_{P_S, \tilde{Z}}[|\ell(h_s(S), \tilde{Z}) - \ell(h_s(S_{(i)}), \tilde{Z})|]. \tag{124}$$

*Proof.* We use the following decomposition of the tilted generalization error;

$$\mathbb{E}_{P_S}\left[\text{gen}_\gamma(h_s(S), S)\right]$$
$$= \mathbb{E}_{P_S}\left[\text{R}(h_s(S), \mu) - \frac{1}{\gamma}\log(\mathbb{E}_{P_S, \mu}[\exp(\gamma\ell(h_s(S), \tilde{Z}))])\right]$$
$$+ \mathbb{E}_{P_S}\left[\frac{1}{\gamma}\log(\mathbb{E}_{P_S, \mu}[\exp(\gamma\ell(h_s(S), \tilde{Z}))]) - \frac{1}{\gamma}\log\left(\mathbb{E}_{P_S}\left[\frac{1}{n}\sum_{i=1}^{n}\exp(\gamma\ell(h_s(S), Z_i))\right]\right)\right] \tag{125}$$
$$+ \mathbb{E}_{P_S}\left[\frac{1}{\gamma}\log\left(\mathbb{E}_{P_S}\left[\frac{1}{n}\sum_{i=1}^{n}\exp(\gamma\ell(h_s(S), Z_i))\right]\right) - \widehat{\text{R}}_\gamma(h_s(S), S)\right].$$

Using Lemma C.4, we have

$$\mathbb{E}_{P_S}\left[\text{R}(h_s(S), \mu) - \frac{1}{\gamma}\log(\mathbb{E}_{P_S, \mu}[\exp(\gamma\ell(h_s(S), \tilde{Z}))])\right]$$
$$\leq \frac{-\exp(-2\gamma M)}{2\gamma}\text{Var}_{P_S, \mu}(\exp(\gamma\ell(h_s(S), \tilde{Z}))),$$

and

$$\mathbb{E}_{P_S}\left[\frac{1}{\gamma}\log\left(\mathbb{E}_{P_S}\left[\frac{1}{n}\sum_{i=1}^{n}\exp(\gamma\ell(h_s(S), Z_i))\right]\right) - \widehat{\text{R}}_\gamma(h_s(S), S)\right]$$
$$= \frac{1}{\gamma}\log\left(\mathbb{E}_{P_S}\left[\frac{1}{n}\sum_{i=1}^{n}\exp(\gamma\ell(h_s(S), Z_i))\right]\right) - \mathbb{E}_{P_S}\left[\frac{1}{\gamma}\log\left(\frac{1}{n}\sum_{i=1}^{n}\exp(\gamma\ell(h_s(S), Z_i))\right)\right]$$
$$\leq \frac{1}{2\gamma}\text{Var}\left(\exp(\gamma\ell(h_s(S), Z_i))\right).$$

Using the Lipschitz property of the log and exponential functions on a closed interval, we have

$$\left|\frac{1}{\gamma}\log(\mathbb{E}_{P_S, \mu}[\exp(\gamma\ell(h_s(S), \tilde{Z}))]) - \frac{1}{\gamma}\log\left(\mathbb{E}_{P_S}\left[\frac{1}{n}\sum_{i=1}^{n}\exp(\gamma\ell(h_s(S), Z_i))\right]\right)\right|$$
$$= \left|\frac{1}{\gamma}\log(\mathbb{E}_{P_S, \mu}[\exp(\gamma\ell(h_s(S), \tilde{Z}))]) - \frac{1}{\gamma}\log\left(\mathbb{E}_{P_S}\left[\exp(\gamma\ell(h_s(S), Z_i))\right]\right)\right|$$
$$\leq \exp(|\gamma| M)\mathbb{E}_{P_S, \mu}[|\ell(h_s(S), \tilde{Z}) - \ell(h_s(S_{(i)}), \tilde{Z})|].$$

Finally, we have

$$
\begin{aligned}
&\mathbb{E}_{P_S}\big[\mathrm{gen}_\gamma(h_s(S), S)\big] \\
&\leq \frac{1}{2\gamma}\mathrm{Var}\big(\exp(\gamma\ell(h_s(S), Z_i)))\big) - \frac{\exp(-2\gamma M)}{2\gamma}\mathrm{Var}_{P_S,\mu}(\exp(\gamma\ell(h_s(S), \tilde{Z}))) \\
&\quad + \exp(|\gamma|M)\mathbb{E}_{P_S,\mu}[|\ell(h_s(S), \tilde{Z}) - \ell(h_s(S_{(i)}), \tilde{Z})|] \\
&\leq \frac{(1 - \exp(\gamma M))^2}{8\gamma}\Big(1 + \exp(-2\gamma M)\Big) + \exp(|\gamma|M)\mathbb{E}_{P_S,\mu}[|\ell(h_s(S), \tilde{Z}) - \ell(h_s(S_{(i)}), \tilde{Z})|].
\end{aligned}
$$

$\square$

We also consider the uniform stability as in (Bousquet & Elisseeff, 2002b).

**Definition H.24** (Uniform Stability). A learning algorithm is *uniform $\beta$-stable* with respect to the loss function if the following holds for all $S \in \mathcal{Z}^n$ and $\tilde{z}_i \in \mathcal{Z}$,

$$
\big|\ell(h_s(S), \tilde{z}_i) - \ell(h_s(S_{(i)}), \tilde{z}_i)\big| \leq \beta, \quad i \in [n].
$$

*Remark* H.25 (Uniform Stability). Suppose that the learning algorithm is $\beta$-uniform stable with respect to a given loss function. Then, using Theorem H.23, we have

$$
\mathbb{E}_{P_S}\big[\mathrm{gen}_\gamma(h_s(S), S)\big] \leq \frac{(1 - \exp(\gamma M))^2}{8|\gamma|}\Big(1 + \exp(-2\gamma M)\Big) + \exp(|\gamma|M)\beta. \tag{126}
$$

Note that for a learning algorithm with uniform $\beta$-stability, where $\beta = O(1/n)$, then with $\gamma$ of order $O(1/n)$, we obtain a guarantee on the convergence rate of $O(1/n)$.

### H.4.3. A PAC-BAYESIAN BOUND

Inspired by previous works on PAC-Bayesian theory, see, e.g.,(Alquier, 2021; Catoni, 2003), we derive a high probability bound on the expectation of the tilted generalization error with respect to the posterior distribution over the hypothesis space.

In the PAC-Bayesian approach, we fix a probability distribution over the hypothesis (parameter) space as prior distribution, denoted as $Q_h$. Then, we are interested in the generalization performance under a data-dependent distribution over the hypothesis space, known as posterior distribution, denoted as $\rho_h$.

---

**Theorem H.26.** *Under Assumption H.1, the following upper bound holds on the conditional expected tilted generalization error with probability at least $(1 - \delta)$ under the distribution $P_S$; for any $\eta > 0$,*

$$
\begin{aligned}
|\mathbb{E}_{\rho_h}[\mathrm{gen}_\gamma(H, S)]| &\leq \frac{\max(1, \exp(-2\gamma M))(1 - \exp(\gamma M))^2}{8|\gamma|} \\
&\quad + \frac{\eta A^2(\gamma)}{8n} + \frac{(\mathrm{KL}(\rho_h\|Q_h) + \log(1/\delta))}{\eta},
\end{aligned} \tag{127}
$$

*where $A(\gamma) = \exp(|\gamma|M)$, $Q_h$ and $\rho_h$ are prior and posterior distributions over the hypothesis space, respectively.*

---

*Proof.* We use the following decomposition of the generalization error,

$$
\mathbb{E}_{\rho_h}[\mathrm{gen}_\gamma(H, S)] = \mathbb{E}_{\rho_h}[\mathrm{R}(H, \mu) - \mathrm{R}_\gamma(H, \mu) + \mathrm{R}_\gamma(H, \mu) - \widehat{\mathrm{R}}_\gamma(H, S)].
$$

The term $\mathbb{E}_{\rho_h}[\mathrm{R}(H, \mu) - \mathrm{R}_\gamma(H, \mu)]$ can be bounded using Lemma C.4. The second term $\mathrm{R}_\gamma(H, \mu) - \widehat{\mathrm{R}}_\gamma(H, S)$ can be bounded using the Lipschitz property of the log function and Catoni's bound (Catoni, 2003). $\square$

*Remark* H.27. Choosing $\eta$ and $\gamma$ such that $\eta^{-1} \asymp 1/\sqrt{n}$ and $\gamma = O(1/\sqrt{n})$ results in a theoretical guarantee on the convergence rate of $O(1/\sqrt{n})$.

