# OpenReview forum: "Generalization  and Robustness of the Tilted Empirical Risk"
_ICML.cc/2025/Conference — ICML 2025 poster_

### Official Review · Reviewer_Z1pS · 2025-03-09

**Overall Recommendation:** 4

**Summary:**

Building on the notion of tilted empirical risk, this paper develops upper bounds of generalization error for tilted empirical risk (defined as tilted empirical risk minus the regular population risk) under negative tilt and moment-bounded loss.  The first set of results (Theorems 3.5 and 3.11) are for the case where there is no distribution shift, where both uniform convergence bounds and information-theoretic bounds are derived. The second set of results (Theorems 4.3 and 4.5) extend the first set to the case with distribution shift. Under distribution shift, the TV distance between the training and testing distribution arises in the upper bound.  The paper then proceeds to discuss the optimized choice of the tilt value for a given pair of training and testing distributions and experimentally investigate the gain of TERM under optimal tilt with respect to ERM (from Appendix G, the optimal tilt seems to have been obtained via grid search). Finally, the paper studies TERM with KL regularization, deriving Gibbs posterior as the optimal solution (resembling its classical counterpart) and an information-theoretic generalization bound thereof.
Similar results for bounded losses are provided in Appendix H together with PAC-Bayes, Rademacher complexity and stability bounds.

**Claims And Evidence:**

I did not fully verify the proofs, but the proof techniques appear standard and the theoretical results look correct. The experimental results look convincing.

**Essential References Not Discussed:**

The coverage of existing generalization bounds on distribution shift and domain adaptation (BTW, when you want to measure distribution shift, in practice you pretty much deal with the scenario where you are given a labelled training set and an unlabelled testing set with shifted distribution, then you are entering the regime of unsupervised domain adaptation) is not adequate. In fact, there is a large volume of literature in this context, all missing from discussion. Below is a tiny sample and I encourage the authors to dig more.

1. Zou, et al, Towards Robust Out-of-Distribution Generalization Bounds via Sharpness, ICLR2024.

2. Wang and Mao, On f-Divergence Principled Domain Adaptation: An Improved Framework,  NeurIPS2024.

3.  Ye et al, Towards a Theoretical Framework of Out-of-Distribution Generalization, NeurIPS2021

Regarding (in-distribution) information-theoretic and PAC-Bayes generalization bounds,  the paper covers some early references, but more recent developments are largely left out. The authors may wish to consult a recent review article
 Hellstrom, et al, Generalization Bounds: Perspectives from Information Theory and PAC-Bayes, Foundations and Trends in Machine Learning, 2025.

**Experimental Designs Or Analyses:**

The experimental designs and results are valid.

**Methods And Evaluation Criteria:**

Methods and evaluation protocols and criteria look sound to this reviewer.

**Other Comments Or Suggestions:**

The bounds on distribution shift all involve the TV distance of the training and testing distributions.  Without knowing these distributions, the bounds cannot be computed or estimated. It would be much more useful if the authors could derive such bounds where the distribution shift is measured by the training and testing samples (rather than by their distributions), much like those developed in domain adaptation literature, e.g. in the flavor of Theorem 3 in Ben-David and Blitzer ''A theory of learning from different domains'', 2009. If this can be done, one can exploit the optimal choice of the tilt value for the given training and testing samples (rather than for the two distributions, to which we do not have access in practice).

**Other Strengths And Weaknesses:**

The work is comprehensive in its theoretical development on generalization upper bounds related to tilted empirical risk, but somewhat weak in terms of the application and usefulness of their results. Specifically,

1. What benefit can one exploit from this work?

2. Is there any algorithmic improvement that can be derived or inspired from the theoretical results of this paper?

The paper will be greatly improved if results along this line are adequately developed, and I am open to increase my score if these questions are appropriately addressed.

**Questions For Authors:**

See Other Strengths And Weaknesses and Other Comments Or Suggestions.

**Relation To Broader Scientific Literature:**

The paper falls into the broad category of learning theory. It builds upon the work of Li et al, which introduces tilted empirical risk, and serves as its natural and necessary extensions.

**Theoretical Claims:**

I went through all proofs but without tracing every step. The results make sense to me.

---

> ### Author Rebuttal · Authors · 2025-03-31
>
> We thank the reviewer for their comments, and generally positive assessment of the paper. We will address their concerns as detailed below.
>
> > Distribution shift and domain adaptation literature
>
> **R1:** Thank you for introducing the works on generalization bounds under distribution shift and domain adaptation.
> - Our focus in this work is on tilted empirical risk in supervised learning scenarios with clean or noisy training samples. In contrast, domain adaptation typically involves unlabeled samples, which differs from our setting. The type of distribution shift we consider primarily arises from noisy labels or outliers. Nevertheless, we will include the aforementioned references and others to expand the discussion on related work in this space and better contextualize our contributions.
> - One of the key contributions of our work is the study of tilted empirical risk under unbounded loss functions, assuming only a bounded $(1+\\epsilon)$-th moment of the loss. In contrast, works such as [1], [2], and [3] assume bounded loss functions. We hope that our results can be combined with those in [1–3] to extend their applicability to heavy-tailed scenarios. We will include these references in our revision to better situate our work within the existing literature.
>
> Finally, thanks for mentioning [5]. We will include it  in the discussion of the related work.
>
> >What benefit can one exploit from this work? Is there any algorithmic improvement that can be derived or inspired from the theoretical results of this paper?
>
> **R2:** The main aim of this work is to provide a theoretical foundation for the tilted empirical risk which is introduced in [4]. Furthermore, our results help to **bridge a theoretical gap** by showing how tilting affects generalization bounds and excess risks, beyond intuition or heuristics. This can be particularly useful for practitioners designing robust models in supervised learning tasks, including classification, regression, and robustness-aware optimization. For instance, algorithms could:
> - Employ tilted empirical risk as an objective in training robust classifiers or regressors,
> - Use our generalization guarantees to guide regularization schemes. In particular, a tilted Gibbs posterior as a learning algorithm is novel. We consider it as future work to study this algorithm in practice.
> - Tuning $\\gamma$ is a main bottleneck for deploying the tilted loss framework (for example automatic tuning of tilt parameters). We can observe that in many cases  the data-driven $\\gamma$ performs better than the ERM solution in the distribution shift scenario for both Gaussian and Pareto outliers. The data-driven approach for selection of tilt under distribution shift scenario is not proposed in [4]. In the no-distribution shift scenario, we would like to emphasize that TERM with a small negative tilt value for the case of i.i.d. samples had not been explored in prior work including [4], and the usefulness of TERM in this scenario is actually uncovered by the theoretical developments.
>
> > TV distance and other divergences
>
> **R4:** Thanks for raising this point. For this purpose, we use Definition C.4 in [1]:
> $$d\_{\\mathcal{H} \\Delta \\mathcal{H}}(\\mu;\\tilde\\mu)  := 2 \\sup\_{\\mathcal{A}(h) \\in \\mathcal{A}\_{\\mathcal{H} \\Delta\\mathcal{H}}} \\Big| Pr\_{\\mu}(\\mathcal{A}(h)) \- Pr\_{\\tilde \\mu}(\\mathcal{A}(h)) \\Big|$$
>
> where $\\mathcal{H} \\Delta \\mathcal{H}$ is defined as $\\mathcal{H} \\Delta \\mathcal{H} := \\{ h(x) \\oplus h'(x) : h, h' \\in \\mathcal{H} \\}$, $\\mathcal{A}\_{\\mathcal{H} \\Delta \\mathcal{H}}$ represents the  learning algorithm space under the hypothesis $\\mathcal{H} \\Delta \\mathcal{H}$ and $\\oplus$ is the XOR operator, e.g., $\\mathbb{I}(h'(x) \\ne h(x))$.
>
> Then, applying Corollary C.7 in [1] to proof of Proposition 4.2, we have,
>
> $$\\frac{1}{|\\gamma|}\\Big|\\log(E\_{\\mu} \\exp(\\gamma\\ell(h,Z))) \- \\log(E\_{Z\\tilde \\mu}\\exp(\\gamma\\ell(h,Z))) \\Big|\\leq \\frac{d\_{\\mathcal{H} \\Delta \\mathcal{H}}(\\mu; \\tilde \\mu)}{\\gamma^2} \\frac{\\exp(|\\gamma|\\kappa\_u)-\\exp(|\\gamma|\\kappa\_s)}{\\kappa\_u-\\kappa\_s}$$
>
> where $d\_{\\mathcal{H} \\Delta \\mathcal{H}}(\\mu; \\tilde \\mu)$ can be estimated using training and test data samples.  Note that, for linear empirical risk, we cannot derive the result in terms of $d\_{\\mathcal{H} \\Delta \\mathcal{H}}(\\mu; \\tilde \\mu)$ for unbounded loss functions.  We clarify this discussion in the revised manuscript.
>
> ---
> **References:**
> - [1]: Zou, et al, Towards Robust Out-of-Distribution Generalization Bounds via Sharpness.
> - [2]: Wang and Mao, On f-Divergence Principled Domain Adaptation: An Improved Framework.
> - [3]: Ye et al, Towards a Theoretical Framework of Out-of-Distribution Generalization.
> - [4]: Tian Li, et al. On tilted losses in machine learning: Theory and applications.
> - [5]: Hellstrom, et al, Generalization Bounds: Perspectives from Information Theory and PAC-Bayes,

---

> > ### Comment · Reviewer_Z1pS · 2025-04-03
> >
> > I would like to thank the authors for their responses. I have no further questions and, in light of the clarifications provided, I will increase my score.

---

> > > ### Author Response · Authors · 2025-04-06
> > >
> > > Dear Reviewer Z1pS,
> > >
> > > We just wanted to sincerely thank you for taking the time to carefully read our rebuttal and for your thoughtful consideration of our responses. We truly appreciate the constructive feedback you provided throughout the review process, and we are grateful for your support and for the updated evaluation of our work.
> > >
> > > Your detailed comments and suggestions were very helpful to us in improving our paper, and we are glad that our clarifications could address your concerns.
> > >
> > > Thank you again for your time, effort, and support.
> > >
> > > Best regards,
> > >
> > > Authors

---

### Official Review · Reviewer_HKuD · 2025-03-10

**Overall Recommendation:** 4

**Summary:**

This paper gives a detailed and extensive study of the tilted empirical risk, focusing on particular on generalization bounds. Both uniform convergence bounds and algorithm-dependent information-theoretic bounds are provided, and robustness guarantees under distribution shifts are analyzed. On the basis of the theoretical results, a data-driven approach for determining the level of tilt is proposed and evaluated experimentally. Finally, inspired by the information-theoretic bounds, a KL-regularized version is studied and the optimal posterior is determined and analyzed.

## update after rebuttal

I thank the authors for their response. I retain my positive evaluation.

**Claims And Evidence:**

Yes. The experimental results do not necessarily support making strong claims, but the paper does not overstate them either.

**Essential References Not Discussed:**

N/A

**Experimental Designs Or Analyses:**

See above

**Methods And Evaluation Criteria:**

Overall, yes. It is not clear whether the results reported in Table 2 are particularly informative, though, as the risk is very low already for normal ERM. Some applications to real-world datasets could be interesting, but the current approach makes sense in order to illustrate the results. Also, the experiments on linear regression indicate a sizeable gap between the data-driven tilt parameter and the optimal one. Some further discussion of this, and the potential sources of looseness, could be useful. (Presumably related to $n\rightarrow\infty$ step).

**Other Comments Or Suggestions:**

The content of the tables and figures is often small and somewhat blurry. The term “non-linear generalization error” is not particularly specific—using an alternative term could be good if possible.

1. $I_1$, $I_3$, and $I_4$ are used but seemingly no $I_2$.
2. In Thm. 4.5, “if (a) or (b) hold” before the equation is superfluous?
3. $P^*_{H|S}$ doubly defined in Eqs. (23) and (24)
4. Line 419 right column: “boudned”

**Other Strengths And Weaknesses:**

The theoretical and algorithmic contributions of this paper are strong. However, the presentation could be improved. The main body of the paper consists of dense mathematical statements with various conditions and parameters, and not much intuition or analysis is provided. For example, providing the reader with some interpretation of the various cases in Thm 3.11 and the effect on the bound could be beneficial—at present, several shorthand notations need to be unpacked and cases with nested conditions need to be interpreted without hints.

**Questions For Authors:**

1. What is the proper interpretation of the cases in Thm. 3.11 and the $\zeta$ parameter?
2. Can you elaborate on when the proposed $\gamma_{\text{data}}$ is expected to work well or not? Can the approach be altered for finite-data settings?

**Relation To Broader Scientific Literature:**

This paper is nicely positioned in the literature, and covers a lot of previously unaddressed questions in the use of tilted empirical risk minimization. The inclusion of practical guidelines for e.g. parameter selection strengthens this further.

**Theoretical Claims:**

I checked up to App. E in some detail and did not identify any issues.

---

> ### Author Rebuttal · Authors · 2025-03-31
>
> We thank the reviewer for their comments, and generally positive assessment of the paper. We will address their concerns as detailed below.
>
> > Also, the experiments on linear regression indicate a sizeable gap between the data-driven tilt parameter and the optimal one. Some further discussion of this, and the potential sources of looseness, could be useful.
>
> **R1:** In deriving $\\gamma\_{data}$, we employ an asymptotic approach assuming $n \\to \\infty$ ( the sample size approaches infinity). Consequently, this introduces a theoretical gap between the data-driven and the optimal parameter selection. Nonetheless, the data-driven tuned tilt demonstrates superior performance compared to the Empirical Risk Minimization (ERM) approach. Therefore, TERM with data-driven tilt  can be helpful in practice. We will mention in the section on conclusion and limitations that a better and more realistic data-driven approach to selecting $\\gamma$ could be an area for future work.
>
> > What is the proper interpretation of the cases in Thm. 3.11 and the $\\zeta$ parameter?
>
> **R2:** Theorem 3.11 is derived by using the upper and lower bounds on the expected non-linear generalization error (Proposition 3.9 and 3.10). Note that, for the upper bound in Proposition 3.9, due to the sub-exponential assumption, we have two cases. Then, for small values of $\\kappa\_t$ and assuming $\\frac{2I(H;S)}{|\\gamma|^{1+\\epsilon}\\kappa\_t^{1+\\epsilon}}\>n$, we can achieve the convergence rate of $O(1/n^{-\\epsilon/(1+\\epsilon)})$. However, for the lower bound, we introduced the $\\zeta$ parameter to handle the logarithm function as shown in the proof of Proposition 3.10. Similarly to the upper bound, for small values of $\\kappa\_t$ and $\\zeta$, we can expect to achieve again the rate of $O(1/n^{-\\epsilon/(1+\\epsilon)})$.
>
> > Can you elaborate on when the proposed $\\gamma\_{data}$ is expected to work well or not? Can the approach be altered for finite-data settings?
>
> **R3:** To address the finite-data setting, we need to focus on minimizing the bound presented in Theorem 4.3, which involves a complex interplay of exponential, polynomial, and inverse power terms of $\\gamma$. This complexity motivates our consideration of the asymptotic regime, which provides better theoretical insights into the parameter behavior. Empirically, our experiments validate this approach, showing that the data-driven tilt consistently outperforms the Empirical Risk Minimization (ERM) solution—which is desired.
>
> Finally, thanks for pointing out some typos in the draft. They are fixed now.

---

### Official Review · Reviewer_wDGv · 2025-03-11

**Overall Recommendation:** 3

**Summary:**

This paper investigates the generalization error of the tilted empirical risk (TER), a non-linear risk metric for supervised learning introduced by Li et al. (2020). The study focuses on the robustness regime under negative tilt, where TER is used to mitigate the impact of noisy outliers.
The paper provides uniform convergence and information-theoretic bounds on the tilted generalization error (the difference between population risk and tilted empirical risk) for unbounded loss functions with a finite 1+\epsilon-th moment. The paper extends TER’s applicability by analyzing its robustness against noisy training outliers. Theoretical guarantees for TER are provided under distribution shift, showcasing its stability compared to traditional empirical risk minimization. The paper includes experimental evaluations that validate the theoretical bounds.

**Claims And Evidence:**

This paper is primarily theoretical, with detailed proofs provided to support the claims.

**Essential References Not Discussed:**

Not I am aware of.

**Experimental Designs Or Analyses:**

The experiments serve more as a sanity check and are limited to logistic regression and simple linear regression. This is OK, as the primary contribution of the paper is theoretical.

**Methods And Evaluation Criteria:**

The evaluation in Section 5 appears solid. The achieved population risk using the data-driven approach for selecting gamma outperforms ERM in the presence of outlier noise.

**Other Comments Or Suggestions:**

Strength: The results hold under more general conditions of bounded moments, rather than relying on the sub-Gaussian assumption or a bounded loss function.

Weakness: In Section 3.1, only finite hypothesis class bounds are provided, with limited discussion on how to generalize to a continuous hypothesis class.

**Other Strengths And Weaknesses:**

Strength: The results hold under more general conditions of bounded moments, rather than relying on the sub-Gaussian assumption or a bounded loss function.

Weakness: In Section 3.1, only finite hypothesis class bounds are provided, with limited discussion on how to generalize to a continuous hypothesis class.

**Questions For Authors:**

The authors keep emphasizing negative gamma, which I understand is due to the inequality direction, requiring negative gamma to provide a valid upper bound. However, since all the bounds depend on the absolute value of gamma, what is the practical benefit of using a negative gamma? Is there any intuitive theoretical explanation?


For the discussion after Theorem 4.3, does the improvement of TER under distribution shift stem from its performance bound depending on total variation rather than KL divergence? Could the authors elaborate further on why TER with negative tilting is essential for achieving this improvement?

## update after rebuttal
Thanks for the rebuttal. I believe this is a solid paper, and I am maintaining my original score.

**Relation To Broader Scientific Literature:**

The paper extends prior work on tilted empirical risk (Li et al., 2020) by establishing theoretical generalization bounds under negative tilt, connecting to broader studies on robust learning, generalization error analysis, and risk minimization in the presence of outliers and distribution shifts.

**Theoretical Claims:**

I did not verify all the proofs of the theoretical results in detail, but I reviewed the main steps for Proposition 3.2, Lemma 3.7, and Proposition 3.9, and they appear reasonable to me.

---

> ### Author Rebuttal · Authors · 2025-03-31
>
> We thank the reviewer for their comments and generally positive assessment of the paper. We will address their concerns as detailed below.
>
> > finite hypothesis class
>
> **R1:** We thank the reviewer for this valuable comment. Indeed, while Section 3.1 focuses on finite hypothesis class bounds, the approach can be naturally extended to continuous hypothesis classes using well-established techniques.
>
> For example, one can construct an $\\epsilon$-net over the hypothesis space, thereby discretizing the space. In this construction, we select a finite subset $ H' \\subset \\mathbb{R}^m $ such that for every $ h \\in H $, there exists a $ h' \\in H' $ with $ \\|h \- h'\\| \\leq r $. By applying our finite hypothesis result to this discretized set and controlling the approximation error through the Lipschitz property of the loss function, we effectively generalize our bounds to the continuous case. This method is exemplified in \[2\], which demonstrates that “for an uncountable hypothesis space, we can always convert it to a finite one by quantizing the output of the smallest set $ H' $ such that for all $ h \\in H $ there is a $ h' \\in H' $ with $ \\|h \- h'\\| \\leq r $, where the Lipschitz maximal inequality (Lemma 5.7 in \[3\]) is derived using a similar quantization technique.
>
> Alternatively, one may also derive infinite hypothesis space results using VC-dimension arguments in binary classification:
>
> First, we observe that Lemma 2 in \[1\]—a symmetrization lemma—applies to functions of the form $f(Z)=\\exp(\\gamma \\ell(h,Z))\\in\[0,1\]$. Using this lemma, we replace the expected value $E\[f(Z)\]$ with its empirical average over an independent ghost sample. Next, we apply Bernstein's inequality to bound the deviation of the empirical average from the true expectation. In doing so, we substitute the variance term $Var(\\exp(\\gamma \\ell(h,Z)))$ by $|\\gamma|^{1+\\epsilon}E\[|\\ell(h,Z)|^{1+\\epsilon}\]$. The VC-dimension of the hypothesis class appears when we apply a union bound over a finite cover of the function class induced by the hypotheses using Sauer’s lemma to bound the growth function via the VC-dimension. Finally, as in the proofs of Propositions 3.3 and 3.4, we take the logarithm of both sides of the resulting inequality and rearrange the terms to derive the final generalization bound.
>
> Furthermore, we can also apply the similar approach to derive upper bounds based on covering numbers as introduced in \[1\].
>
> We will clarify this discussion in the Appendix of the revised manuscript.
>
> > negative and positive tilt
>
> The theoretical guarantees developed for negative tilt cannot be directly extended to the positive tilt scenario. For more details please check **R4** in response to **Reviewer 8SUL**. Addressing this limitation and exploring positive tilt in TERM is an avenue for future work, as mentioned in the Conclusion section. The primary application of tilted empirical risk for negative tilt is enhancing robustness to outliers. We have expressed the bounds in terms of $|\gamma|$ to facilitate a more intuitive understanding of the results. We will add a remark on positive tilt theoretical challenges.
>
> >  Improvement of TER under distribution shift
>
> Under an unbounded loss function assumption with a bounded $(1+\\epsilon)$-moment for $\\epsilon\\in(0,1\]$, we can derive an upper bound on generalization error using the Tilted Empirical Risk (TER) with a negative tilt. Specifically, by leveraging the negative tilt parameter $\\gamma \< 0$, we ensure the boundedness of $\\exp(\\gamma \\ell(h,Z))$, which allows us to establish an upper bound in terms of total variation distance. The total variation distance is bounded for all outlier distributions. However, the KL divergence can be unbounded. Therefore, the upper bound on TER is bounded for all outlier distributions. In contrast, for linear empirical risk, while can derive an expression for an upper bound in terms of the KL divergence, this expression can be unbounded for some outlier distributions, due to the unboundedness of the KL divergence.
>
> ---
>
> **References:**
>
> [1]: ​​Bousquet, Olivier, Stéphane Boucheron, and Gábor Lugosi. "Introduction to statistical learning theory."
> [2]: A. Xu and M. Raginsky. Information-theoretic analysis of generalization capability of learning
> algorithms.
> [3]: R. van Handel. Probability in high dimension.

---

### Official Review · Reviewer_8SUL · 2025-03-13

**Overall Recommendation:** 3

**Summary:**

This paper studies the generalization error of tilted empirical risk (TERM), a method for fair and robust learning in empirical studies. The paper first studies in-distribution generalization, considering unbounded loss, negative tilt parameters, and finite hypothesis spaces, and gives a convergence rate of $O(1\sqrt n)$ if the second moment of the loss is finite. Subsequently, the analysis is expanded to distribution shifts. The derived bounds include two bias terms: one arising from the use of tilted loss and the other from the total variation between train and test distributions. A data driven approach is proposed to optimize for the tilt to trade-off between in-distribution and out-of-distribution generalization. Finally, the paper examines mutual-information regularized tilted empirical risk minimization, achieving a 1/n convergence rate, which aligns with findings for the Gibbs posterior, but gets rid of their sub-Gaussian assumption.

**Claims And Evidence:**

The comparison between TERM and ERM is not sufficiently addressed. L292-L312 in section 4 claims that an upper bound for ERM in terms of total variation is not feasible. A lower bound for ERM's generalization error would have supported this argument. Additionally, the comparison between generalization gaps are insufficient, because the total excess risk is the sum of the empirical risk and the generalization gap. An upper bound for the total excess risk would consolidate the result.

**Essential References Not Discussed:**

Related works are clearly discussed.

**Experimental Designs Or Analyses:**

Simulation studies are conducted to validate the data-driven selection of the tilt.

**Methods And Evaluation Criteria:**

The insights are not fully addressed for the in-distribution generalization of TERM (section 3). How does its excess risk compare to that of ERM? It is stated in the abstract that TERM has a novel application under no distribution shift. However, its advantage over ERM is not revealed in this case.

**Other Comments Or Suggestions:**

NA

**Other Strengths And Weaknesses:**

This paper is well-written, with coherent logical flow. Propositions serve as proof sketches.

**Questions For Authors:**

Is there an analytical form for the $\gamma$ that minimizes the upper bound in Theorem 4.3?

**Relation To Broader Scientific Literature:**

This paper is the first to establish generalization bounds for TERM, as claimed in the Related Work section. Notably, TERM improves robustness against outliers with negative tilt and improves robustness under subpopulation shift with positive tilt. However, this paper only addresses the negative tilt, thereby somewhat limiting its significance.

**Theoretical Claims:**

I did not observe obvious errors throughout the theory though I have not checked the proofs. The results are built on finite hypothesis class and negative tilts.

---

> ### Author Rebuttal · Authors · 2025-03-31
>
> We thank the reviewer for their comments, and generally positive assessment of the paper. We will address their concerns as detailed below.
>
> >Comparison between TERM and ERM
>
> **R1:** We should clarify that we cannot derive an upper bound for ERM in terms of total variation distance if the loss function is unbounded as it would require bounding  the quantity
> $$\\sup\_{\\ell(h,Z)\\in\\mathcal{L}\_{\\epsilon}}|\\mathbb{E}\_{Z\\sim \\mu}\[\\ell(h,Z)\]-\\mathbb{E}\_{\\tilde Z\\sim \\tilde \\mu}\[\\ell(h,\\tilde Z)\]| $$
>
> where $\\mathcal{L}\_{\\epsilon}$ is the set of loss functions with bounded $(1+\\epsilon)$-th moment.
> Note that if the loss function can take arbitrarily large values, then small differences in distributions ($\\mu$ an $\\tilde \\mu$) can lead to potentially unbounded differences in expected risk.
>
> >Total excess risk
>
> **R2**: Due to space limitations, we only briefly mentioned in Lines 290–292 that *"Using Lemma 3.7, we can derive an upper bound on the excess risk under distribution shift."* Specifically, this upper bound on excess risk is twice the generalization error bound obtained via the uniform convergence approach. As a result, we are able to compare the excess risk bounds for both TER and ER. The upper bound is as follows,
>
> $$
> \begin{aligned}\\mathfrak{E}\_{\\gamma}(\\mu)&\\leq \\frac{4\\exp(|\\gamma| \\kappa\_s)}{(1-\\zeta)|\\gamma|}\\sqrt{\\frac{|\\gamma|^{1+\\epsilon}\\kappa\_s^{1+\\epsilon}B(\\delta)}{n}} \+\\frac{8\\exp(|\\gamma| \\kappa\_s)B(\\delta)}{3n|\\gamma|(1-\\zeta)} \+\\frac{2|\\gamma|^{\\epsilon}}{1+\\epsilon}\\kappa\_u^{1+\\epsilon}+\\frac{2\\mathbb{TV}(\\mu,\\tilde{\\mu})}{\\gamma^2 }\\frac{\\big(\\exp(|\\gamma|\\kappa\_u)-\\exp(|\\gamma|\\kappa\_s)\\big)}{(\\kappa\_u-\\kappa\_s)},
> \end{aligned}
> $$
>
>  where  $B(\delta)= \log(\mathrm{card}(\mathcal{H}))+\\log(2/\delta)$. Note that, under out-of-distribution, our bound on excess risk for TER is in terms of total variation distance. In contrast, we can not derive an upper bound on excess risk of linear empirical risk with unbounded loss function in terms of total variation distance.
>
> > In-distribution generalization of TERM
>
> **R3:** Thank you for your comment. For TERM under no-distribution shift, we conducted experiments in the Appendix using data driven $\\gamma\_{data}$ derived from Theorem 3.5. In particular, in lines 1519-1536 (Appendix G), we provide experiments for logistic regression without outliers. Furthermore, in lines 1616-1619, we provide experiments to show a data-driven approach for linear regression without outliers (no distribution shift). We thus observe that TERM has an application in the no-distribution shift scenario under heavy-tailed distributions.
>
> > Positive tilt
>
> **R4:** Characterizing the generalization of TERM with a positive tilt is not a goal of our paper. We would like to clarify the main challenges of deriving results for TERM performance with positive tilt ($\\gamma\>0$) under unbounded loss functions (\*\*Assumption 4.1\*\*). Certain results and theoretical tools in our work are specific to negative tilt and do not extend to positive tilt under unbounded loss function assumption. Specifically:
>
> - **Bernstein inequality** and **the exponential term $\\exp(\\gamma\\ell(h,z))$**: For positive tilt, the exponential term $$\\exp(\\gamma\\ell(h,z))$$ becomes unbounded under Assumption 4.1. As a result, the Bernstein inequality, as utilized in Theorems 4.2 and 4.3 for negative tilt, cannot be applied to positive tilt. In contrast, for negative tilt, $$\\exp(\\gamma \\ell(h, z))$$ remains bounded even for unbounded loss functions.
>
> - **Lemma C.9:** The inequality
>
> $$\\begin{aligned}\\mathbb{E}\[X\]-\\frac{1}{\\gamma}\\mathbb{E}\[e^{\\gamma X}\]\\leq \\frac{|\\gamma|^{1+\\epsilon}}{1+\\epsilon}\\mathbb{E}\[|X|^{1+\\epsilon}\],\\end{aligned}$$
>
>  which holds for $0\\leq X$ and $\\gamma \< 0$, is not applicable to positive tilt. Notably, the proof of Lemma C.9 relies on the inequality $e^{\\gamma X} \\leq 1 \+ \\gamma X \+ \\frac{|\\gamma X|^{1+\\epsilon}}{1+\\epsilon} $, which is valid only for $\\gamma X \\leq 0.$
>
> Therefore, the theoretical guarantees developed for negative tilt cannot be directly extended to the positive tilt scenario. Addressing this limitation and exploring positive tilt in TERM is an avenue for future work, as mentioned in the Conclusion section.
>
> > analytical form for the $\\gamma$
>
> **R5**: The bound in Theorem 4.3 involves **exponential**, **polynomial**, and **inverse powers** of $\\gamma$. There may be a numerical minimizer for this upper bound, but the complexity of the expression prevents us from solving analytically in a tractable way. For this purpose, we analyzed the behavior of the bound in the asymptotic regime $\\gamma=0$ and $\\gamma\\rightarrow \-\\infty$.

---

### Decision · Program_Chairs · 2025-05-01

**Decision:**

Accept (poster)

**Comment:**

The paper contributes to the analysis of generalization error of tilted empirical risk (TERM). In particular, the paper presents new results with applications to robust learning and learning under distribution shift.

All the reviewers agree of the general merits of the paper and the solid theoretical results presented. I suggest the authors to update the manuscript based on the reviewers' comments. For instance, it would be useful for the readers if the authors discuss cases with infinite hypothesis classes. It would be also good if the authors improve the paper's presentation including interpretations of the main messages offered by the theoretical results.